# Practical Experience of Filtered Tailings Technology in Chile and Peru: An Environmentally Friendly Solution

**Carlos Cacciuttolo Vargas [1],* and Giovene Pérez Campomanes [2]**

1   Civil Works and Geology Department, Catholic University of Temuco, Temuco 4780000, Chile
2   Facultad de Ingeniería, Universidad Continental, Huancayo 12001, Peru; gperezc@continental.edu.pe
*   Correspondence: carlos.cacciuttolo@gmail.com or ccacciuttolo@uct.cl

**Abstract:** In the last 20 years many mining projects around the world have applied a tailings deposition technology named "dry stacking of filtered tailings" at tailings storage facilities (TSFs). This technique produces an unsaturated cake that allows storing this material without the need to manage large slurry tailings ponds. The application of this technology has accomplished: (i) an increase in tailings water recovery, (ii) a reduction of the TSF footprint (impacted areas), and (iii) a decrease in the risk of physical instability, being TSFs self-supporting structures under compaction (such as dry stacks), and (iv) a better regulator and community perception satisfying the need of stable TSFs. This paper presents the main features, benefits, and advances in filtered tailings technology applied in Chile and Peru with emphasis on: (i) filtering technology evolution over the last decade: description of main equipment, advantages, and disadvantages, (ii) design considerations for main TSF geometrical configurations, tailings transport and placement systems, TSF water management, TSF operational and emergency plans, and TSF progressive closure, (iii) operation experiences at site-specific conditions, (iv) technology acceptance in regulatory frameworks, (v) lessons learned and advances, and (vi) new trends and future developments, considering technical, environmental, regulatory frameworks and cost-effective manners.

**Keywords:** filtered tailings; water recovery; compaction; technology performance

## 1. Introduction

Dry climate, water scarcity, community issues, and environmental constraints around the globe, make the efficient use of water an important aspect of mining [1]. For this reason, equipment manufacturers (cyclones, thickeners, filters, pumps, etc.), and consulting engineering firms are innovating to develop new solutions and appropriate technologies to address these challenges [2].

In recent years, the improvements in tailings dewatering technologies (thickening and filtering), have allowed increased water recovery. These technologies have been successfully applied for production rates up to 50,000 mtpd (metric tonnes per day); showing performance improvements on large-scale projects with high ore production rates [1]. In this scenario, there is still a need for more reliable equipment for filtering processes on a large scale, focusing on the tailings water recovery enhancement for its reuse in mining processing. Figure 1 shows the tailings continuum concept [3].

In current Chilean and Peruvian large-scale mining in dry climate areas, most typical tailings disposal schemes consist of conventional or slightly thickened at modest levels of tailings solids weight concentration (Cw 48–52%). Conventional TSFs have dams built of the coarse fraction of tailings obtained by hydrocyclones or have slightly thickened tailings deposits with dams built of borrowed material. Conventional tailing dams may have water recoveries as high as 65–75% in very well-operated TSFs, which means they have appropriate tailings distribution, good control of the pond (volume and location), and adequate seepage recovery. In conventional dams, water at the settling pond is decanted

by floating pumps, or decant towers, and dam seepages are collected by a drainage system and cutoff trench systems. However, a high seasonal evaporation rate can substantially reduce water recovery from the pond area, and infiltration from the pond in contact with natural soil can produce water losses. Some mining operations with this technology are: (i) Cerro Verde (Peru), Cuajone and Toquepala (Peru), Los Pelambres (Chile) and Los Bronces (Chile) [4,5]

Thickened Tailings Disposal (TTD) technology requires more background data than conventional tailings disposal. In the conventional approach, the properties of tailings are fixed by the concentrator plant, whereas in a TTD impoundment, the properties of the tailings and their placement are "engineered" to suit the topography of the disposal area [6]. The behavior of tailings in the two approaches is entirely different. In conventional disposal, tailings segregate as they flow and settle out to an essentially flat deposit, whereas in TTD technology, a sloping surface is obtained. The principal difference is that, in TTD technology, tailings are thickened before discharge to a homogeneous heavy consistency that results in laminar non-segregating flow. In this way, TTD produces high water recovery (80 percent of tailings water recovery) and a self-supporting deposit with sloping sides, requiring small dams. Some mining operations with this technology are: (i) Toromocho (Peru), Constancia (Peru), Centinela (Chile), and Sierra Gorda (Chile) [4,5].

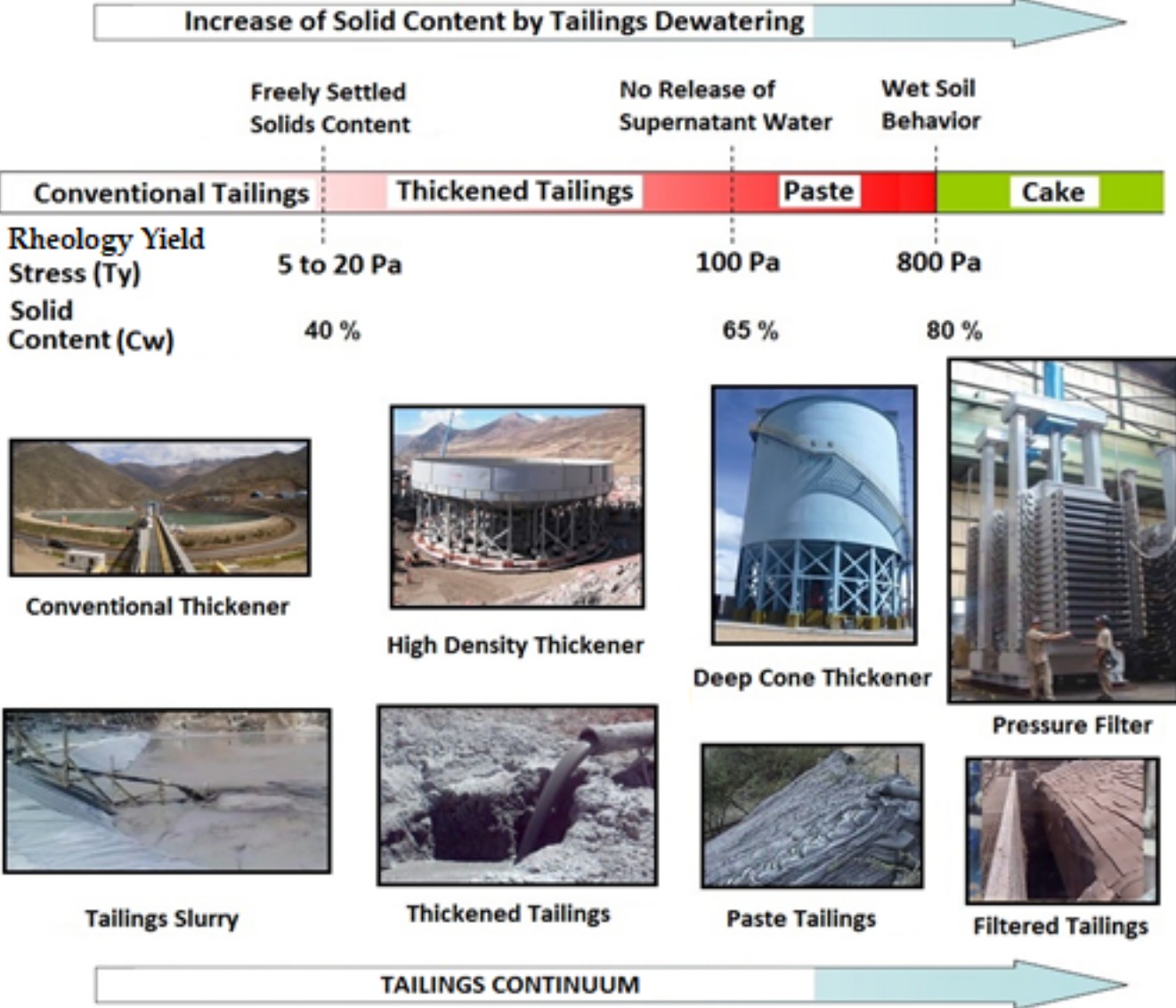

**Figure 1.** Dewatering Tailings Technologies–Tailings Dewatering Continuum [11].

Paste Tailings Technology has been applied on a small production scale because a limitation of equipment manufacturing ability exists. This method permits obtaining a medium make-up water requirement. However, in some cases, there are difficulties in tailings transportation requiring the use of positive displacement pumping, resulting in the highest capital/operating costs. The main advantage of this method is that large dams are not required, only small dams are needed. Some mining operations with this technology are: (i) Chungar (Peru), Cobriza (Peru), Las Cenizas (Chile), and Alhue (Chile) [4,5].

Finally, in the last 20 years, many mining projects around the world have applied a tailings disposal technology called dry stacking of filtered tailings. This technique produces an unsaturated cake that allows storage of this material without the need to manage large slurry tailings ponds. The application of this technology has accomplished: (i) an increase in water recovery from tailings (90 percent), (ii) a reduction of TSF footprint (impacted areas), and (iii) a decrease in the risk of physical instability because TSFs are self-supporting structures under compaction (such as dry stacks), and (iv) a better community perception. Some mining operations with this technology are: (i) Cerro Lindo (Peru), Catalina Huanca (Peru), El Peñon (Chile), and Mantos Blancos (Chile) [4,5].

## 2. Tailings Filtering Equipment Development and Advances

Filter suppliers have gained experience carrying out projects during the last decades, learning that each particle size distribution (PSD) and mineralogy of tailings exhibit their own unique filtering behavior, making efficient and reliable solid/liquid separation units [6–10].

### 2.1. Vacuum Filters

Coarse tailings with low percentages of clay minerals can be handled by vacuum filters with low flocculant consumption. The main characteristics of commercial vacuum equipment are presented below [8,11,12]:

#### 2.1.1. Horizontal Belt Filters

- Equipment Functionality Principle: Vacuum, a filter cloth belt receives tailings and by vacuum suction pressure the filtrate (water) is obtained, and a tailings cake is formed. See Figure 2.
- Equipment Operation Mode: Continuous with shorter dewatering and washing cycles.
- Equipment Capacity Range: 1000–2000 mtpd.
- Equipment Filtration Area/Solids Loading Range: 30–200 $m^2$/300–1000 $kg/h/m^2$.
- Cake Moisture Content Typical Range: 15–25%.
- Application: Gold tailings, cake washing high efficiency, and CN/Au high recovery.
- Flocculant Consumption: 20–40 g/t.
- Sensitivity: Low performance at high elevations (masl), and with high percentages of <74 μm clay minerals.

#### 2.1.2. Ceramic Disc Filters

- Equipment Functionality Principle: Vacuum and capillary action, a ceramic disc with microscopical pore structure rotates on a tank with tailings, and by vacuum suction pressure the filtrate (water) is obtained, and on the disc surface a tailings cake is formed by drying. See Figure 3.
- Equipment Operation Mode: Continuous with shorter dewatering and washing cycles.
- Equipment Capacity Range: 1000–2000 mtpd.
- Equipment Filtration Area/Solids Loading Range: 45–145 $m^2$/575–1500 $Kg/h/m^2$.
- Ceramic Disc Features: Diameter range of 1–5 m, with a maximum of 15 units by tank.
- Cake Moisture Content Typical Range: 12–18%.
- Application: Copper tailings, low energy consumption, high equipment availability.
- Flocculant Consumption: 40–70 g/t.
- Sensitivity: Low performance at high elevations (masl), and with high percentages of <74 μm clay minerals.

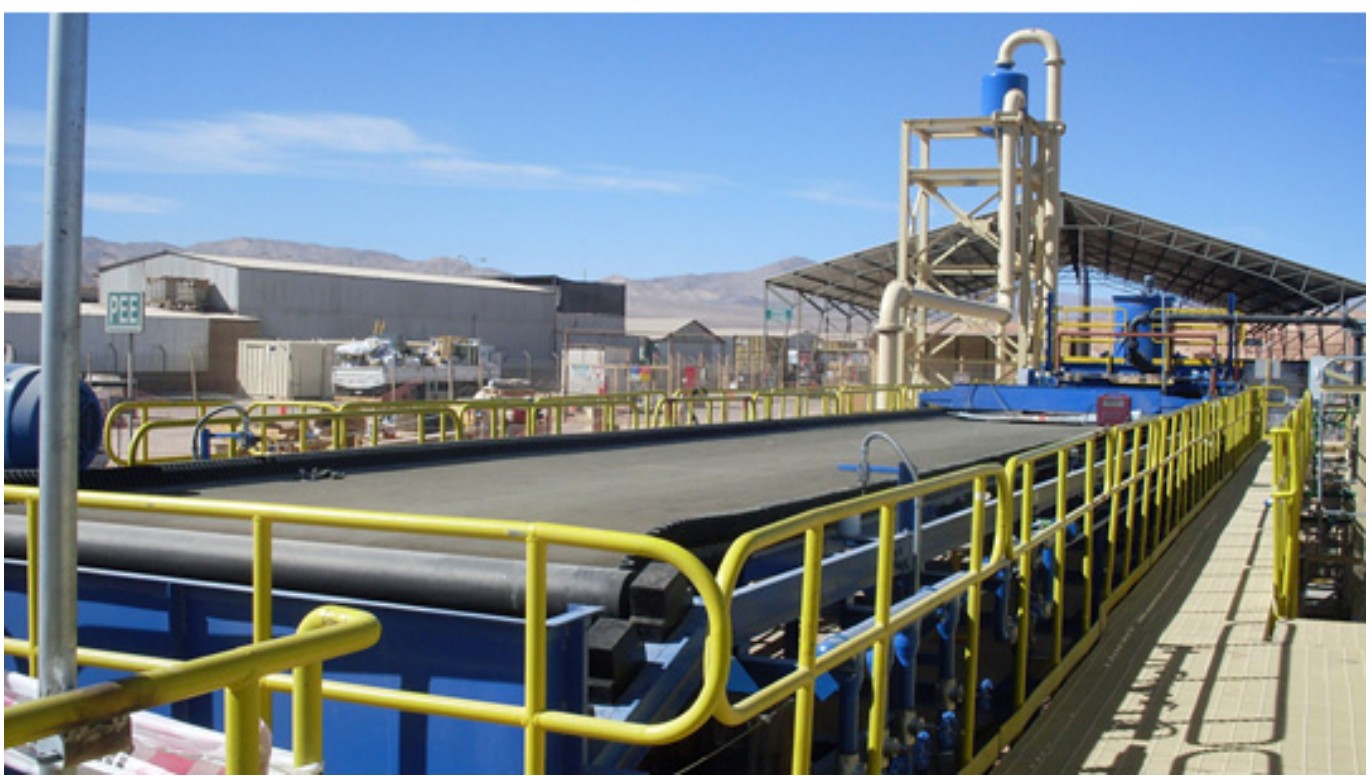

**Figure 2.** Typical Tailings Horizontal Belt Filter [13].

*2.2. Pressure Filters*

Cases where tailings have a fine PSD and require high throughput of tailings are preferentially dealt with by filter press. The main characteristics of commercial industrial filter press equipment are presented below [11,14]:

Vertical Plate and Frame Automatic Filter Press

- Equipment Functionality Principle: Pressure by hydraulic mechanisms, a series of plates are covered with a filter cloth when the plates are pressed and clamped, forming chambers. Tailings are pumped into the chambers, and then filtrate (water) is retired by compression air application. Finally, the filter is opened, the cake is removed and the procedure is repeated. See Figure 4.
- Equipment Operation Mode: Batch, with larger dewatering and washing cycles.
- Equipment Capacity: 1000–14,000 mtpd.
- Equipment Filtration Area/Solids Loading Range: 25–2000 $m^2$/50–1000 Kg/h/$m^2$.
- Cake Moisture Content Typical Range: 10–15%.
- Flocculant Consumption: Not needed, but required filter cloth maintenance per plate.
- Sensitivity: Low equipment availability, which requires stand-by equipment.
- Application: Copper and gold tailings, not sensitive at high elevation (masl).

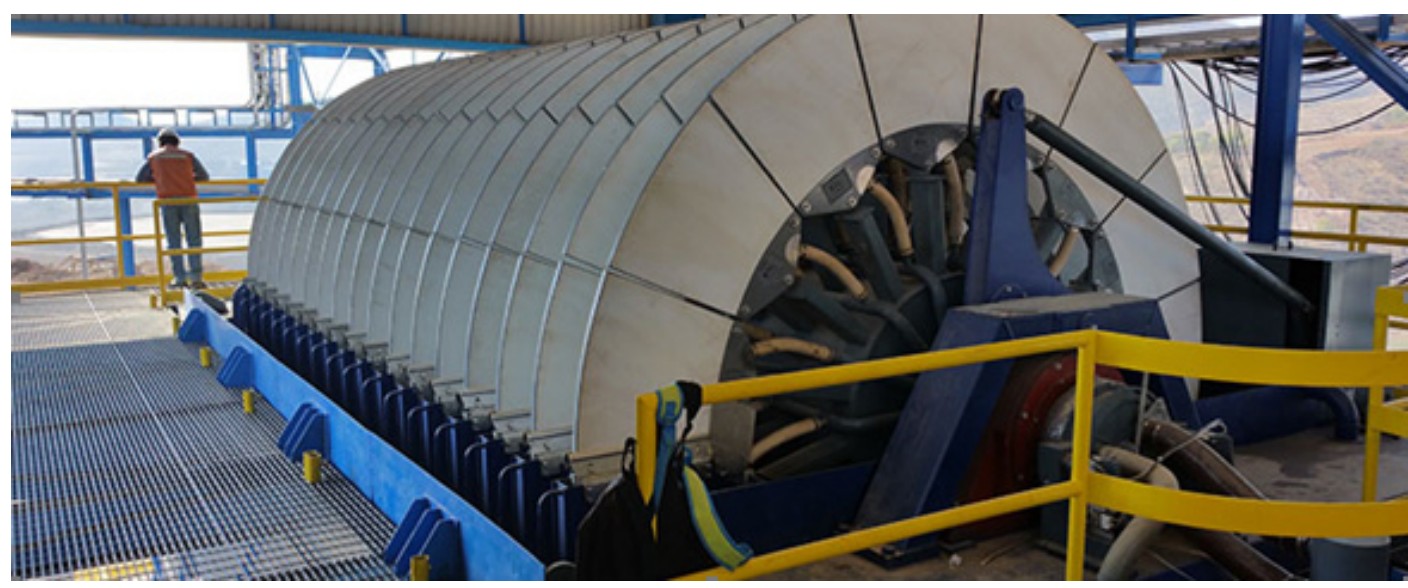

**Figure 3.** Typical Tailings Ceramic Disc Filter [15].

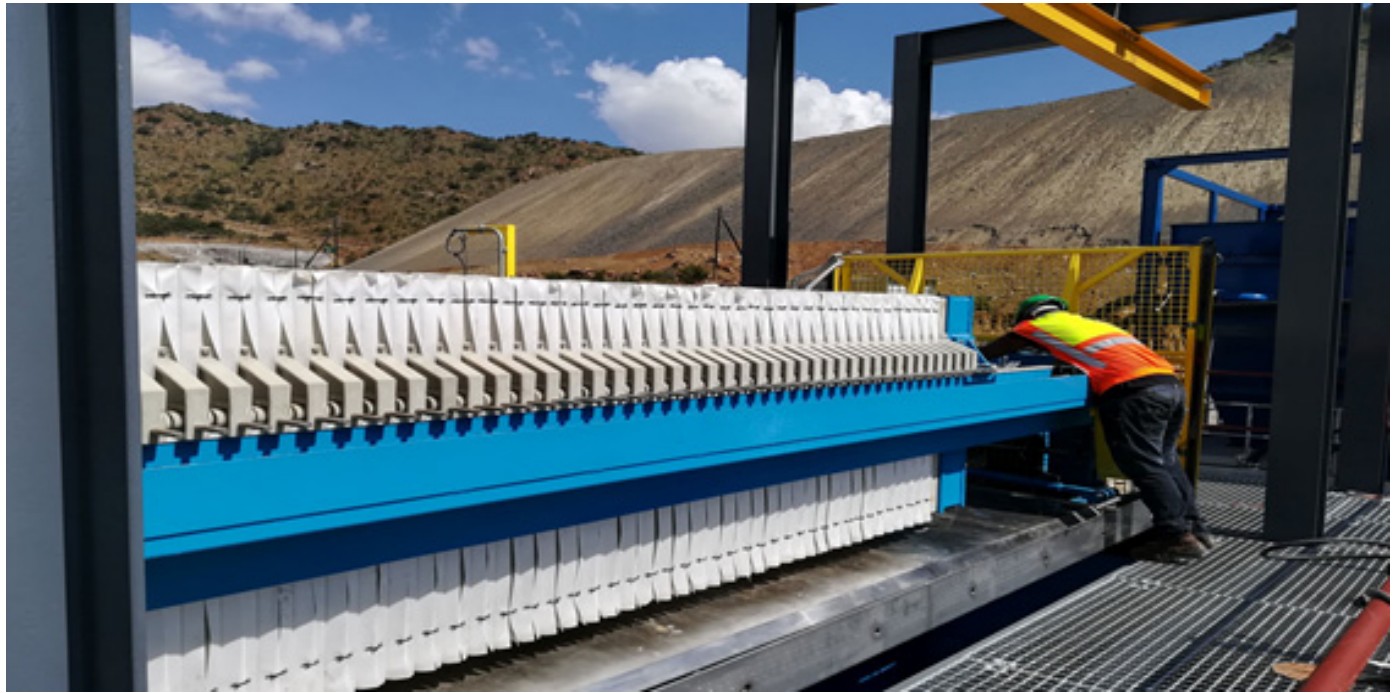

**Figure 4.** Typical Tailings Vertical Plate and Frame Automatic Filter Press [9].

### 3. Filtered Tailings Conveyance System Improvements

The design of a dry stack TSF needs to consider a filtered tailings conveyance system that needs to be compatible with the dry stack construction sequence/plan, using conventional conveyance/haulage and mechanical placement equipment [7,9,16]. The main filtered tailings conveyance types are presented below.

### 3.1. Truck Haulage

Filtered tailings cake is piled at the discharge of the filter plant in a temporary stockpile from where haul trucks (25 tons or 60 tons of capacity depending on the project) are loaded using front-end loaders (Figure 5) to be sent to the tailings deposit [17]. To have safe truck

transport it is necessary to build dual-lane roads that allow connecting the filter plant with the TSF, and provide adequate access roads within the TSF [17].

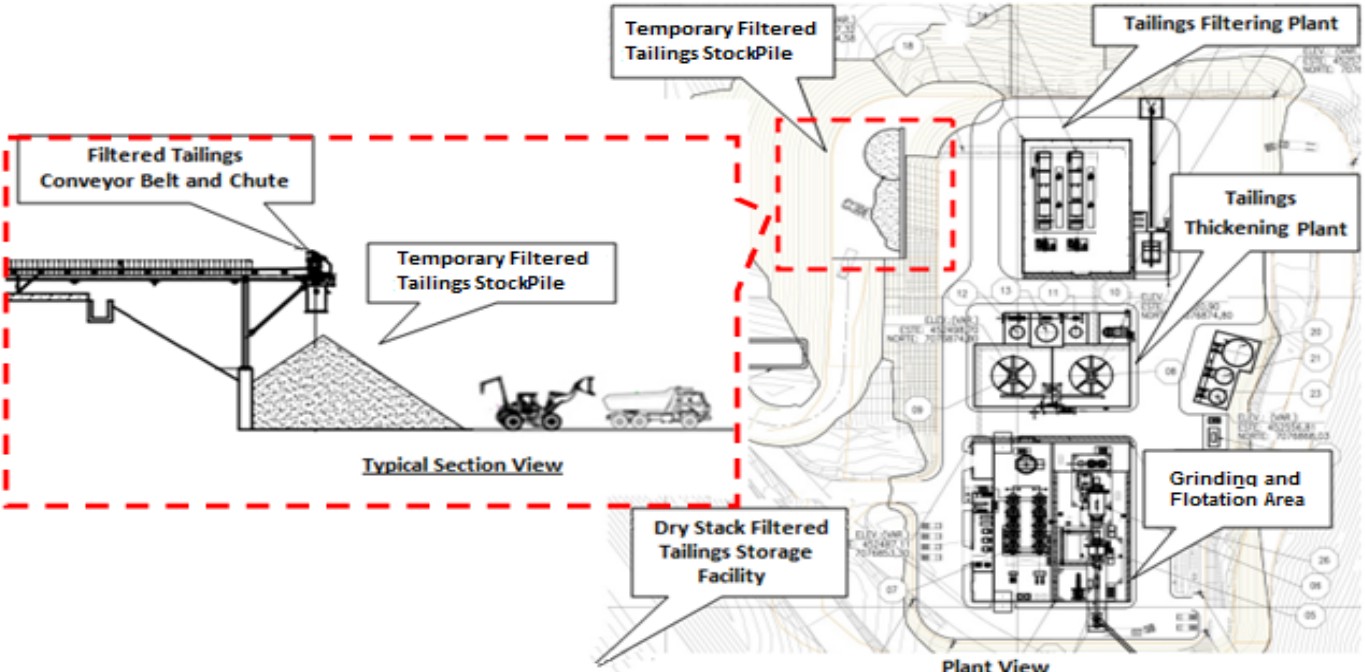

**Figure 5.** Typical Layout and Profile of Filtered Tailings Temporary Stockpile [18].

The main issue associated with the placement of the filtered tailings by truck is usually trafficability. The filtered tailings are generally produced above the optimum moisture content for compaction as determined by the compaction Proctor Standard Test (ASTM D698) [19]. This means that a construction/operating plan is required to avoid trafficability difficulties. Typically, strategies to overcome the low tailings bearing capacity involve some combination of strengthening, use of very small equipment (D6R Dozers), and running the equipment on thin lifts of capping material rather than directly on the tailings [20].

Mining operations with low throughput and filtering technology implemented has the TSF located quite close to the filter plant, and in this case, tailings are usually hauled by truck to the TSF (Figure 6) [7,20–22].

### 3.2. Conveyor Transport

Filtered tailings cake is discharged from the filter plant to a fixed conveyor belt to be sent to the tailings deposit. Then at the TSF, this conveyor belt transfers the cake by a chute or hopper to a mobile conveyor belt system which has the function of disposal of the filtered tailings at the TSF [22,23]. The main typical conveyor systems used to transport filtered tailings are:

- Grasshopper conveyors and spreaders, or shiftable conveyors and radial stackers.
- Mobile stacking conveyor (MSC) with translating tripper boom.

In order to provide adequate conveyor belt mechanical availability and not to have tailings spillage, it is recommended that the moisture content of filtered tailings cakes be lower than about 20% [23,24]. The main issues associated with the placement of the filtered tailings by the conveyor system are usually bearing capacity and conveyor system alignment and positioning control.

MSC offers versatility and flexibility which means that it can move linearly, radially, up or down a slope without having to follow a horizontal line. For the horizontal movement, each MSC bridge is equipped with crawlers to maintain alignment, and hydraulic cylinders to maintain level assisting the passage of the tripper car (Figure 7) [23,24]. These alignment functions are controlled via a set of field instruments coupled to a computerized control

system. Control of movement, as stated previously is achieved via instrumentation and a PLC control system.

In general mining operations with a throughput of over 5000 mtpd implement dry stack TSF with conveyor transport, considering: (i) trafficability constraints, (ii) necessity for use of more truck fleet and increased costs for fuel, and (iii) necessity for a system that allows disposal filtered tailings at high production rates, accomplish the construction program of TSF dry stack (Figure 8) [11,19,20,25].

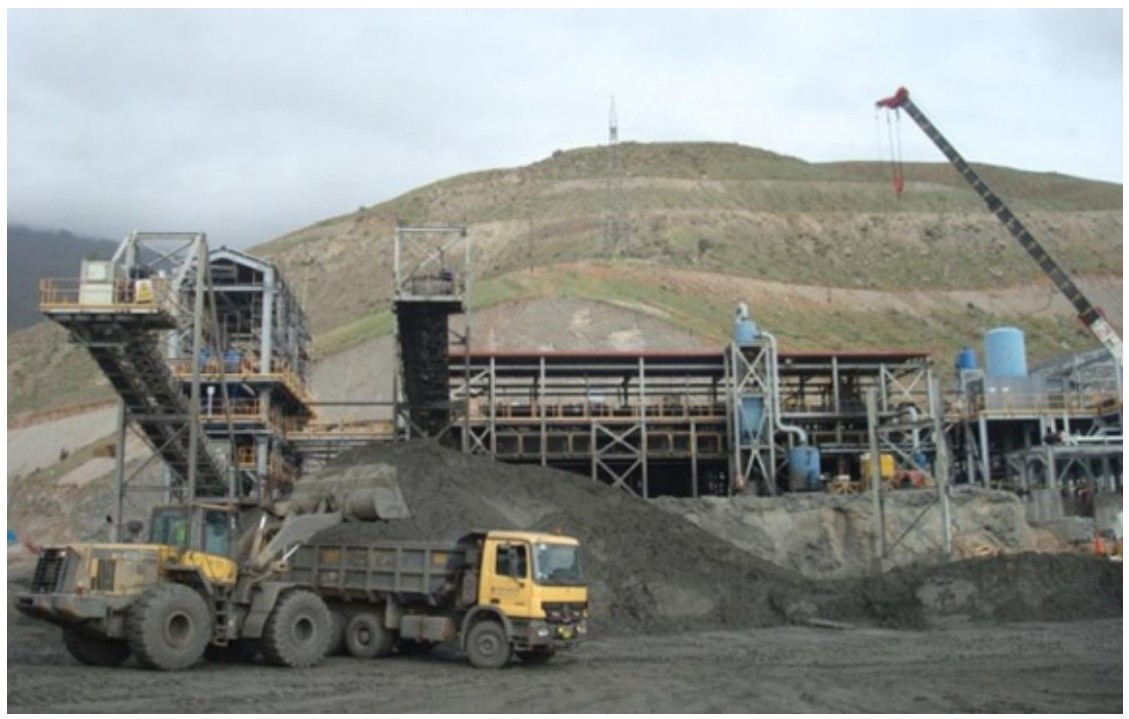

**Figure 6.** Typical View of Temporary Filtered Tailings Stockpile [20].

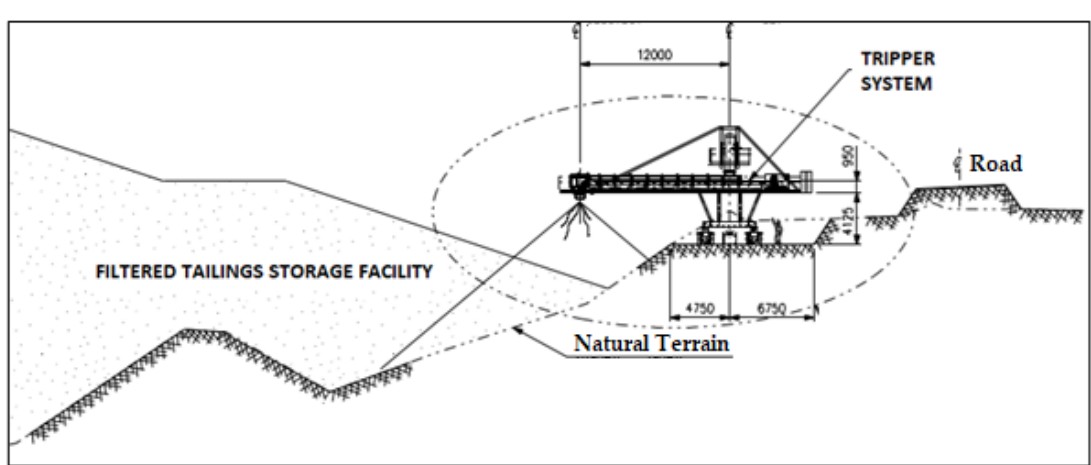

**Figure 7.** Typical View of Filtered Tailings Facility using Conveyor Belt and Tripper System [23].

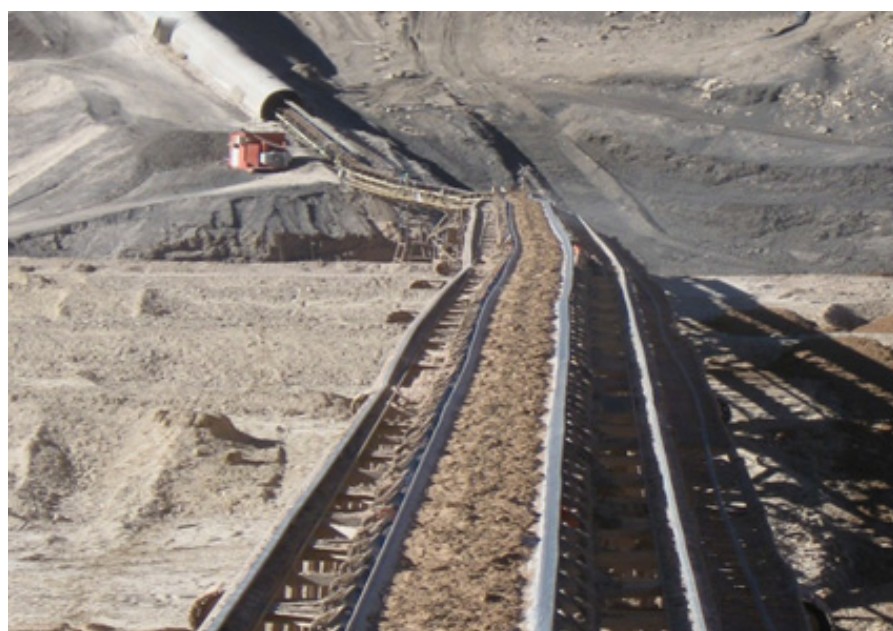

**Figure 8.** Typical View of Filtered Tailings Transported by a Conveyor Belt [23].

## 4. Filtered Tailings Disposal Scheme and Placement Procedures

The design of dry stack filtered TSF geometry needs to be compatible with how the dry stack can be practically constructed using the selected haulage/conveyance and placement equipment. Mechanical re-handling of tailings by dozing and/or compaction with compactors needs to allow the construction of stackable geometries under the land topography of the TSF site [11]. An adequate placement strategy needs to be applied to accomplish the filtered tailings disposal scheme in different zones of the TSF and coordinate the works for progressive TSF closure and land reclamation [7,9].

### 4.1. Filtered TSF Geometry Configuration and Disposal Scheme

The dry stack filtered TSF consists of the placement of successive compacted filtered tailings lifts that form a stable platform, conformed by terraces and berms. The typical design values for the characteristic geometrical elements of dry stack TSF in seismic areas are: (i) local side slopes on order to H:V = 3.0:1.0, (ii) 5 m bench width to provide enough space for vehicle traffic and geotechnical instrumentation installation, (iii) terraces at a maximum height of 5 m, obtaining overall side slopes on order to H:V = 3.5:1.0. Steeper side slopes need a review of the filtered tailings geotechnical parameters to assure the stack stability [11,22].

#### 4.1.1. Flat Topography Configuration

The dry stack TSF in a flat terrain is restricted by the surrounding mine facilities. The construction of this type of dry stack TSF is relatively easy, either on flat terrain or terrain contouring on gentle alluvial fans such as in the Chilean and Peruvian Atacama desert [11]. The filtered tailings are stacked in relatively thin lifts (0.3 m typically). The successive placement of compacted lifts forms the TSF terraces or pads. The construction of the TSF terraces is from the bottom area to the top area, decreasing the filtered tailings placement surface when the dry stack filtered TSF is raising [11].

In some cases, the perimeter slopes of the TSF are designed to provide a perimeter buttress dam, constructed with borrow or waste rock materials. This buttress dam provides physical stability and dust control and is part of the progressive reclamation activities [7,22]. In these cases MSC are preferred for transportation purposes, bulldozers are used for spreading, and smooth drum vibrating compactors for cake compaction (if required).

Figure 9 presents a typical arrangement and disposal scheme of dry stack TSF on flat terrain.

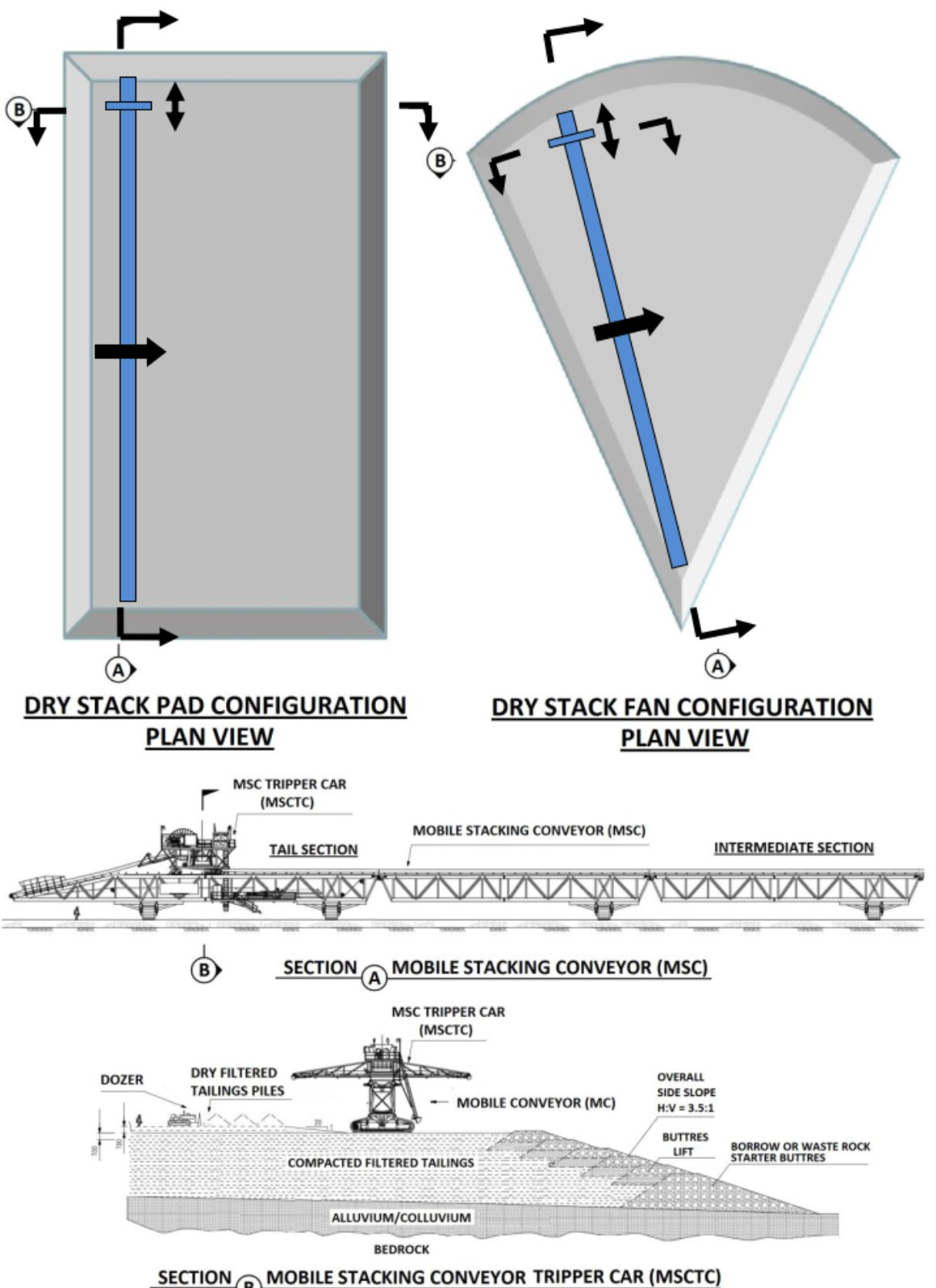

**Figure 9.** Dry Stacked TSF Disposal Scheme with MSC–Construction Method [11].

### 4.1.2. Valley Topography Configuration

The dry stack TSF in an abrupt terrain is restricted by the surrounding land topography. The construction of this type of dry stack TSF needs a strategic plan. In these cases down valley placement or up valley placement methods can be applied [11,20–22]. The up valley placement method considers that filtered tailings are stacked in relatively thin lifts (0.3 m typically). The successive placement of compacted lifts forms the TSF terraces on the valley landform. In these cases, a perimeter buttress dam is provided and has the same function described above [22]. Down valley placement can be applied at upset conditions to disposal in a temporary TSF [11]. Figure 10 presents the arrangement plant view and growth scheme of the filtered tailings dry stack TSF. The dry stack TSF is constructed in the upstream direction of the typical Andes valley topography. The construction of the TSF terraces is from the lower sector to the higher sector of a valley basin, according to the typical geometric characteristics, in order to facilitate an adequate construction with haul trucks, bulldozers, and smooth drum vibrating rollers, assuring the TSF physical stability [11,20–22]. Figure 11 shows an example of filtered tailings dry stack TSF that is a mass composed of seven terraces and berms.

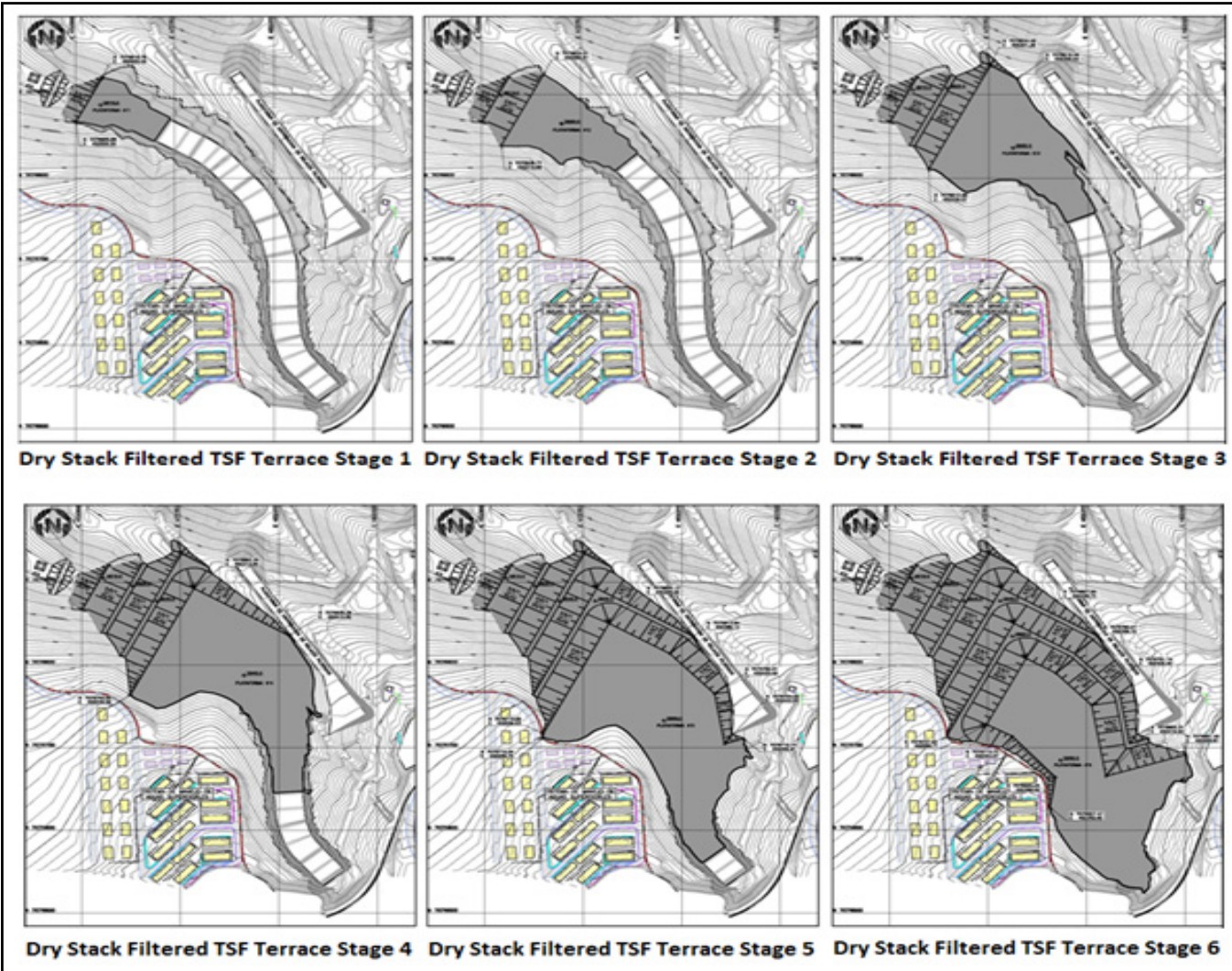

**Figure 10.** Typical Dry Stack TSF Disposal Scheme and Placement—Layout View [18].

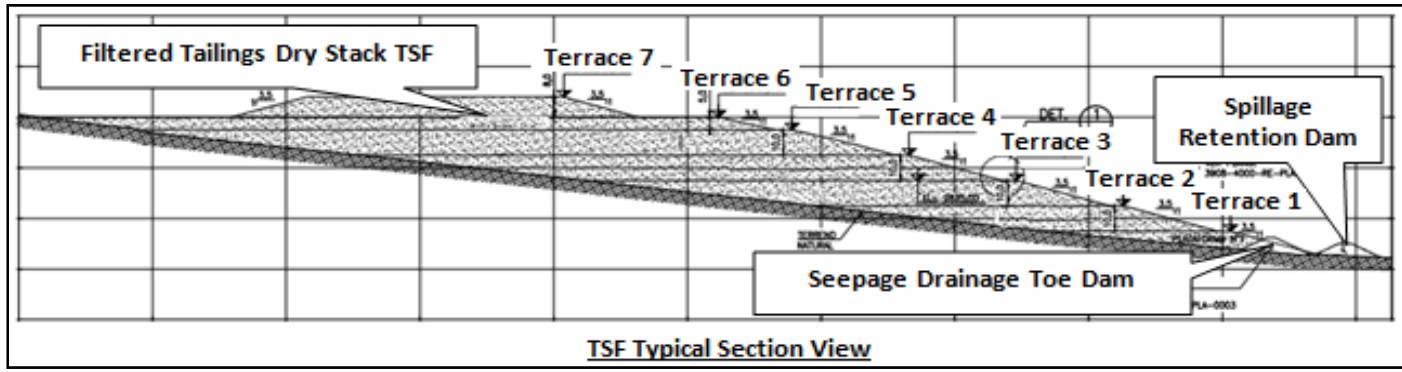

**Figure 11.** Typical Dry Stack TSF Geometrical Configuration—Typical Cross Section View [18].

*4.2. Filtered Tailings Placement Issues*

TSF construction and operating quality assurance and quality control (QA/QC) programs are required to avoid trafficability difficulties, especially at high raising rates of TSF, because trafficability drops as moisture content rises [3,7,9,22].

The filtered tailings are deposited in layers, of 30 cm thickness typically, which are exposed to reach their range of optimum moisture content of 12–14%, and are then compacted to reach 95% Proctor Standard (ASTM D698) [3,7,9,19,22].

Moreover, in high seismic areas, there is often a design requirement to compact the filtered tailings to a higher density at the TSF. QA/QC is applied at the mining operation units [20,25].

4.2.1. Spreading Operation

- Filtered tailings are dumped in piles by haul trucks or MSC tripper boom. Typical moisture content is below 20%, several percent below tailings saturation.
- Dozers or graders spread the filtered tailings piles into 0.3 m loose lift (Figure 12).
- For adequate MSC movement, a good practice is to ensure no excessive temporary filtered tailings piles are laid in front of the tracks crawlers, and improved operations will be obtained by running a grader or dozer in front of the MSC prior to movement.

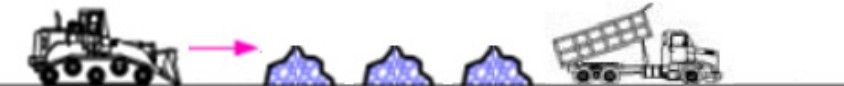

**Figure 12.** Filtered Tailings Spreading Operation–Construction Method.

4.2.2. Drying Activities

- Drying is relevant, different placement zones are required, wet (saturated) zones, and dry (unsaturated) zones. The effects of sun, wind, and the aid of earthmoving machinery allow filtered tailings to dry and reach high compaction densities.
- Filtered tailings should be spread by a dozer for some days after the placement (Figure 13). The drying time depends on cake moisture content, climate, and day or night conditions drying needs to reach a density value near optimum moisture content.

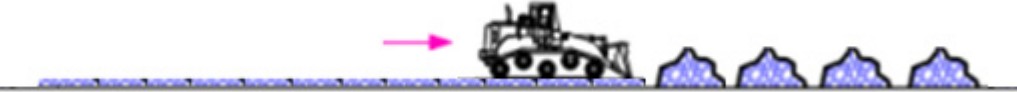

**Figure 13.** Filtered Tailings Drying Operation–Construction Method.

### 4.2.3. Compaction Field Trial

- Smooth Drum Vibratory Compactor equipment is adequate to provide compaction for TSF access roads, filtered tailings lifts, and bearing capacity for dozer, haul trucks, and MSC [11,20].
- After the drying period is accomplished, the compaction (if required) proceeds with passes of a smooth drum vibrator compactor (Figure 14). Compaction field trials are recommended to determine the optimal number of compaction passes to obtain the density on order to 95% of Proctor Standard (ASTM D698) [19], providing, in this manner, adequate bearing capacity for equipment trafficability.

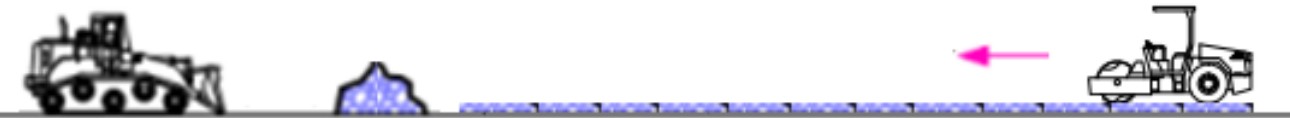

**Figure 14.** Filtered Tailings Compaction Field Trial—Construction Method.

### 4.3. Progressive Rehabilitation/Closure Liability Issues–Dust Emission Control

The lack of a tailings supernatant pond, very low seepage from the unsaturated tailings, and high degree of structural stability, allow dry stacks TSFs to develop progressive reclamation in many instances. A closure cover material is provided to manage runoff erosion, and create an appropriate ground surface for project reclamation (Figure 15) [3,7,11]. Dust emission controls will play an important role as good design and proper implementation will provide the primary control mechanism for dust in accordance with regulatory air quality requirements. Some dust control alternatives are: soil cover, top soil/revegetation cover, binder material, or chemical agglomeration [11].

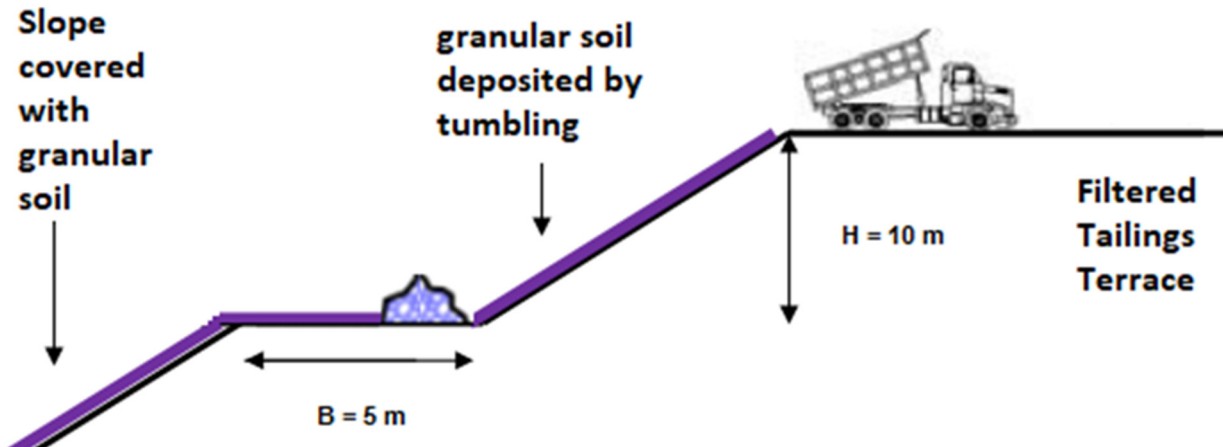

**Figure 15.** Filtered Tailings Storage Facility Covered with Granular Soil–Progressive Closure Activities.

### 4.4. Contingency Plan–Upset Conditions

An Operating, Maintenance, and Surveillance (OMS) Manual is necessary to address adequate procedures to operate, maintain, and monitor the performance of the dry stack TSF, ensuring its operations are in accordance with its design, meet regulatory framework, and provide an emergency and response plan to act during upset conditions [11,18,26]. Some emergency plans and design features are mentioned below.

- Thickener capacity: Thickeners need to be designed with over capacity to constantly feed the filtering plant.
- Contingency TSF site: When the filter plant is under an upset condition event, thickened tailings are diverted to an emergency pond. Once this event is overcome the

thickened tailings are returned to the filtering plant in such an amount to not exceed the filtering design capacity.

- Stand by filter units: Additional filter units are needed to support changes in potential tailings feed upset conditions by ore PSD variability, or mineralogy changes. These additional units are needed to maintain high mechanical availability of the filter plant.
- Stand-by conveyor: It is anticipated that a secondary conveyor system consisting of a bypass diverter or radial stacker conveyor is needed to allow temporary disposal of the filtered tailings stack, for placement with dozers while the primary MSC is inactive due to relocation, maintenance, or upset conditions. Alternatively, additional mobile conveyor systems might be implemented, such as grasshopper conveyors, or spreaders.
- Optimal moisture content handling: The tailings moisture content in the filters discharge can sometimes exceed the maximum moisture value admissible to be placed in the stack. In these cases, treatment moisture reduction is applied by spreading the tailings in thin layers and passing plows to aerate them. This treatment can be performed on the stack footprint provided sufficient room or in a special area in the vicinity of the facility.

## 5. Filtered TSF Physical and Hydrological Stability

To ensure the physical stability and protect the structure of a dry stack TSF, it is necessary to have proper water management at the construction, operation, and closure stages. This means to address the management of (i) the precipitation and runoff (natural water), and (ii) the excess process water [18,22].

### 5.1. Geometric Design Criteria, Tailings Geotechnical Properties, and Physical Stability

The filtered tailings are commonly deposited in layers of 30 cm thickness, which are exposed to reach their optimum moisture content such as 12%, and are then compacted to reach 95% Proctor Standard (ASTM D698) [19]. The filtered dry stack TSF is a mass composed of a number of terraces (or benches) and berms, which needs to be subjected to static and pseudo-static stability analysis (Figure 16). The dry stack TSF is constructed in the upstream direction of a valley topography [11,18–21]. Table 1 describes the main geometric design criteria and geotechnical properties for filtered copper tailings storage facilities commonly registered in mining operations:

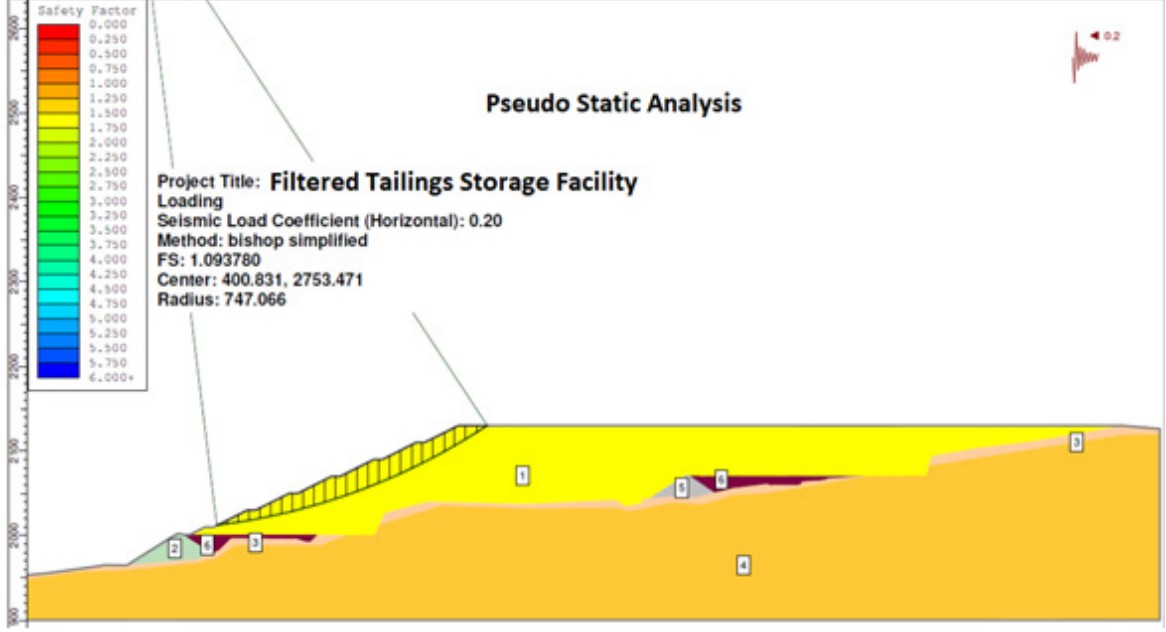

**Figure 16.** Typical Dry Stack TSF Stability Analysis.

**Table 1.** Geometric Design Criteria and Geotechnical Properties of Dry Stack TSF [18,22].

| Geometric Design Criteria of Filtered TSF | | | Geotechnical Properties of Filtered Tailings | | |
|---|---|---|---|---|---|
| **Characteristics** | **Range** | **Unit** | **Characteristics** | **Range** | **Unit** |
| Number of terraces (benches) | 3–8 | - | Particle size distribution $P_{80}$ | 50–90 | (μm) |
| Maximum terrace height | 7–10 | (m) | Solid Specific Gravity | 2.6–4.0 | - |
| Minimum berm width | 2–5 | (m) | Fines Content | 30–60 | (%) |
| TSF terrace local slope | H:V = 3.0:1.0 | - | Permeability | $10^{-5}$–$10^{-6}$ | (cm/s) |
| TSF global slope (all terraces) | H:V = 3.5:1.0 | - | Cohesion/ friction angle–c/φ | 0/25–35 | $(t/m^2)/(°)$ |
| TSF maximum height (all terraces) | 70–80 | (m) | Residual Shear Strength Su/p' | 0.15–0.30 | - |

*5.2. Water Management and Hydrological Stability*

- Considering the use of filtered tailings technology, the physical and hydrological stability of the TSF has to be assured, which implies designing civil works where the surface and underground water flows are managed adequately [3,22]. It is important to maintain these tailings with low moisture levels, close to optimum moisture content, avoiding saturation, to be able to achieve high dry densities and, therefore, adequate shear resistances assuring the stability of the pile [3,22]. The following paragraphs present projected civil works to carry out proper water management at a filtered tailings storage facility.

- Spillage Retention Dam: This civil work is a dam of 5 m in height typically, located downstream of the TSF aimed to contain potential downstream spills of filtered tailings, as a result of potential rainfalls.

- Underdrainage System: It is the base drainage system, formed by an excavated trench with unwoven geotextile, gravel, and filter materials, located at the bottom stream of the valley site, in order to capture potential seepages from the filtered tailings. This system is important due to the collection of potential seepage from tailings or contact waters. Figure 17 presents a typical drain section and hauling road to handle of filtered tailings.

- Seepage Collection Sump: The drainage sump is located typically approximately 5 m downstream of the spillage retention dam, allowing collection of potential seepages from the filtered tailings. This sump is entirely lined with a geotextile and geomembrane. To remove the accumulated water from this sump, one option is to install a submergible pump and recover it to a tank truck for its removal. In case of an excessive accumulation of water by an extreme rainfall event, the sump must have a safety spillway to eliminate excess water.

- Rainfall Diversion Ditches: The filtered TSF projects must include the construction of perimeter channels, one for the right margin, and another for the left margin of the valley. These channels are formed by ditches of trapezoidal section lined, such as concrete cloth ditch liner, with average slopes of 1%, and using corrugated steel pipelines in lengths with slopes over 10%. Both channels collect water runoffs, restricting their entrance to the filtered tailings dry stack deposit, and discharging these flows downstream of the TSF. A collection and diversion system for non-contact water with the TSF (runoff water), consisting of perimeter ditches (lined with geoweb/concrete or precast concrete cloths) (Figure 18).

- An interception system for contact surface water and any impacted groundwater or seepage that may result from the dry stack TSF. "On stack" water should be managed by routing flows to engineered temporary channels by a terrace, sized with grades and sections to convey water and to control erosion and settling. The seepage collection system usually consists of a cutoff and an underdrain system.

- Both systems convey the contact water to collection ponds where water quality is controlled. If the water quality meets the standards, it is possible to discharge it to

natural courses. If it is not, water is treated or pumped back to be reused in the process. Monitoring wells are installed downstream, to control the water quality periodically.

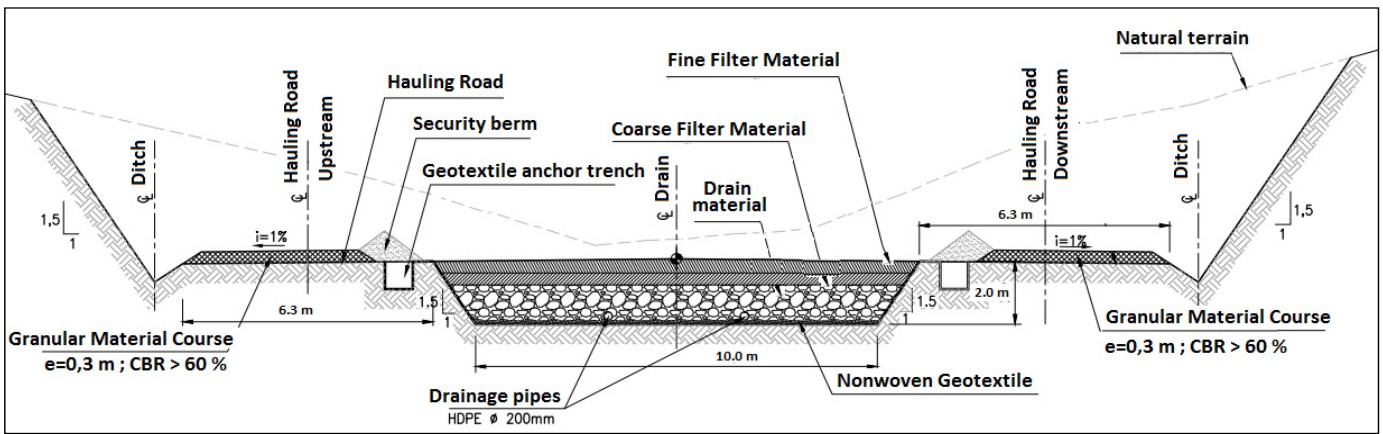

**Figure 17.** Filtered TSF Seepage Underdrainage System—Typical Cross Section [18].

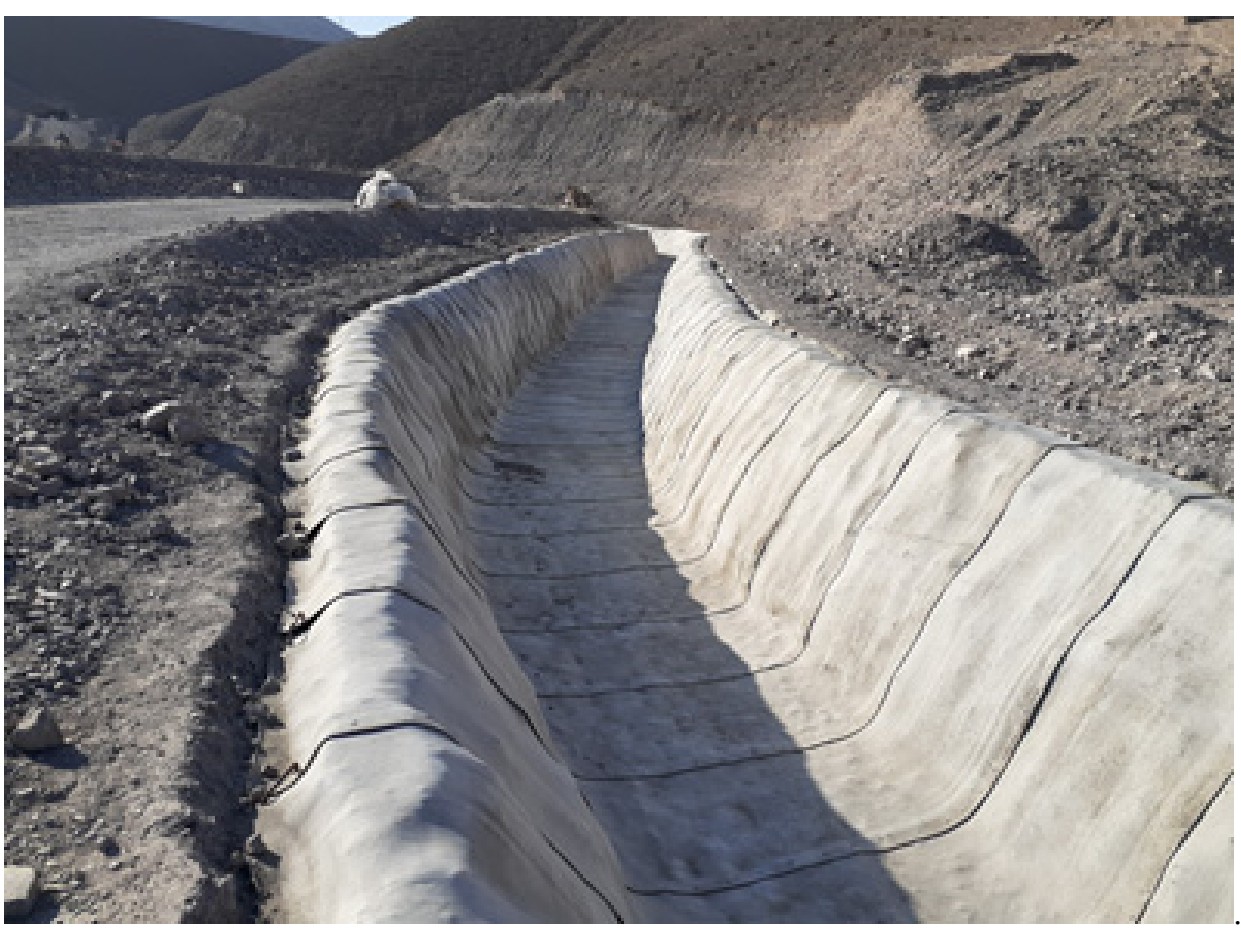

**Figure 18.** Collection and diversion system for non-contact water–perimeter ditches [27].

### 5.3. Operational and Constructability Issues

To construct a stable filtered dry stack tailings deposit, assurance controls of the quality of the construction procedures and materials used are required [20,21]. Operators must control and perform follow-up geometric, geotechnical, and compaction features of deposited filtered tailings, to assure that design specifications are strictly followed during the operation of the project [3,22]. Table 2 shows controls that must be complied with in a QA/QC (Quality Assurance and Quality Control) plan.

**Table 2.** Measurements and Controls of Geometry, Compaction and Geotechnical Properties for Filtered TSFs (QA/QC) [20,25].

| Measurements and Controls of Filtered TSF Geometry | | | |
|---|---|---|---|
| **Type of Control** | **Value** | **Unit** | **Control Frequency Recommendation** |
| Local slope between terraces | 3.0:1.0 | H:V | Monthly |
| TSF Global slope | 3.5:1.0 | H:V | Every 2 constructed terraces |
| Minimum berm width | 2-5 | m | Per terrace |
| Maximum terrace height | 5-10 | m | Per terrace |
| **Measurements and Controls of Filtered Tailings Compaction Parameters** | | | |
| **Type of Control** | **Value** | **Unit** | **Control Frequency Recommendation** |
| Recommended tailings thickness lift | 30–35 | cm | Per compacted tailings lift |
| Maximum dry compacted tailings density | 1.9–2.6 | t/m$^3$ | Per 5.000 m$^3$ of compacted tailings |
| Optimal tailings moisture content | 12–14 | % | Per compacted tailings lift |
| **Measurements and Controls of Filtered Tailings Geotechnical Properties** | | | |
| **Type of Control** | **Value** | **Unit** | **Control Frequency Recommendation** |
| Grain size distribution | ML-CL | - | Monthly |
| Solids specific gravity | 2.65–3.75 | - | Monthly |
| Casagrande piezometer lecture | Phreatic line lecture | m | Monthly per terrace |

## 6. Tailings Filtering Experiences—State of Practice

The following table summarizes experiences of operations that have implemented the dry stack filtered tailings technology. The main characteristics and data of cases are provided in Table 3.

**Table 3.** Filtered Tailings Storage Facility Global Experiences.

| TSF Name | Country | Tailings Throughput (mtpd) | PSD $d_{50}$ ($\mu$m)/Ore Type | Gravimetric Moisture Content w (%)/PSD | Type of Filter/Number of Filter/Filtration Area per Filter ($m^2$) | Transport/ Spread/ Compaction Method | Reference |
|---|---|---|---|---|---|---|---|
| Metales (*) | Mexico | 120,000 | 60/(Au) | 14 (TT) | Pressure Filter/(14)/860 | MSC/D/SDVC | [28] |
| Media Luna | Mexico | 14,000 | 65/(Au) | 14 (TT) | Pressure Filter/(15)/100 | MSC/D/SDVC | [28] |
| El Sauzal | Mexico | 5300 | 60/(Au) | 16 (TT) | Belt Filter/(03)/73 | HT/D/SDVC | [28] |
| Alamo Dorado | Mexico | 3500 | 65/(Au) | 15 (TT) | Belt Filter/(02)/73 | HT/D/SDVC | [28] |
| Greens Creek | USA | 1500 | 55/(P*) | 14 (TT) | Pressure Filter/(02)/70 | HT/D/SDVC | [17,29] |
| Efemçukuru | Turkey | 1500 | 50/(Au) | 15 (TT) | Belt Filter/(02)/23 | HT/D/SDVC | [15,30] |
| Pogo | USA | 2500 | 55/(Au) | 18 (TT) | Pressure Filter/(03)/90 | HT/D/SDVC | [26] |
| Chingola | Zambia | 50,000 | 90/(Cu) | 17(TT) | Belt Filter/(26)/80 | CB-RS/D/NA | [31] |
| Karara | Australia | 50,000 | 85/(Fe) | 15 (TT) | Pressure Filter/(30)/100 | MSC/D/SDVC | [23] |
| Rosemont (*) | USA | 75,000 | 65/(Cu) | 18 (TT) | Pressure Filter/(14)/860 | MSC/D/SDVC | [32] |
| Casposo | Argentine | 1000 | 85/(Au) | 14 (TT) | Belt Filter/(02)/60 | HT/D/SDVC | [6,33–35] |
| Cerro Lindo | Peru | 5000 | 65/(P*) | 12 (SL) | Belt Filter/(02)/73 | HT/D/SDVC | [20,21,25] |
| Poderosa | Peru | 700 | 95/(Au) | 15 (TT) | Pressure Filter/(2)/100 | HT/D/SDVC | [36,37] |
| Curaubamba | Peru | 2000 | 79/(Au) | 15 (TT) | Pressure Filter/(2)/100 | HT/D/SDVC | [38,39] |
| Catalina Huanca | Peru | 1850 | 74/(Cu) | 17 (TT) | Pressure Filter/(2)/100 | HT/D/SDVC | [40,41] |
| Tambomayo | Peru | 1500 | 85/(Au) | 14 (TT) | Pressure Filter/(2)/100 | HT/D/SDVC | [42,43] |
| Chungar | Peru | 4200 | 70/(Cu) | 14 (SL) | Pressure Filter/(3)/100 | HT/D/SDVC | [44] |
| Potrerillos | Chile | 1300 | 60/(Cu) | 12 (SG) | Ceramic Disc Filter/(02)/45 | HT/D/SDVC | [45,46] |
| Mantos Blancos | Chile | 12,000 | 95/(Cu) | 18 (UF) | Belt Filter/(03)100 | CB-RS/D/NA | [13,19,46] |
| La Coipa | Chile | 20,000 | 85/(Au) | 18 (TT) | Belt Filter/(12)/100 | MSC/D/NA | [13,46] |
| El Peñon | Chile | 3500 | 60/(Au) | 20 (TT) | Belt Filter/(04)/54 | HT/MG/T | [13,46] |
| El Gato | Chile | 5500 | 74/(Cu) | 16 (TT) | Ceramic Disc Filter/(02)/60 | HT/D/SDVC | [46,47] |
| Tambillos | Chile | 3000 | 84/(Cu) | 16 (TT) | Ceramic Disc Filter/(03)/60 | HT/D/SDVC | [46,48] |
| Tambo de Oro | Chile | 750 | 85/(Au) | 14 (TT) | Pressure Filter/(1)/100 | HT/D/SDVC | [46,49] |
| Salares Norte | Chile | 5500 | 80/(Au) | 15 (TT) | Pressure Filter/(3)/100 | HT/D/SDVC | [46,50] |
| El Espino | Chile | 20,000 | 84/(Cu) | 15 (TT) | Pressure Filter/(15)/100 | MSC/D/SDVC | [46,51,52] |
| El Indio | Chile | 3000 | 80/(Au) | 20 (TT) | Pressure Filter/(2)/100 | HT/D/SDVC | [46,53] |
| Huasco | Chile | 5000 | 75/(Fe) | 20 (TT) | Pressure Filter/(4)/100 | HT/D/SDVC | [46,54,55] |

**Nomenclature**: PSD: Particle Size Distribution, TT: Total Tailings, UF: Cycloned Tailings Sand, SL: Slimes, SG: Slag Copper Tailings, P*: Polymetallic Ore (Cu-Pb-Zn), CB-RS: Conveyor Belt–Radial Stacker, MSC: Mobile Stacking Conveyor, HT: Haul Truck, D: Dozer, MT: Motor Grader, T: Tractor, SDVC: Smooth Drum Vibratory Compactor, and NA: Not Applied, compaction by a D7R Dozer self-weight. (*): project under regulator environmental/construction permits process.

## 7. Successful Cases in Chile and Peru

The development of filtering technologies has changed the criteria used in evaluating the benefits of increased tailings density. Nowadays, considering a more stringent regulatory framework, more concentrator plants apply tailings filtering technologies together with optimized tailings disposal schemes to recover water, minimize TSF footprints, and comply with regulations. The following paragraphs present various successful cases in South America specifically in mining projects in Chile and Peru.

*7.1. La Coipa Filtered TSF–Valley Topography Configuration–Atacama Desert–Chile*

La Coipa gold and silver mine treated 20,000 tonnes of ore per day at the processing facilities, however, nowadays this mine is under closure stage. The mine is located in Chile´s Region III, in the Atacama Desert area of the Chilean Andes, at 3800 m above sea level, roughly 130 km from Copiapó City. The climate is typical of Andes mountain region conditions. Annual precipitation is approximately 50 mm with essentially this entire amount falling in the winter months (May–September) as snow [56,57].

Tailings were processed in a filtration plant by 12 vacuum belt filters, obtaining a cake with 18% moisture content, and then transported by conveyor belt system to the main TSF or auxiliary TSF [56,57].

The conveyor belt system had a 1700 m long fixed conveyor belt and an 825 m long MSC, which delivered the filtered tailings at the main TSF. It is important to mention that the conveyor belts cited had a negative slope that allows power generation, thus contributing to global savings for the tailings transport system energy consumption [56,57].

The main TSF which approximately parallels the original topography is developed with an up valley placement method where the MSC rotates around a fixed point (pivot) in one direction and in the other and allowed the MSC tripper to dispose of the tailings along the slope, forming a fan shape TSF. Dozers carried out spreading and compaction activities on filtered tailings lifts of average 20–30 cm in thickness (Figure 19) [56,57].

Individual lifts include an inwardly sloping bench which creates zero lift thickness and intersection with the natural ground. Each lift creates an inter-ramp slope (step-in) to create the overall slope angle prescribed by geotechnical design. Until the surface of the TSF lift is uniformly tilted about 5% upstream, the MSC rotation point is changed upstream, shortening the transfer belt approximately 75 m to start the construction of a new TSF lift [13]. The following figure shows the disposal of filtered tailings with MSC at the main TSF.

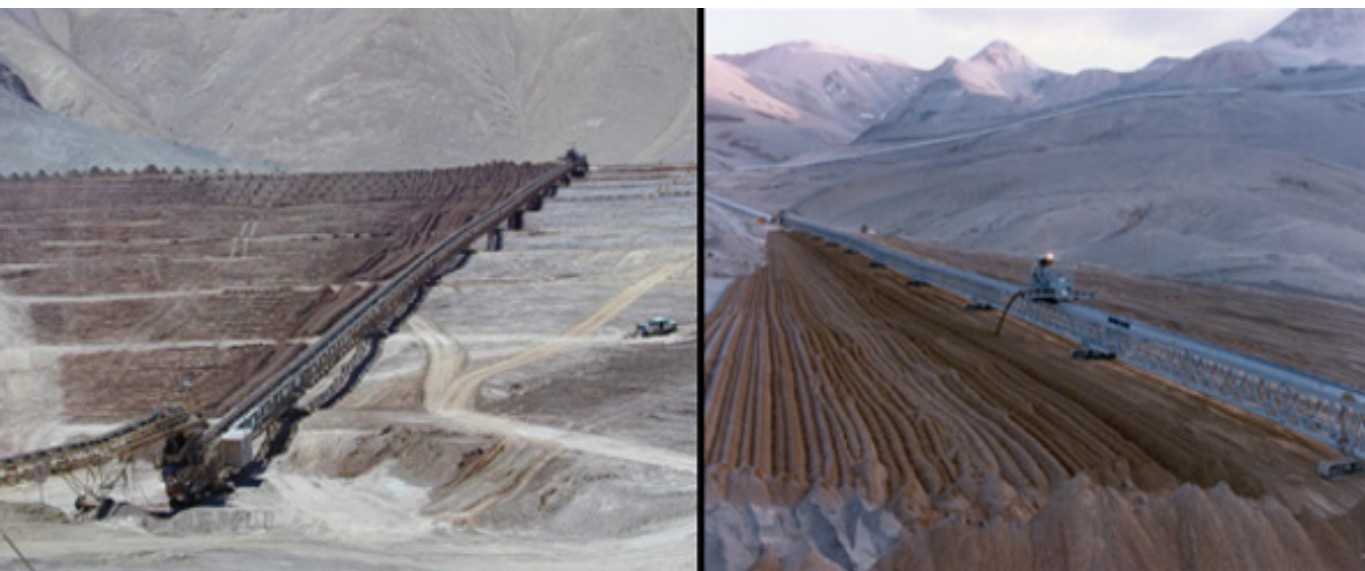

**Figure 19.** La Coipa Mobile Stacking Conveyor (MSC) at Rako TSF [24,58].

The auxiliary TSF is developed by the down valley placement method applied at MSC upset conditions, where saturated filtered tailings flow along roughly repose angle slopes, buttressed at the toe by a containment dyke and downstream supported by a sedimentation collection pond (Figure 20) [56,57].

Nowadays, the main TSF and auxiliary TSF are developing closure and post-closure plans, and carrying out reclamation programs allowing for the re-establishing of native vegetation and landforms [56,57].

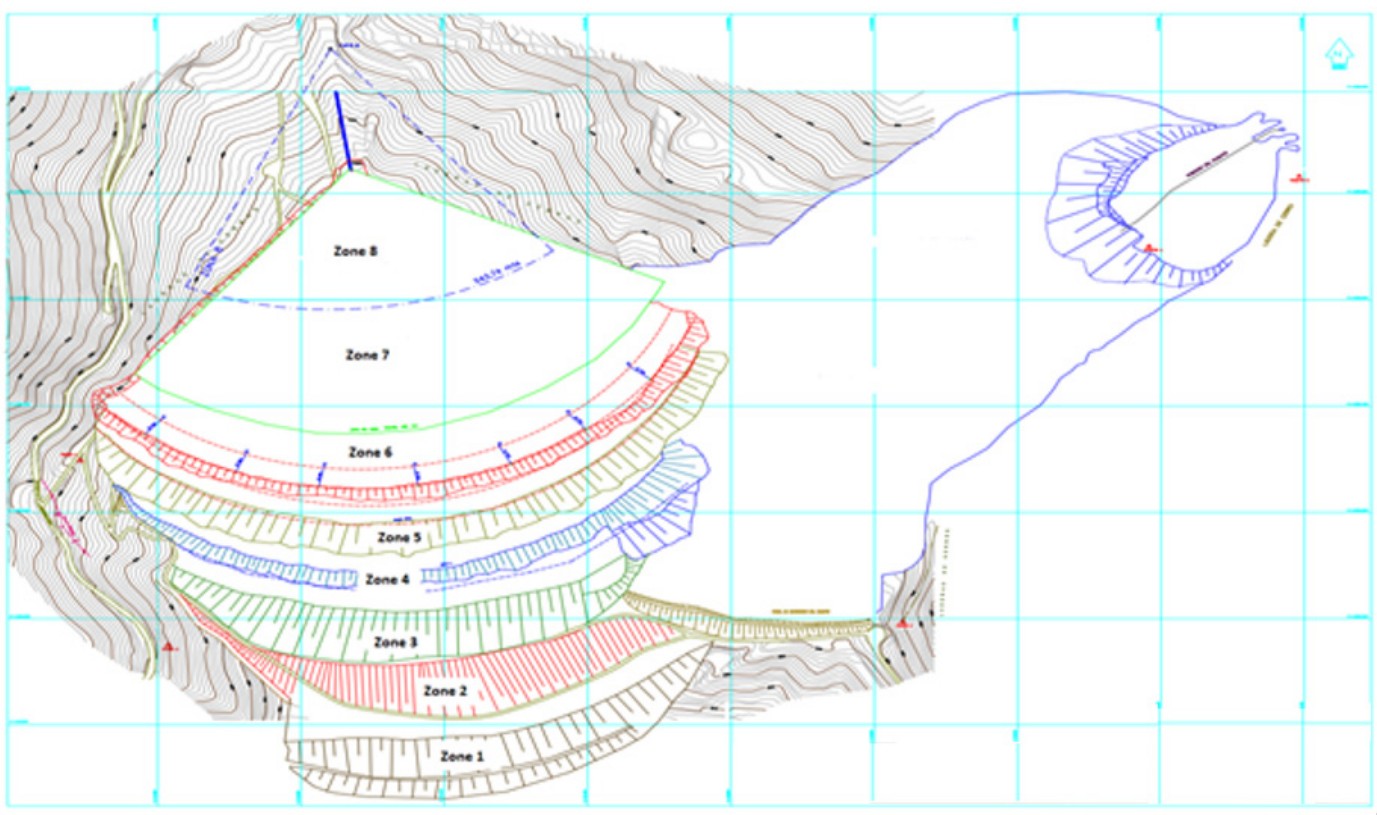

**Figure 20.** La Coipa Rako TSF Layout [24,58].

### 7.2. El Indio–Dry Stack of Filtered Tailings–Andean Region of Chile

El Indio copper-gold mine is located in the Chilean Andes Area approximately 180 km east of La Serena City. Actually, the El Indio mine is in post-closure phase. The mine processed 3000 tonnes per day (tpd) of ore for obtaining polymetallic minerals gold, silver, and copper. The climate is typical of the Andes mountain region of dry conditions, with variable winter precipitations (May–September), mainly as snow. During the operation phase, the mine had an open pit and underground activities, with two waste rock dumps, three process/metallurgic plants, and three tailings storage facilities.

Final tailings from the mill are thickened to 50% to 60% solids by two 30.5 m diameter thickeners. Approximately 50% of the thickened tailings are filtered by two Edwards and Jones pressure filters. Filter cake containing approximately 18% to 22% moisture is conveyed by belt conveyors and front end loader to deposition modules where they are stabilised by compacting. The remaining final tailings, which are not filtered, flow by gravity to a tailings pond. Water reclaimed from the tailings pond and thickener overflows is pumped to a reservoir for reuse in the mill [53,59].

Tailings and waste facilities managed in El Indio Mine were: (i) Pastos Largos TSF, (ii) El Indio TSF, (iii) dry tailings modules (filtered tailings), and (iv) sedimentation pond (polishing pond). Figures 21 and 22 show the filtered dry stack tailings modules during the operating phase, located at Malo river stream [53,59].

The mining company negotiated a voluntary agreement with the Chilean Region IV regulatory authorities after 20 years of mine lifetime to carry out the closure stage of the El Indio Mining project, as there was no legislation yet in place in Chile focused on mine closure. One of the key components of the closure plan was surface water management works developed with the overall objective to "establish a physically and chemically stable drainage system with minimal maintenance and monitoring requirements" (Robledo and Meyer, 2007). As a consequence of mining activities, the Malo River stream was modified in several sectors with diversion civil works. The main closure activities were: (i) restoration

of the Malo River drainage system in the process plant area and on TSFs by the construction of engineered lined channels (lined with rockfill and cobblestones), and (ii) abandonment of the existing Malo river diversion system [53,59].

Tailings and waste facilities managed in El Indio Mine were: (i) Pastos Largos TSF, (ii) El Indio TSF, (iii) dry tailings modules (filtered tailings) (Figure 22), and (iv) sedimentation pond (polishing pond) (Figure 23). Specific closure works at the TSFs were considered including surface grading, placement of a cover to prevent hydraulic and wind erosion, and the construction of spillways to manage storm flows. The sedimentation pond is located at Malo River downstream of TSFs and stores approximately 66,000 m$^3$ of sediments (As, Pb, and cyanide among others). Sediments were transported by haul trucks to Pastos Largos TSF for disposal. The pond was removed and the Malo River was restored to its natural stream. Tailings were processed in a filtration plant by press filters, obtaining a cake with 18% of moisture content [53].

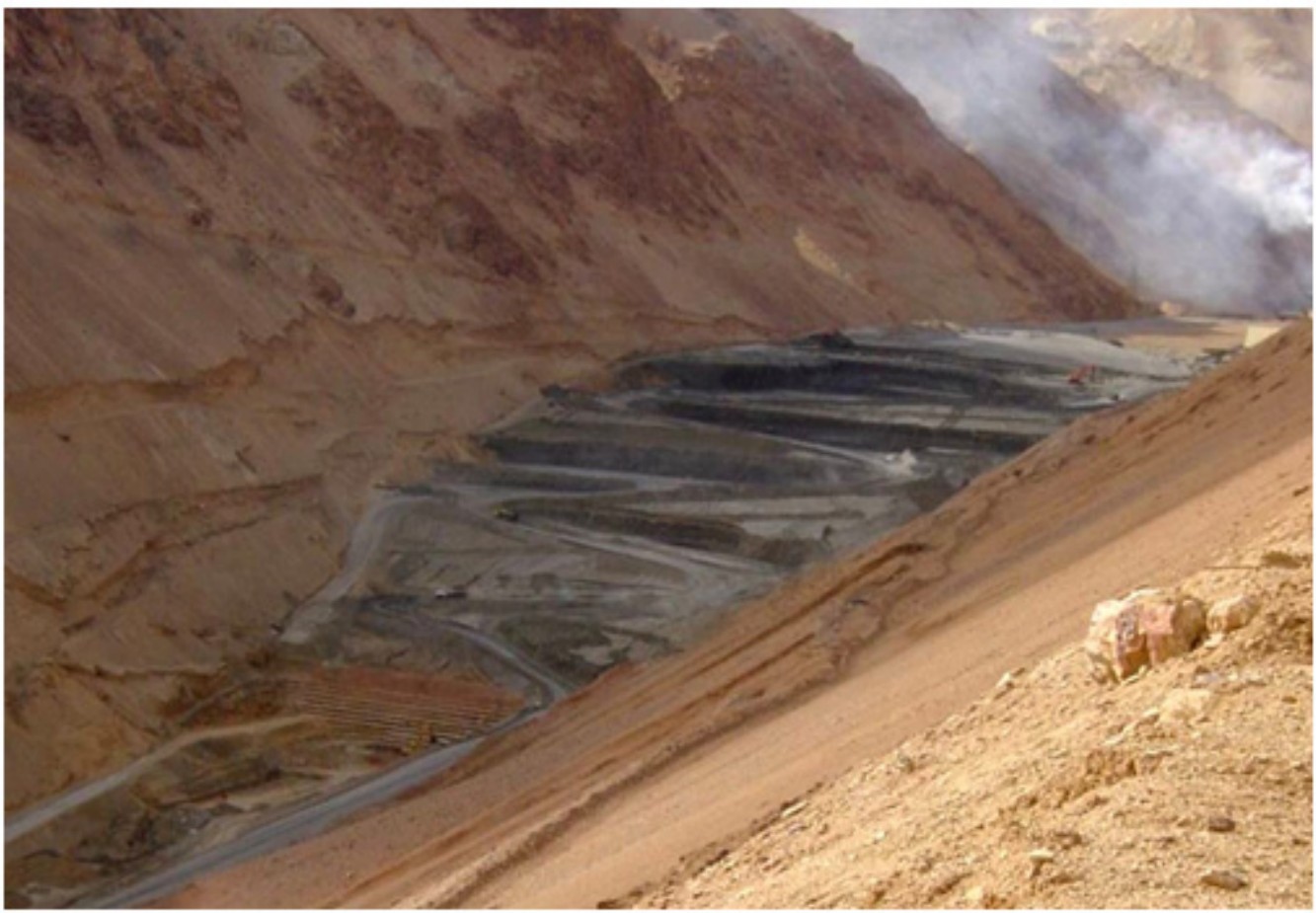

**Figure 21.** El Indio Filtered TSF Front View [53].

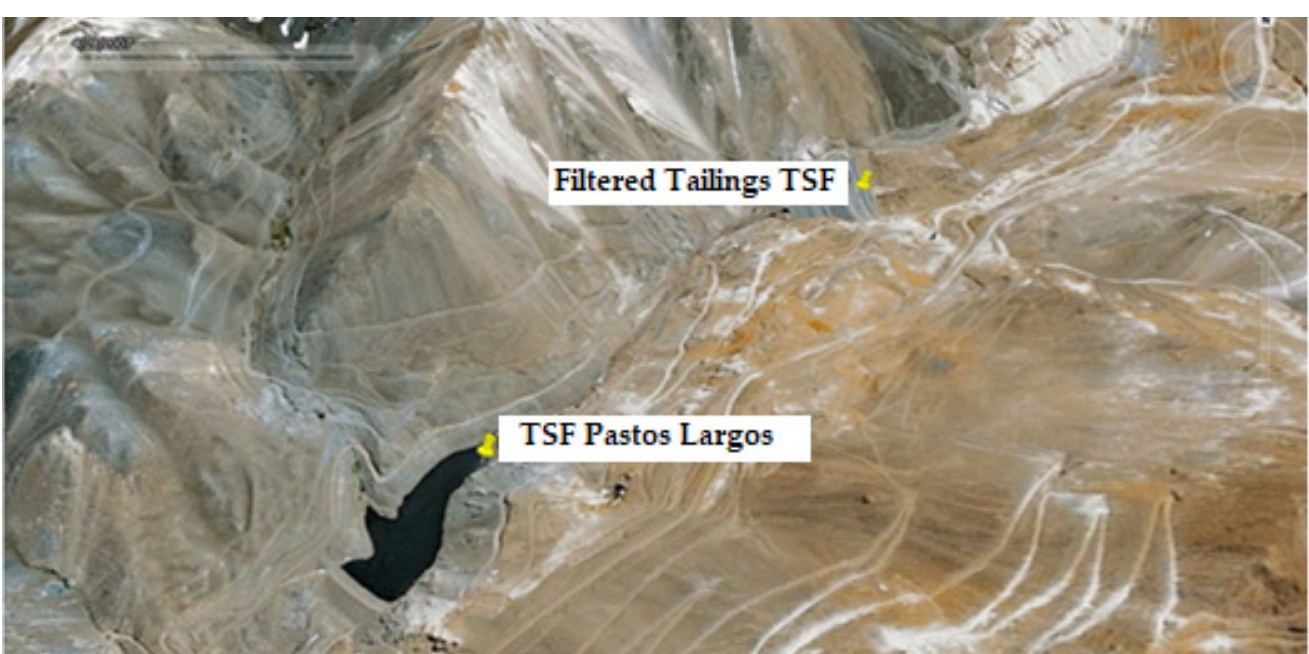

**Figure 22.** El Indio Filtered TSF Layout [53].

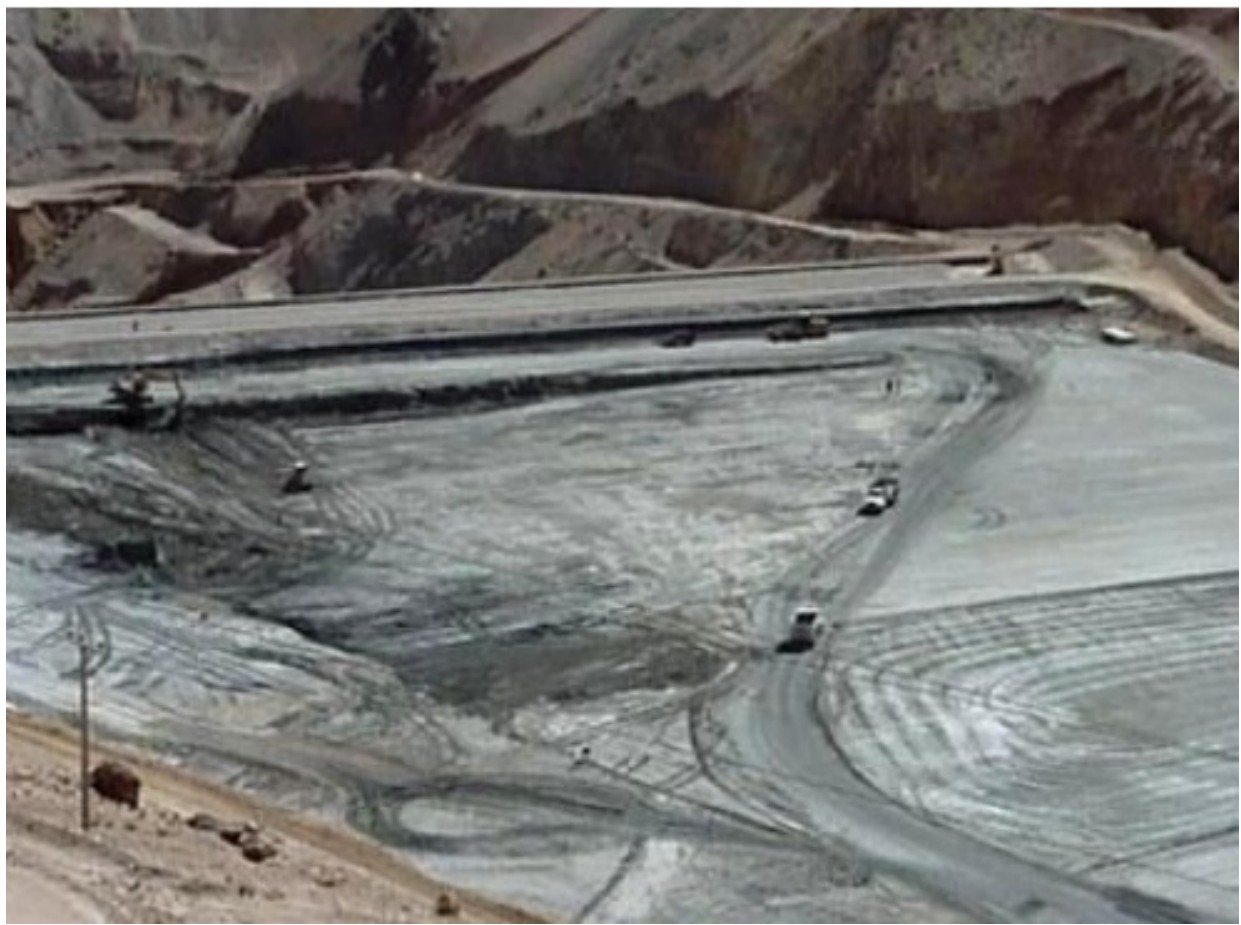

**Figure 23.** El Indio Filtered TSF Overview–Haul Trucks and Dozers works [53].

### 7.3. El Espino Filtered TSF–Flat Topography Configuration–Atacama Desert–Chile

El Espino is a copper-gold project that considers the construction of a LX-SX-EW plant for the treatment of 6.3 ktpd of leachable mineral, and a concentrator plant for the treatment of sulfide ore at a rate of 20,000 mtpd. The estimated oxide reserve is 7.2 ktpy of copper cathodes, while the sulfides production is 32,800 tonnes per year of fine copper and 25,000 oz per year of copper contained in concentrates, for a useful life of almost 18 years [51,52,60].

El Espino mine has incorporated a high dewatering technology of tailings, consisting of thickener/horizontal pressure filter plants with a 20,000 mtpd throughput, which produces filtered tailings composed predominantly of copper-gold. Tailings were processed in a filtration plant by press filters, obtaining a cake with 15% of moisture content [51,52,60].

Then the filtered tailings with a moisture content of 15% are delivered to the TSF area by a conveyor, disposed of by a MSC, spread by dozers in 0.3 m lift, and finally compacted by smooth drum vibratory compactors (Figure 24). The TSF has a flat topography configuration and will have a final footprint area of 113 Ha, with a lifetime of 18 years. The filtered tailings are stacked at the TSF in three stages, with a TSF final height of 45 m (Figure 25). To further assist the stacking of filtered tailings, the tripper car itself is equipped with a translating boom which can run 370 m along the MSC and allows for accurate placement of the ground in front or behind the MSC. The translating boom is programmed to detect the side slope of the TSF terrace via a laser probe and is able to lift (layer) the filtered tailings accurately along this terrace level. The crawlers are advanced together using limits to ensure alignment between bridges [51,52,60].

This tailings management technology was chosen due to two main aspects: (i) total cost savings over the mine lifetime (construction + operation + closure liability), and (ii) efficient water recovery. The following figure shows the MSC disposal filtered tailings at El Espino TSF.

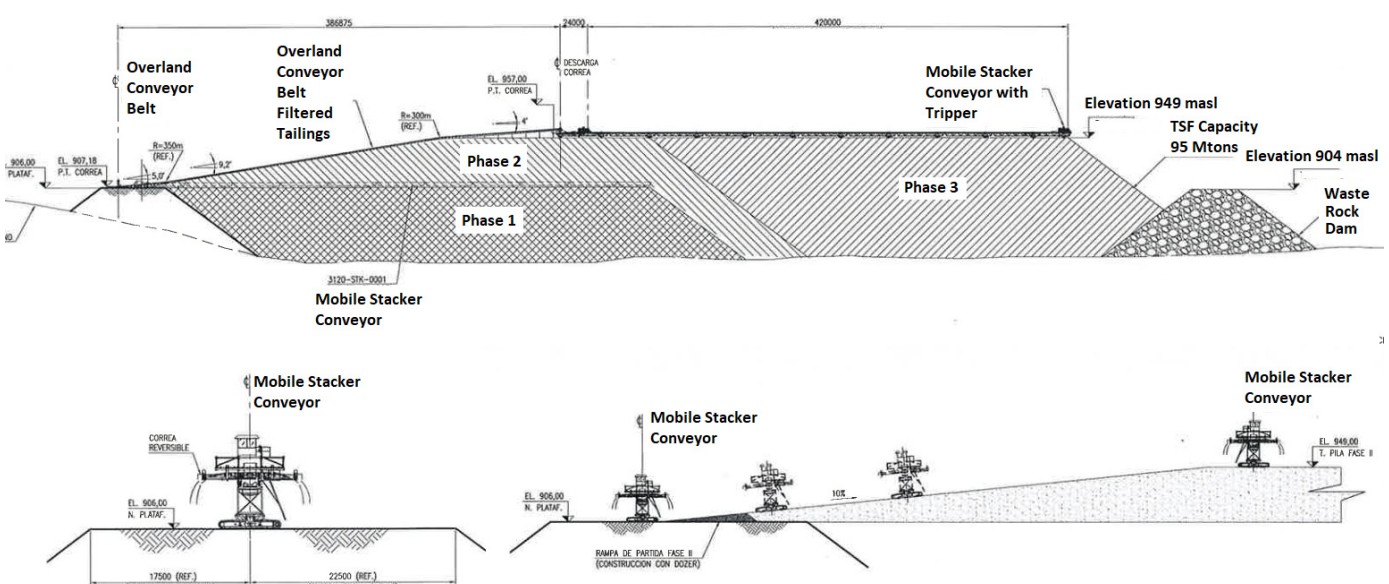

**Figure 24.** El Espino Mobile Stacker Conveyor with Tripper–Typical Cross Section [51,52,60].

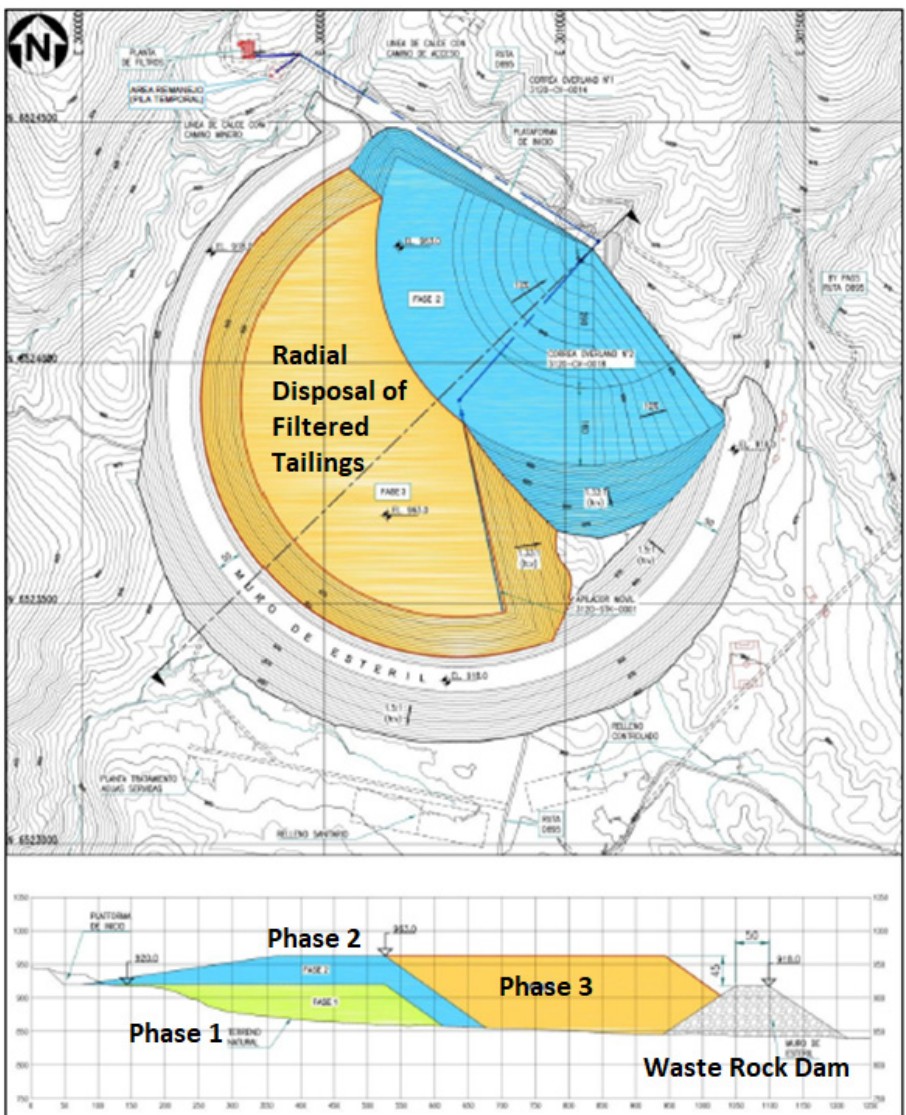

**Figure 25.** El Espino Filtered Tailings Storage–Layout and Typical Cross Section [51,52,60].

*7.4. El Gato Creek Filtered TSF–Flat Topography Configuration–Atacama Desert–Chile*

The S.C. Minera Atacama Kozan copper mine is located approximately 15km southeast of the town of Copiapó in northern Chile. The owners, a joint venture consisting of Nittetsu Mining Co. Ltd., a Japanese firm, and Inversiones Errazuriz Ltd., a Chilean company, have been mining copper at this location since 1998 [47,61].

Due to the expansion of reserves consigned in the update of available resources, prepared by an expert through drilling campaigns (period 2011–2016), the Atacama Kozan Copper Project for the production of tailings of 5500 mtpd considers the formation of a Deposit of Filtered Tailings with a total capacity to store 12 Mm$^3$ of tailings (Figure 26). The new tailings deposit will use the filtered tailings methodology, since it is environmentally more favorable and with the competitive advantage of maximizing water recovery [47,61].

The extension in the useful life of the Company will not imply a modification in the unit operations or variation in the level of treatment, both Mine and of the Plant. This increase in useful life meant developing a project to deposit more tailings since the old El Gato tailings dam ends its useful life in 2020 [47,61].

Tailings are transported and deposited at a nominal production average of 5500 mtpd. Considering that the concentrator and filtered tailings plant will have a continuous operation, with a mechanical availability of 85% and operational utilization of 90%, an annual

production equal to 1,425,000 tonnes of dry tailings is estimated. The Filtered Tailings Deposit has a total capacity to store 12 Mm$^3$ of tailings, equivalent to 24,456,736 tonnes of filtered tailings (Figure 27). Then, based on the average production of filtered tailings, the operating life of the Filtered Tailings Deposit is 17 years [47,61].

The filtering plant allows the solid-liquid separation process of the tailings to be carried out, using a vacuum ceramic disc filter system (Figure 28). In this process, a filtered cake with a 16% moisture content on a wet basis is obtained. The plant has been designed for a nominal average daily capacity of 5500 mtpd [47,61].

From the filtered tailings stockpile, located near the filtering plant in the upper southwest sector of the current tailings dam, a path is planned for the transit of trucks that will transport the filtered tailings to the deposit.

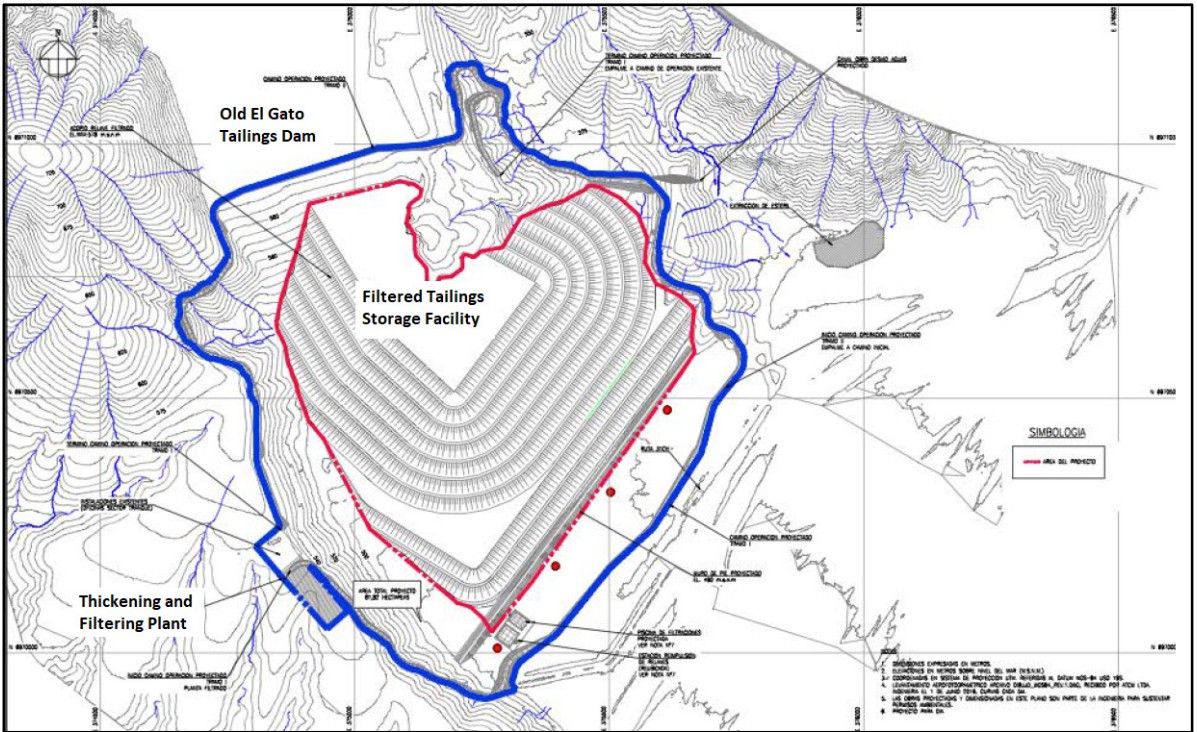

**Figure 26.** El Gato Creek Filtered Tailings Storage Facility—Layout [27,47,61].

The filtered tailings deposit system corresponds to a mechanized method, consisting of transport from the temporary storage area to the filtered tailings deposit, in 25-tonne hopper trucks, allowing the placement of the tailings in the storage areas using a bulldozer, forming layers of filtered tailings, compacted by 10-ton compacting rollers until reaching the maximum dry compacted density, equivalent to 95% of the Proctor Standard [19].

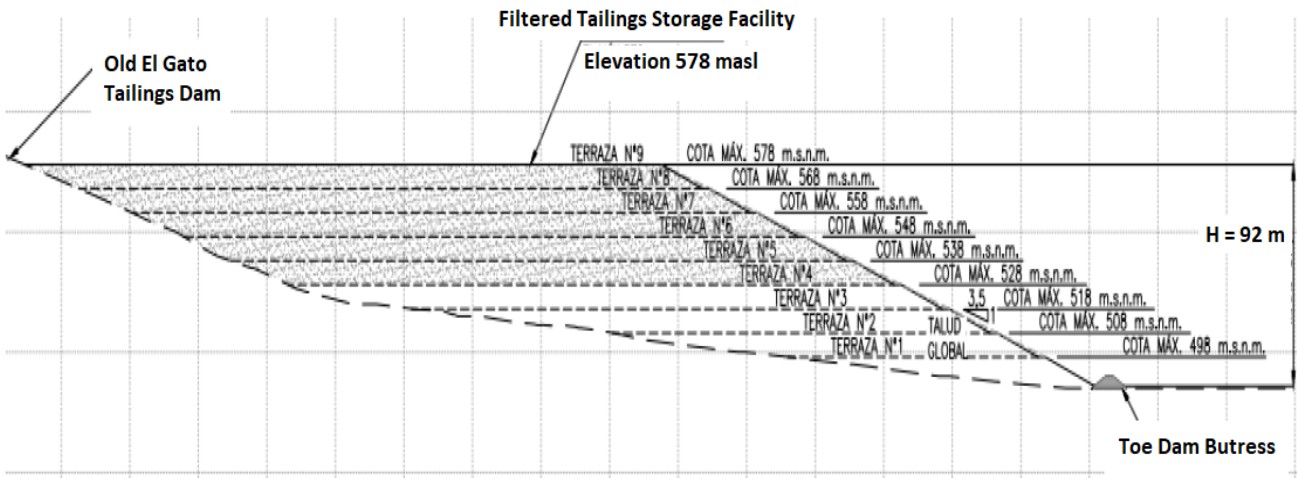

**Figure 27.** El Gato Creek Filtered Tailings Storage Facility–Typical Cross Section [27,47,61].

**Figure 28.** El Gato Creek Filtered Tailings Plant [27,47,61].

### 7.5. El Peñon Filtered TSF–Flat Topography Configuration–Atacama Desert–Chile

Yamana Gold owns 100% of the high-grade gold and silver mine of El Peñón, located in the Atacama desert, in the II Region of Chile and 160 kilometers southeast of the port city of Antofagasta and 1,800 m above sea level. The El Peñón Deposit is exploited underground,

by a Bench and Fill. Its annual movement is 2.5 million tonnes of ore and 58,000 m of tunnels are built. The water supply is made from deep wells 16 kilometers away from El Peñón [62,63].

The metallurgical process of its minerals is carried out in a plant with a capacity of 4200 mtpd, with a crushing-grinding circuit and agitated cyanidation leaching in tanks, obtaining the metal doré through a Merrill Crowe process and precipitate smelting. The tailings are filtered and deposited in a dry tailings storage facility [62,63].

The process contemplates the washing stages of the ore pulp before being deposited in the tailings deposit, carried out in the CCD circuit with a sterile solution and in the belt filters with a sterile solution. The purpose of washing the solids is to displace the contents of gold and silver towards the washing solutions. However, washing with industrial water in the belt filter (2nd washing) allows the NaCN content to be reduced from 2 g/lt to 200–300 ppm in the liquid impregnated with the tailings [62,63].

The process of depositing tailings in the filtered tailings storage facility (Figures 29–31) begins with the discharge of the filtered material, corresponding to a cake between 18 and 20% humidity, which is sent to a temporary tailings storage facility (in a material transport belt 55CVR01) that has a weight gauge (55WIB01), a wet tailings sampler (55AM01), and a chute (55CH04) for unloading and distribution to the stockpile site. The base of the tailings stockpile is made up of a concrete slab and a retaining wall to protect the belt, with a capacity of 2500 tonnes of tailings and has a floor pump (55PPS04) to collect solutions and cleaning water that are recycled to the Filtering Surge Tank (55TNK01) [62,63].

The stacked tailings are loaded onto trucks (50 tonnes) with a front loader (CAT 966) and transported to the tailings deposit (Figure 32). Once the tailing has been deposited, it is spread in layers of 15 cm, by means of a motor grader [62,63].

To complete the process, the tailings are removed by means of a disc plow and irrigated with industrial water three times a day to achieve the degradation of the cyanide contained in the solution that accompanies the solid (18–20% humidity) (Figure 33). This aeration and irrigation process is repeated for 72 hours, a process with which a tailing with cyanide concentrations of less than 2 ppm is obtained [62,63].

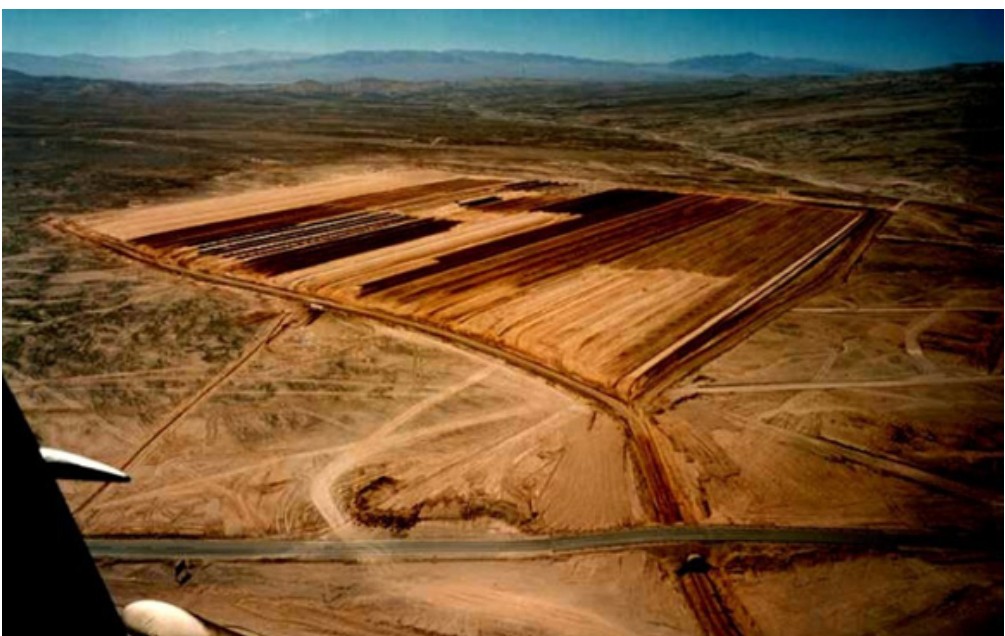

**Figure 29.** El Peñon Filtered Tailings Storage Facility Overview [62,63].

After 72 hours of treatment, having reached a residual cyanide concentration of 2 ppm and once the tailings are dry, a second 15-cm layer of material is deposited and so on until

four layers (50~60 cm) are completed. The tailings are compacted by means of a 10-tonne roller [62,63].

The management of the tailings deposit follows a cyanide degradation, and tailings compaction program, to achieve a mechanically stable deposit area, free of contamination and with access roads that allow the circulation of vehicles involved in the operation [62,63].

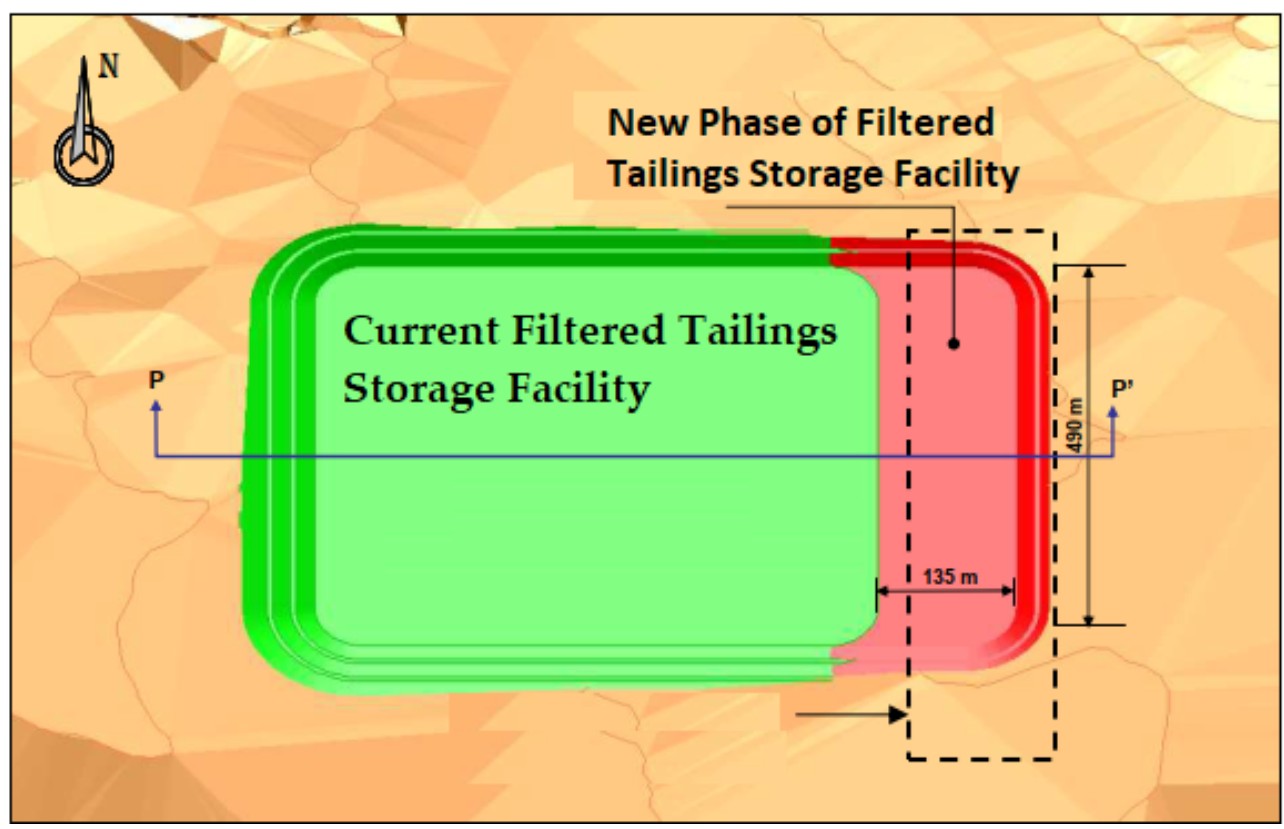

**Figure 30.** El Peñon Filtered Tailings Storage Facility—Old and New Phase Layout [62,63].

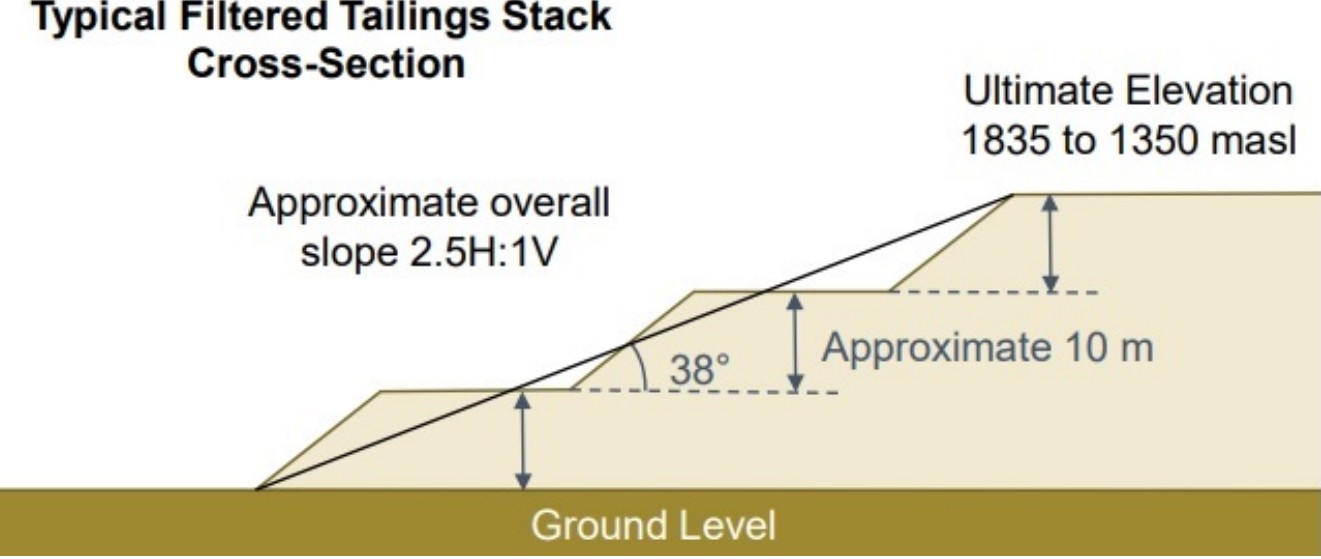

**Figure 31.** El Peñon Filtered Tailings Storage Facility–Old and New Phase Cross Section [62,63].

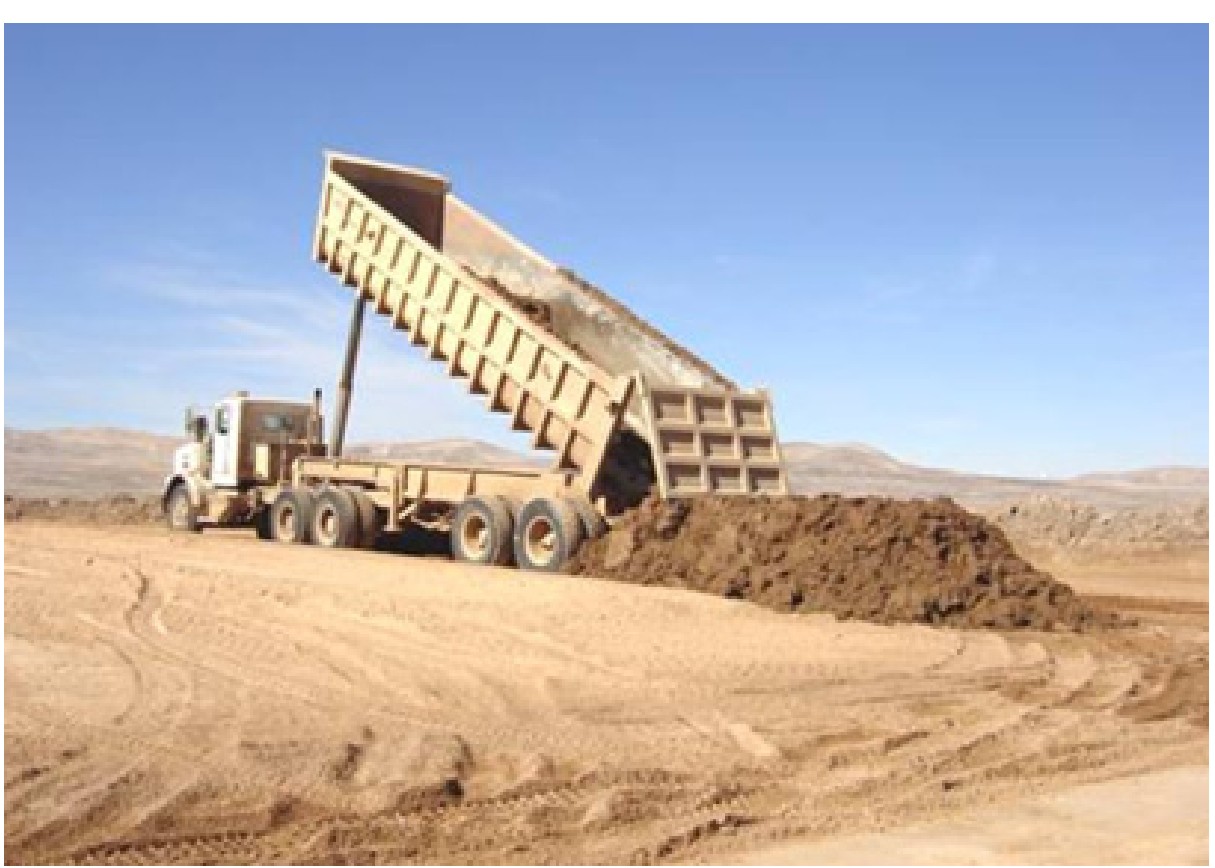

**Figure 32.** El Peñon Filtered Tailings Storage Facility—Haul Truck with Tails Disposal [62,63].

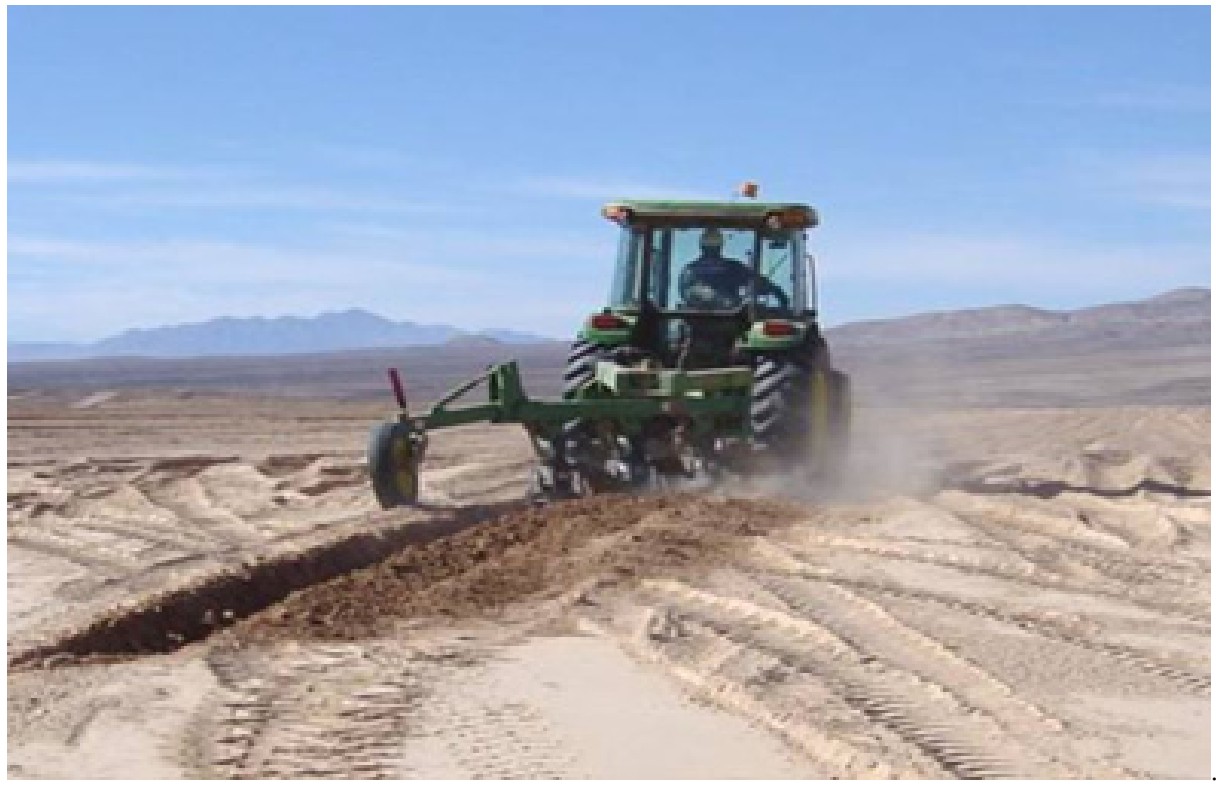

**Figure 33.** El Peñon Filtered Tailings Storage Facility–Tailings are removed by a disc plow [62,63].

### 7.6. Huasco Filtered TSF–Flat Topography Configuration–Atacama Desert–Chile

Valle de Huasco is a mining complex owned by the Chilean steel producer Compañía de Acero del Pacífico (CAP) with operations close to the city of Vallenar, approximately 650 kilometers north of Santiago. The area is comprised of the operating Los Colorados mine and the now-closed El Algarrobo mine. The Huasco Pellet Plant is a mining facility that dates back to 1978, whose objective is to process and add value to the product of iron mining, in the Huasco Commune. Its process consists of receiving pre-concentrated iron minerals from mines located in the Chilean iron strip and submitting them to grinding and concentration through physical processes (gravimetric and magnetic) to generate a commercial product called iron concentrate or pellet feed, with a concentration of magnetite ($Fe_3O_4$) over 90%. Alternatively, this product can be agglomerated and subjected to a thermal process to obtain iron pellets [54,55].

The total tailings from the grinding process are made up of the overflow of the hydro separators and the rejection of the magnetic batteries of each Grinding and Concentration line. Tailings generated are transported to the two Tailings Thickeners, where a flocculant product is added in order to increase the sedimentation rate of solids and optimize water recovery. From this point, the project under study contemplates the installation of a tailings filtering plant and a conveyor belt to connect this plant with a new filtered tailings deposit located 1.2 km south of the Pellet Plant (Figure 34) [54,55].

The start of tailings management corresponds to the discharge from the existing regulating agitator ponds, which will feed the filtering plant with tailings. From these ponds, the tailings are pumped to the filtering plant where the filters with the capacity to process a nominal amount of 5000 mtpd of tailings are fed. The filter plant will have up to four press-type filters available for filtering [54,55].

The Filter Plant has four vertical plate press filters available to filter the tailings until reaching the desired humidity. The process generates filtered tailings with up to 20% humidity, which are disposed of in the new deposit. For its part, the filtered water is returned to the Pellet Plant process [54,55].

At the Filter Plant, through collection belts, the filtered tailings are conveyed to a hopper located in the tailings transfer building, which will transfer it to an overland belt that will convey it to a second transfer station next to the Tailings Deposit. At the end of the belt, a transfer station is located, where there is a closed unloading building, the purpose of which is to allow tailings to be loaded onto trucks with a high load capacity, thus reducing transportation cycles [54,55].

The transfer building will have an alternative discharge system in case of failure of the overland belt. In case of emergency, trucks are located under the alternative exit and will transport the filtered tailings through the existing road of the Pellets Plant to an emergency stockpile located 200 m from the Pellets Filters Plant (Figure 35) [54,55].

At the work front of the filtered tailings deposit, the trucks will unload the tailings, which are spread out in layers using heavy machinery. These layers are compacted. Once the tailings layers have been compacted, they are irrigated with a solution of dust suppressant with water. In turn, the entire path traveled by the trucks, inside and outside the tailings storage facility, is stabilized with a dust suppressant to minimize emissions of particulate dust [54,55].

The filtered tailings deposit is delimited at its base by a toe dam of variable height, between 5 and 10 m high, to reach a height of 40 m, with respect to sea level (Figure 36). The dam is built with borrow material, and its objective is to contain the starting slope of the deposit, ensuring its physical stability and providing support to the first bank. The reservoir is filled in five successive growth stages (terraces) [54,55].

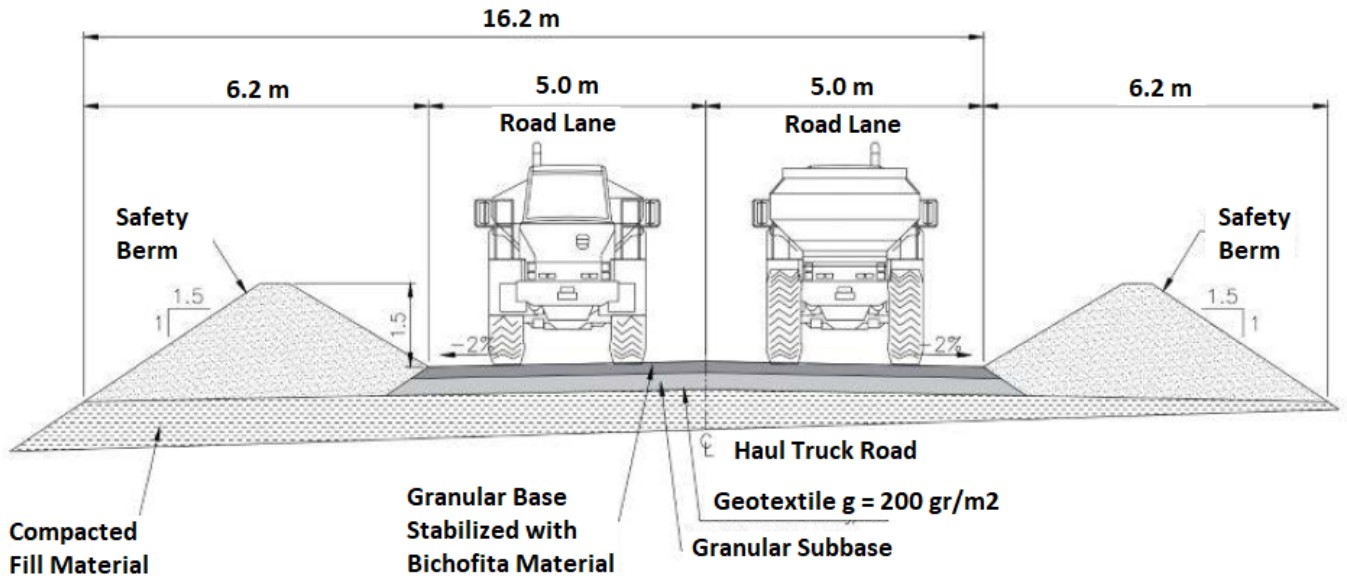

**Figure 34.** Huasco Filtered Tailings Storage Facility—Layout [54,55].

**Figure 35.** Huasco Haul Truck Road at Filtered Tailings Storage Facility—Typical Cross Section [54,55].

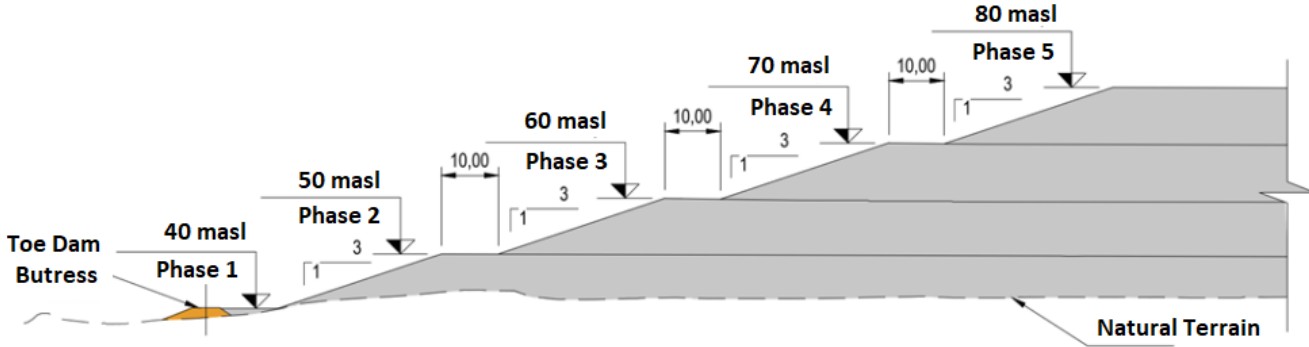

**Figure 36.** Huasco Filtered Tailings Storage Facility—Typical Cross Section [54,55].

*7.7. Salares Norte Filtered TSF–Flat Topography Configuration–Atacama Desert–Chile*

Salares Norte is located between 3900 and 4700 m above sea level, in the middle of the highest peaks of the Andes mountain range and the driest desert in the world. Its height and geographical environment are some of the challenges faced by this project that seeks to process 2 million tonnes of ore per year for the production of metal doré [50,64,65].

The exploitation of gold and silver is carried out in the open pit. The ore is processed by a crushing and grinding system. The extraction is through a hybrid scheme of conventional cyanide leaching -Merrill Crowe- and carbon in pulp for the recovery of gold and silver and deposit of filtered tailings [50,64,65].

The system called wad cyanide detoxification system will occupy an approximate area of 190 m$^2$. Its objective is the detoxification of tailings (reducing the concentration of wad cyanide to less than 15 ppm, before being stockpiled) and the conditioning of moisture in the tailings to its final disposal in the filtered tailings deposit. Sodium metabisulfite is used to detoxify the wad cyanide in the tailings from the coal-in-pulp stage [50,64,65].

The plant, regarding the treatment of cyanide, has been designed taking into account the requirements of the international cyanide code, a non-binding regulation in Chile, which contemplates the highest world standard in the handling of this substance [50,64,65].

Following the detoxification of the tailings, the pulp is taken to a tailings thickener, this thickener is 30 m in diameter and will occupy an area of 1100 m$^2$ inside the plant, where the envelope flow or overflow is sent to the plant for incorporation into the process, while the pulp is conducted through pumps to the tailings filtering area, located approximately 1.2 km from the gold process plant (Figure 37) [50,64,65].

The tailings filtering sector will have an approximate area of 1000 m$^2$, which includes: a thickening sector that includes a flocculant plant and recovered water management ponds; and a filtering area with auxiliary services and a system for returning the water recovered to the process [50,64,65].

The equipment considered in the tailings filtering stage is: (i) A 30 m diameter tailings thickener; (ii) A filter tank of 350 m$^3$ (process water tank); (iii) Three vertical plate type press filters (two in operation and one on standby), and (iv) Two tanks of 1500 m$^3$ for feeding the filter [50,64,65].

The tailings deposit that is on the intermediate platform of the southern waste rock dump, consists of a tailings storage facility previously filtered, self-supporting stockpile of tailings, which are built in layers compacted by means of a vibrating roller, with humidity around the optimum according to the results of Standard Proctor compaction tests, in order to reach the compaction levels established in the Technical Specifications of the Project, which corresponds to a range between 85% and 90% of the Standard Proctor test, which are carried out periodically during the operation of the deposit (Figure 38) [50,64,65].

The filtered tailings deposit is filled at a tailings production rate equivalent to 5500 mtpd, it will also be made up of two platforms with a bank height of 20 m, occupying an area of 54 ha and a volume of 16 million m$^3$ (Figure 39) [50,64,65].

Both the base of the tailings deposit (intermediate platform of the southern waste rock dump) and the slopes of the hills and inclined surfaces of the dump on which the tailings will rest are waterproofed by means of a geomembrane [50,64,65].

The filtered tailings deposit has a system for capturing contacted water consisting of a basal drainage system, collection of water at the foot of the slope, a collection pool downstream of it, and a storage pool located on the south side of the filter plant. It also has a safety parapet downstream to contain potential spills [50,64,65].

Additionally, as the tailings deposit grows, fill material or mine waste is placed on its slope to reduce wind erosion on the said surface [50,64,65].

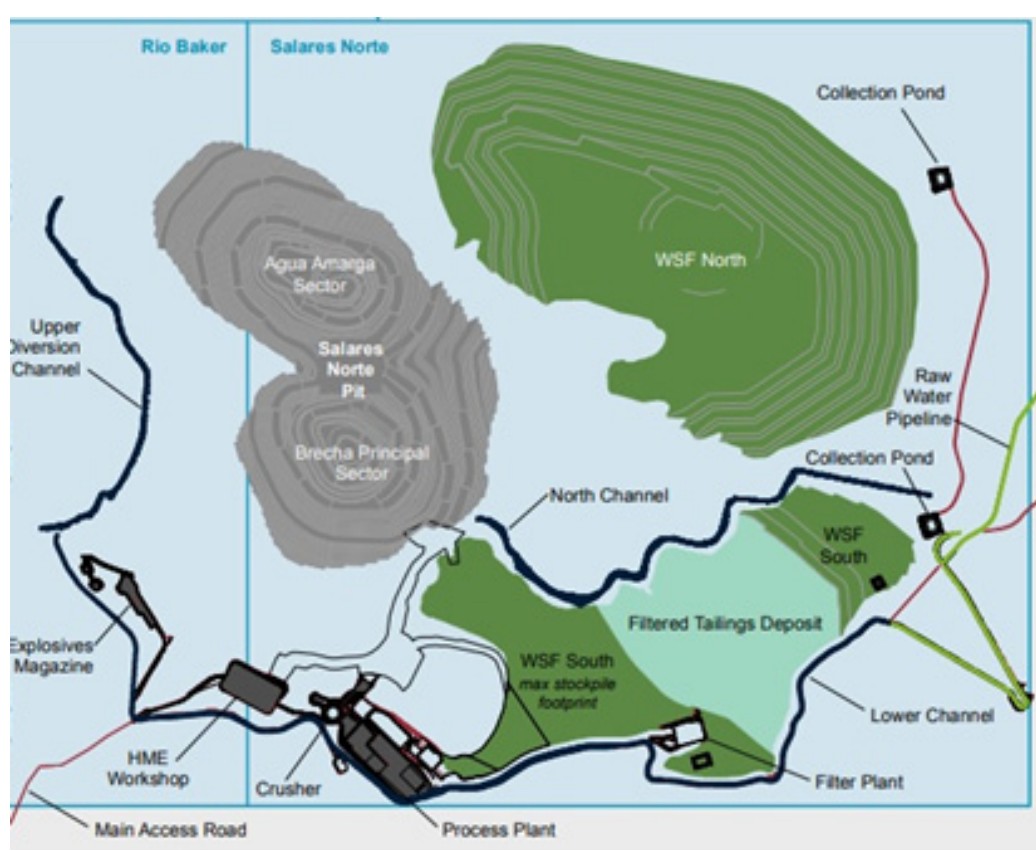

**Figure 37.** Salares Norte Filtered Tailings Storage Facility—Layout [50,64,65].

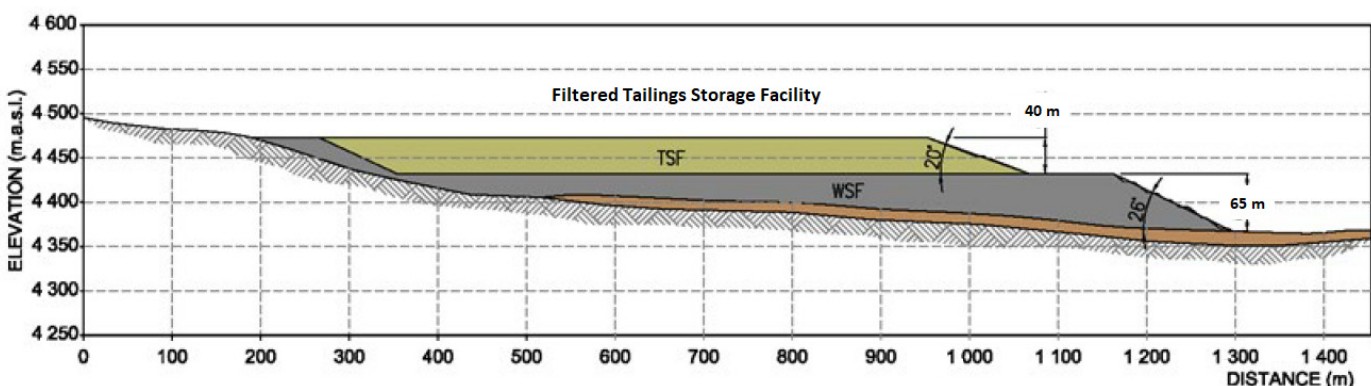

**Figure 38.** Salares Norte Filtered Tailings Storage Facility–Typical Cross Section [50,64,65].

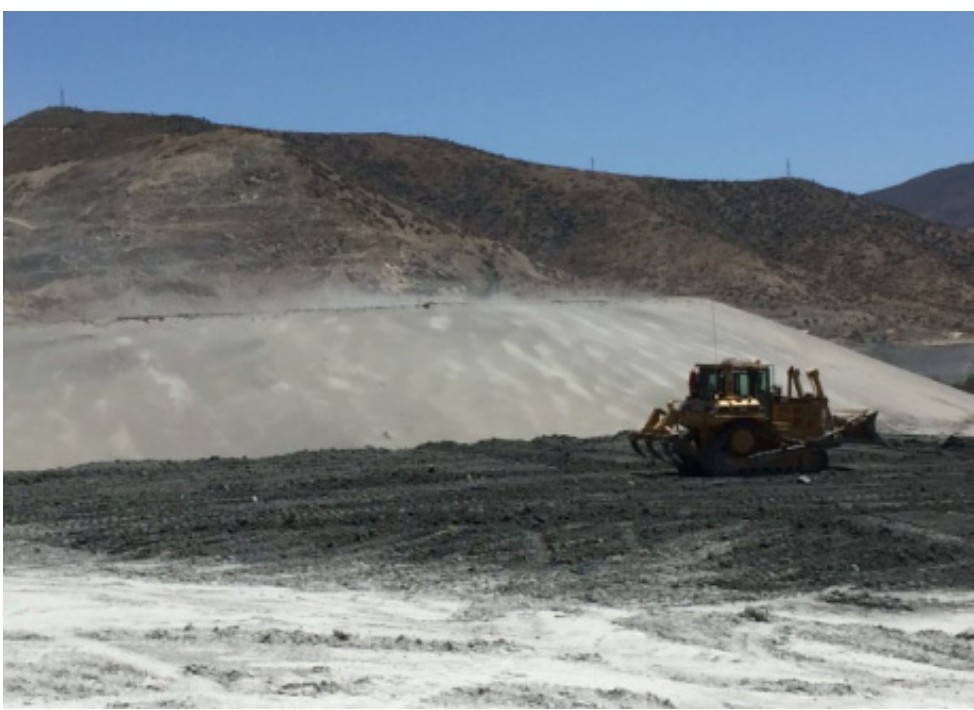

**Figure 39.** Salares Norte Filtered Tailings Storage Facility–Dozer forming filtered tailings layers [50,64,65].

### 7.8. Tambo de Oro Filtered TSF–Valley Topography Configuration–Atacama Desert–Chile

The Tambo de Oro Mining Site is located in the Punitaqui Mining District, 5 km southeast of the town of Punitaqui and 29 km southeast of the city of Ovalle, commune of Punitaqui, Province of Limarí, Region of Coquimbo. The Tambo de Oro Mining Site corresponds to underground exploitation of high-grade gold ore, environmentally authorized for extraction of 15,000 t/month -using the Bench and Fill (B&F) extraction method- and processing, via flotation, of 22,500 t/month, whose final product is gold-copper concentrate and thick gold concentrate obtained by gravitational methods. In this regard, it should be borne in mind that the processing capacity considers the purchase of 7500 t/month of ore from third parties, which, added to the extraction of 15,000 t/month, completes the processing capacity processing of 750 mtpd [49].

Tailings are the fraction of the flotation that is mainly made up of gangue and other species that have no commercial interest because they have low grades of gold, silver, and copper. In this area, the installation of a thickener and a filter system has been considered.

The tailings from the gold flotation are thickened from an initial concentration of 37% by weight to a final concentration of a minimum of 55% and a maximum of 60%. The thickening of the tailings to 55% solids will allow the recovery of around 9.4 m$^3$/h of water that returns to the process and the solids are used for tailings filtration. The tailings thickening circuit is made up of a thickener and an accumulator/conditioning tank that collects the underflow from the thickener and is also the feed point for the tailings filtrate [49].

The process tailings from the underflow of the tailings thickener are accumulated and/or conditioned in a tank equipped with an agitator, from where an automatic filter press with 190 plates of 1200 mm × 1200 mm is fed (Figure 40). With the capacity to process 4.41 m$^3$ per cycle (11 minutes per cycle), this guarantees production with 20 h of operation per day. The product of the tailings filtration will generate flotation tailings with average humidity between 12 and 16%. The filtered material is transported in trucks to the tailings deposit [49].

For the transport of tailings, trucks with a capacity of 14 m$^3$ are used, which will move from the processing plant to the tailings deposit, approximately 500 m east of the

processing plant, with a frequency of 40 trips/day in the highest processing stage of the plant (Figure 41) [49].

The filtered tailings deposit will have a capacity of approximately 864,000 tonnes, compacted at a density of 1.85 tonnes / m$^3$ (Figure 42). The deposit will have a total height of 60 m and covers an area of 4.7 ha. The base of the deposit is prepared in order to ensure a good anchoring of the tailings material to the ground. The tailings disposal slope is H:V = 2.5:1.0, with banks every 15 m and berms 3 m wide (Figure 43). The material is spread out in horizontal layers that do not exceed 30 cm in loose thickness and is compacted until it reaches a minimum dry density of 95% of the maximum compacted density dry from the Standard Proctor test [19,49].

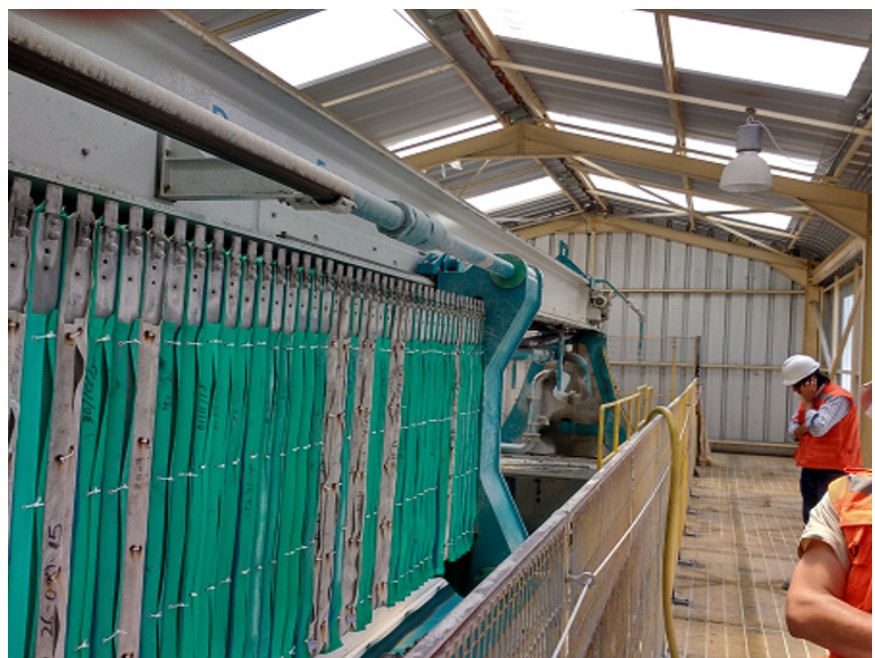

**Figure 40.** Tambo de Oro Tailings Vertical Plate Filter Press [49].

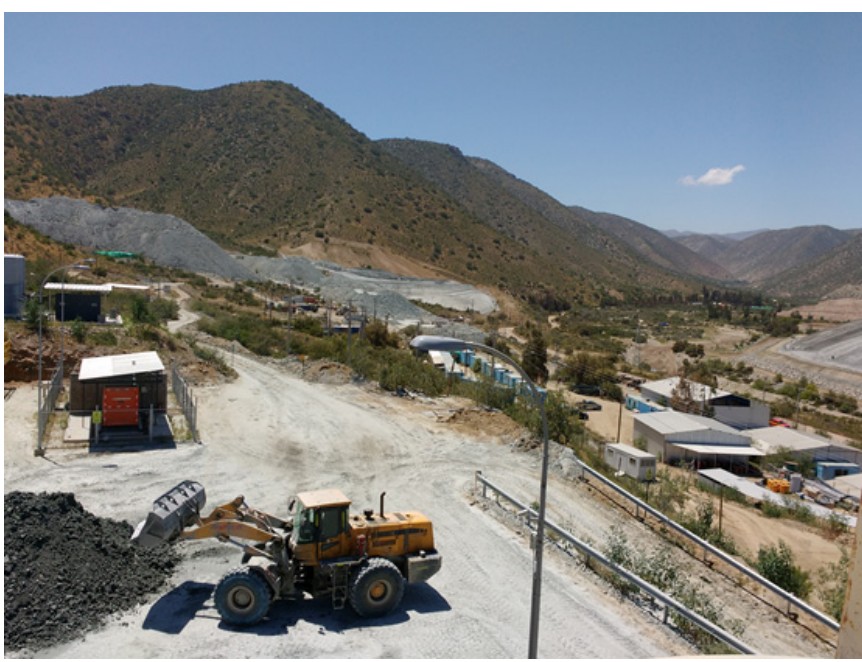

**Figure 41.** Tambo de Oro Filtered Tailings Storage Facility Overview and Front Loader Works [49].

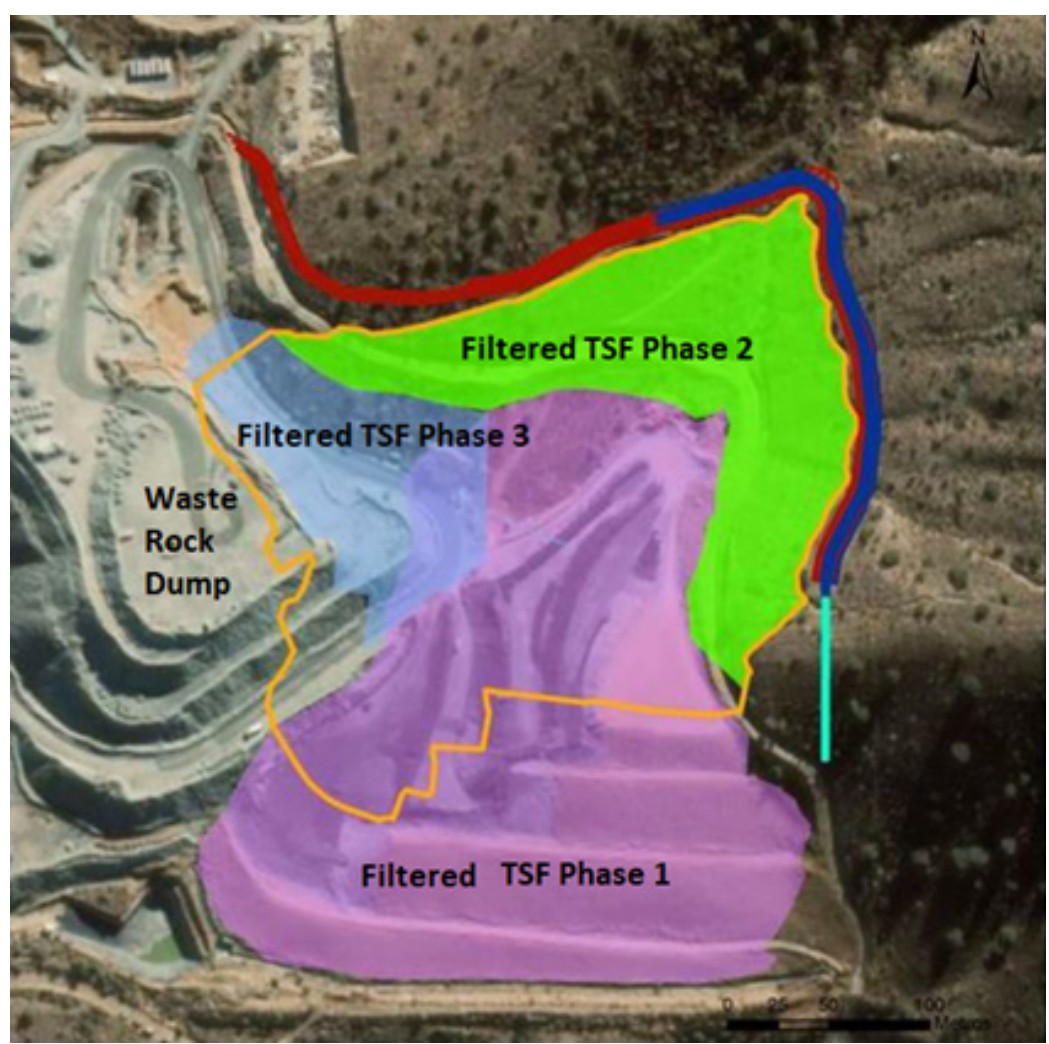

**Figure 42.** Tambo de Oro Filtered Tailings Storage Facility—Layout [49].

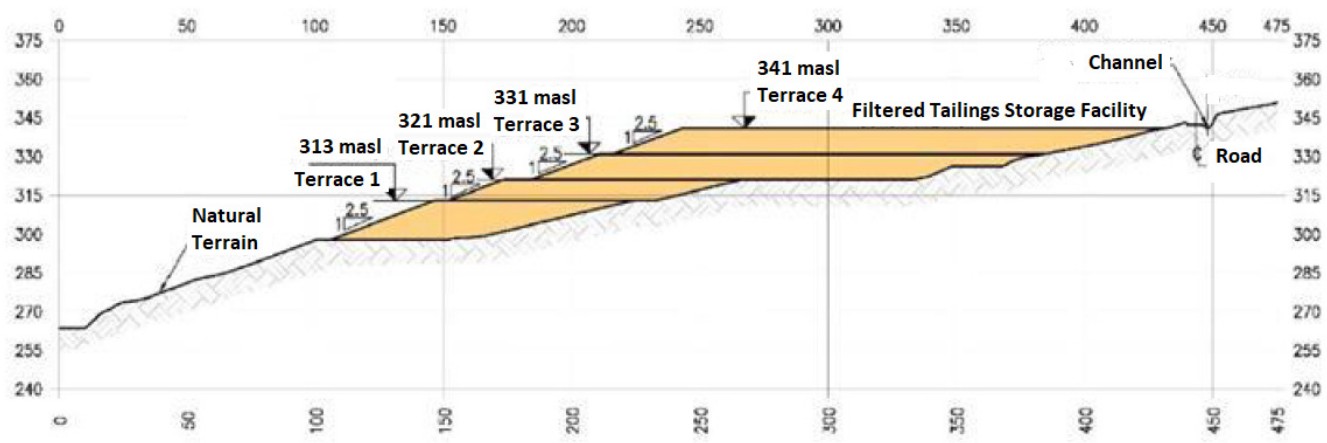

**Figure 43.** Tambo de Oro Filtered Tailings Storage Facility–Typical Cross Section [49].

### 7.9. Tambillos Filtered TSF–Flat Topography Configuration–Atacama Desert–Chile

The Tambillos Mining Site is located in the Coquimbo Mining District, 28 km south of the city of La Serena, Region of Coquimbo, Chile. From Mina Florida, ore is supplied to the Beneficiation Plant for the production of copper concentrate, located four km to the Southeast (SE) of the town of Tambillos [48,66].

In this regard, it should be noted that the useful life of the Project increases by seven years the useful life of the mining operations carried out by Florida Mining Company at its Tambillos Site. The foregoing is a consideration of the existence of sufficient copper ore reserves in Mina Florida to continue processing ore at 3000 mtpd [48,66].

The tailings are transported from the concentrator plant. It will originate in the tailings of the flotation plant, which are driven using the existing caisson and pumps (which currently drive towards the tailings impoundment in operation). The pipe is HDPE 250 PN6 PE100 and will continue to the thickener feed box and projected filtrate.

The tailings thickening unit will consist of a high-efficiency thickener (HRT) with a diameter of 18 m, which has a feed elevator box and will serve as a compact flocculant preparation and conditioning unit (Flocculant C1590). The pulp discharged from the thickener is pumped to the upper loading drawers of each of the filters (three in total), through HDPE 160 PN6 PE100 pipes with a nominal diameter of 160 mm with 50 HP centrifugal pumps (1 in operation and 1 on standby) [48,66].

The filtering unit is enabled on a platform of approximately 2000 m$^2$, where the plant is mounted, as well as the tailings control and loading systems. This unit comprises three ceramic disc filters installed in series, with a filtering area of 120 m$^2$ each (Figure 44). The tailings that have passed the filtering process are unloaded using a front loader, which in turn will load 40-tonne trucks for transport to the deposit area, where they are distributed using bulldozers and compacted using a pneumatic roller (Figure 45) [48,66].

It is important to point out that the location of the filtering unit is optimal from the operational and environmental point of view since it allows for optimizing the travel times of loaded trucks from the filters to the tailings disposal area (distance of 600 m), reduces fuel costs and the operational risk associated with the displacement of a large volume of tailings [48,66].

The civil work to be projected for the management of mining waste consists of a tailings deposit made up of terraces 7 m high each, arranged in such a way as to allow the existence of 5 m berms and a general slope of H:V = 3.0:1.0 (Figures 46 and 47) [48,66]. In specific terms, the filtered tailings facility will have the following geotechnical characteristics:

- Average geotechnical humidity of the tailings cake: 16%.
- Construction method: terraces.
- Platform height: 7 m.
- Berm width: 5 m.
- Local terrace slope: H:V = 2.5:1.0.
- Global Slope: H:V = 3.0:1.0.
- Maximum dry compacted density (95% Standard Proctor): 2.05 t/m$^3$.

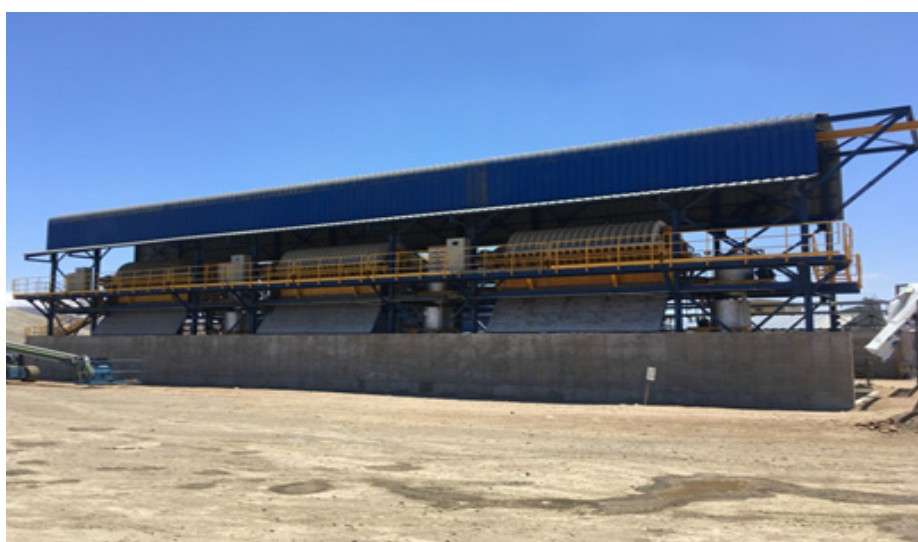

**Figure 44.** Tambillos Tailings Ceramic Disc Filters [48,66].

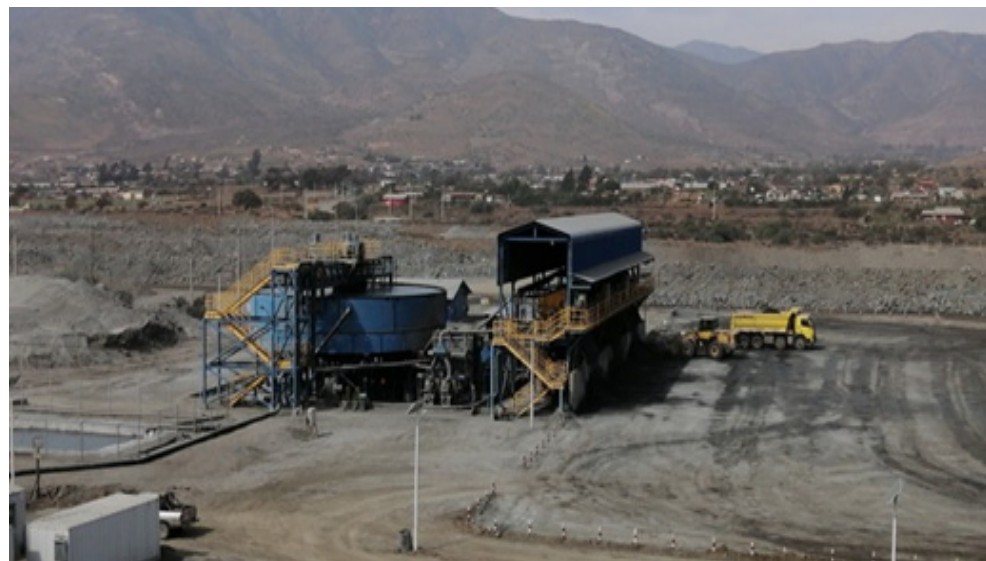

**Figure 45.** Tambillos Tailings Thickening and Filtering Plant [48,66].

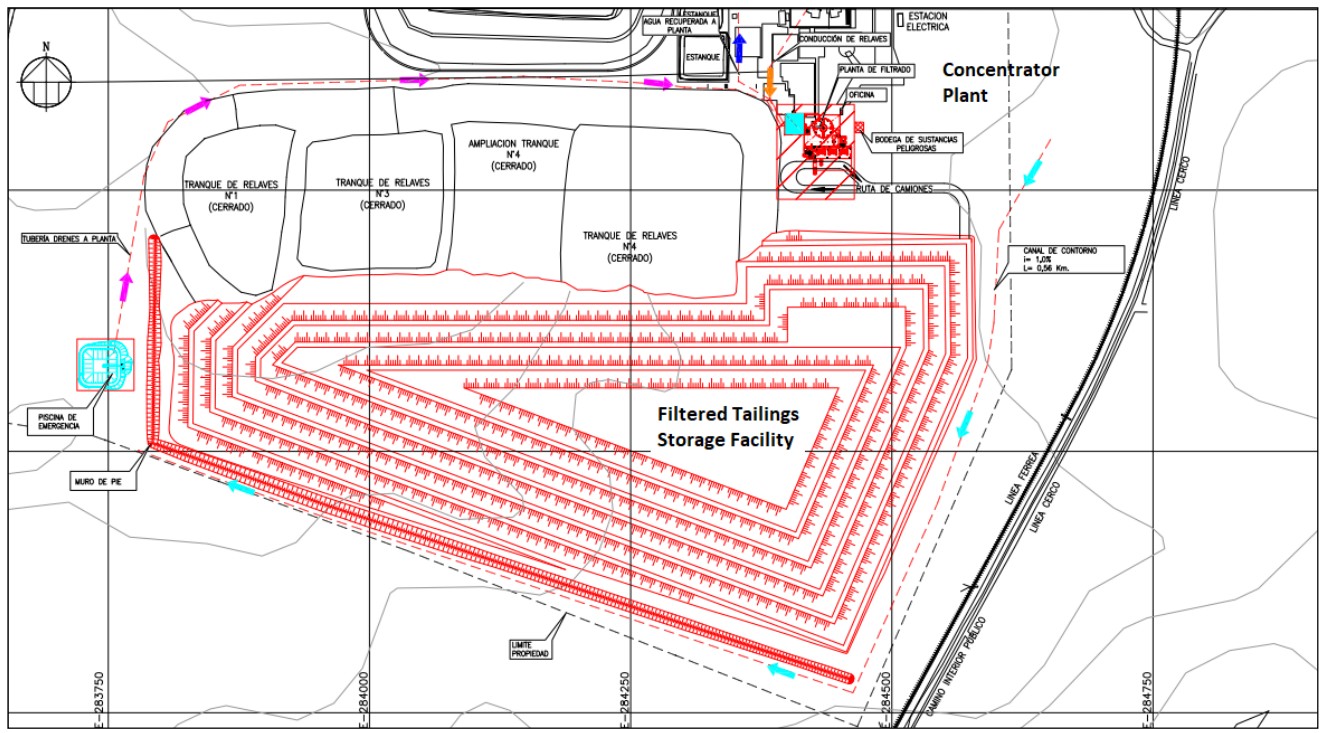

**Figure 46.** Tambillos Filtered Tailings Storage Facility Layout [48,66].

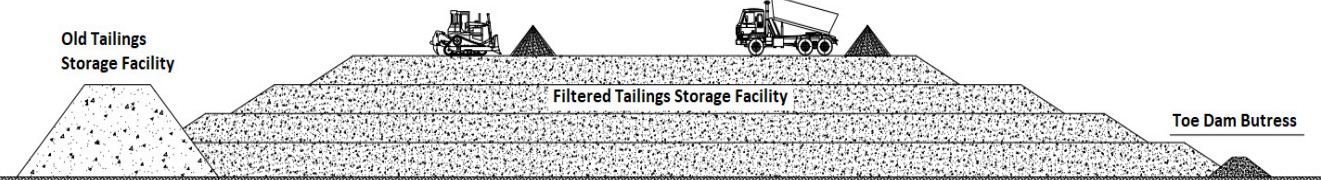

**Figure 47.** Tambillos Filtered Tailings Storage Facility Typical Cross Section [48,66].

### 7.10. Mantos Blancos Filtered TSF–Flat Topography Configuration–Atacama Desert–Chile

The Mantos Blancos copper mine is located 48 km from the Chilean port city of Antofagasta and is 800 m above sea level. Anglo American is the owner of the mine; which is one of the company's six operating divisions in the region. The Mantos Blancos division includes an open-pit mine, crushing plants, and installations for processing oxide and sulfide ore. The mine mainly produces copper. The Sulfide ore plant at Mantos Blancos Division of Anglo American Chile Ltd, which started its operations in 1981, has had a tailings treatment plant from the beginning. Basically, it consists of a classification circuit, slimes thickening, and filtration of the coarse fraction. As this plant is located in one of the most arid lands in the world, coarse tailings are filtered in order to maximize water recovery. Disc filters were used at the beginning. Later in 1984, during the expansion of the concentrate plant to 12,000 mtpd, horizontal belt filters were incorporated and are still being used today. Tailings are finally disposed of in a dam for thickened slimes and in a dam for the coarse fraction, in a 40/60% ratio, respectively, and a solids concentration of 60% and 80%, respectively [18,67,68]

The tailings from the concentrator plant are classified in a station of 500 mm Hydrocyclones, functioning by gravity, in a conical configuration, with four operating units and two on stand-by. This classification system produces a solids distribution of 50–55% and a cut size of d50 between 50–60 microns. The overflow of the cyclones, CW 24–26% of solids, is transported to three thickening units (Larox of 60 m diameter, Dorr Oliver of 44 m diameter, and Eimco of 44 m diameter) for the process of sedimentation, using a dosage of flocculent of 5 gr/ton, that later feeds with a solids content of Cw 55–60% the filtration units and the fine tailings dam in a ratio restricted by the capacity of absorption of fines in the filters. The thickeners recover approximately 63 to 67% of water, which is returned to the concentrator plant (Figure 48) [18,67,68].

The underflow of the cyclones, with a solids content of Cw 65–67%, is sent to three filtration units (band filters of 100 m$^2$ each) and are previously mixed with parts of thickened fine tailings. Tailings cakes are obtained with a 17% moisture content considering a wet basis (equivalent to a 23% moisture content considering a dry basis), which are later transported to the dry stockpile of coarse material by conveyor belts (Figure 48). The filtered water, plus the wash water of the filtering cloth, is recirculated to the thickeners to sediment solids in suspension [18,67,68].

The Filtered Coarse Tailings Storage Facility is developed by down valley placement method applied at Mobile Stacker Conveyor with tripper forming a fan shape, where saturated filtered tailings flow along roughly repose angle slopes, buttressed at the toe by a containment dyke and downstream supported by a sedimentation collection pond. Dozers carried out spreading and compaction activities on filtered tailings lifts of average 20–30 cm in thickness (Figures 49 and 50) [18,67,68].

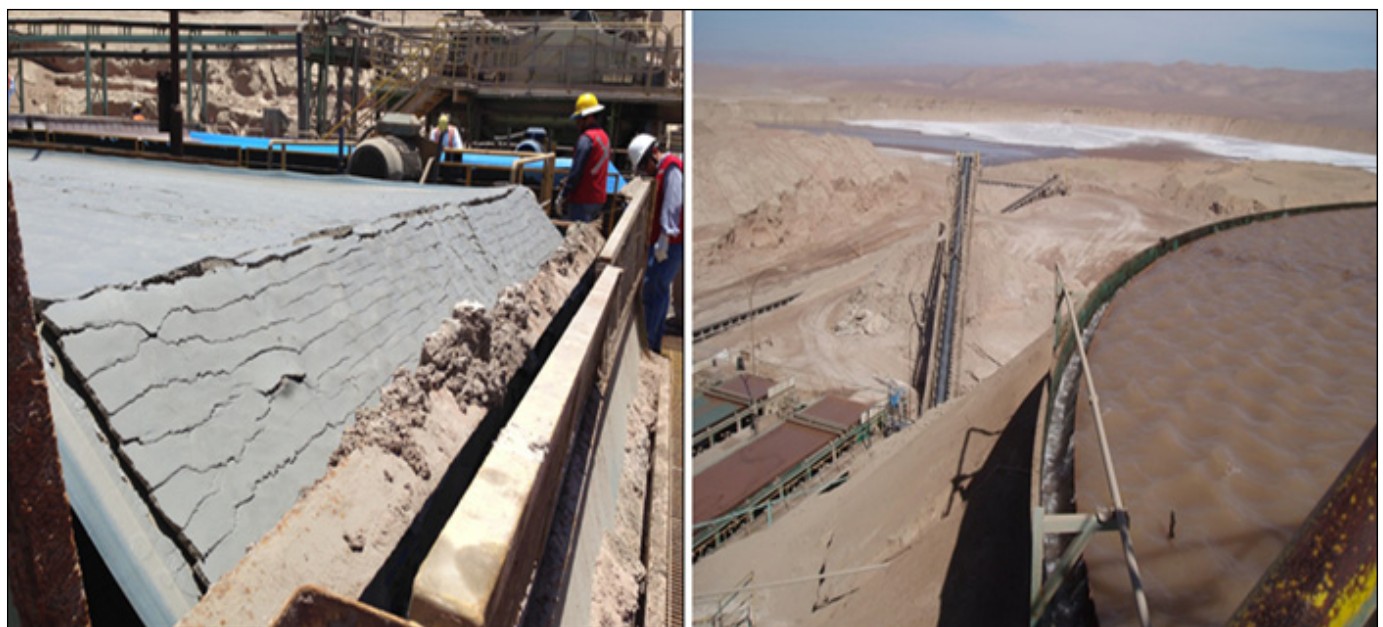

**Figure 48.** Mantos Blancos Filtered Tailings–Belt Filters and TSF overview [18,67,68].

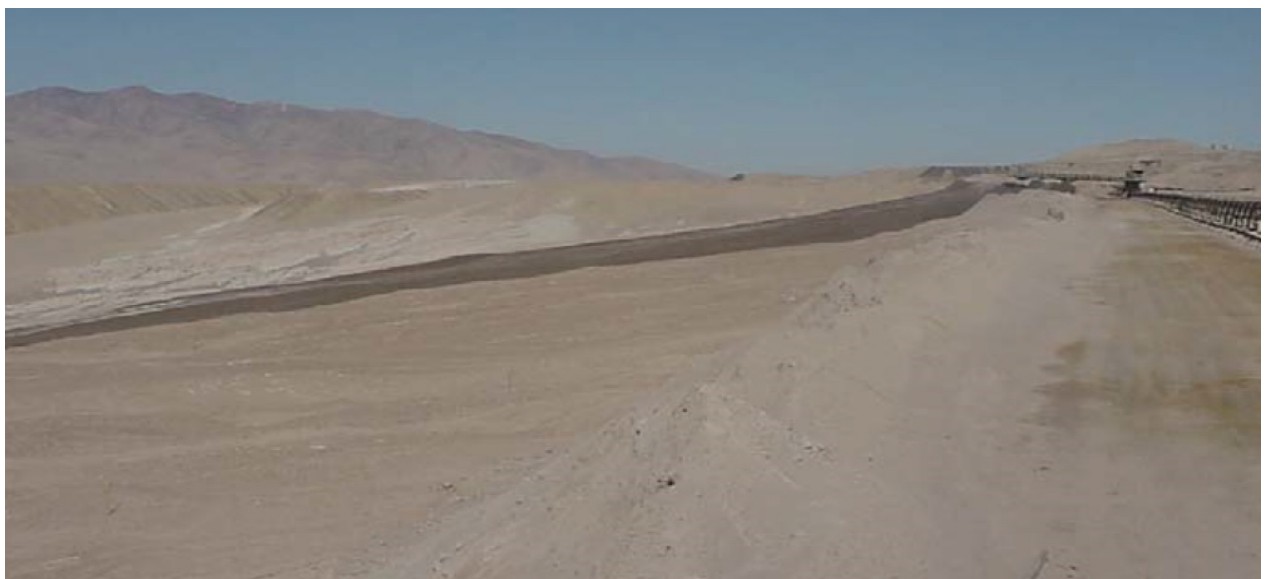

**Figure 49.** Mantos Blancos Disposal of Filtered Coarse Tailings with Mobile Stacker Conveyor and Tripper [18,67,68].

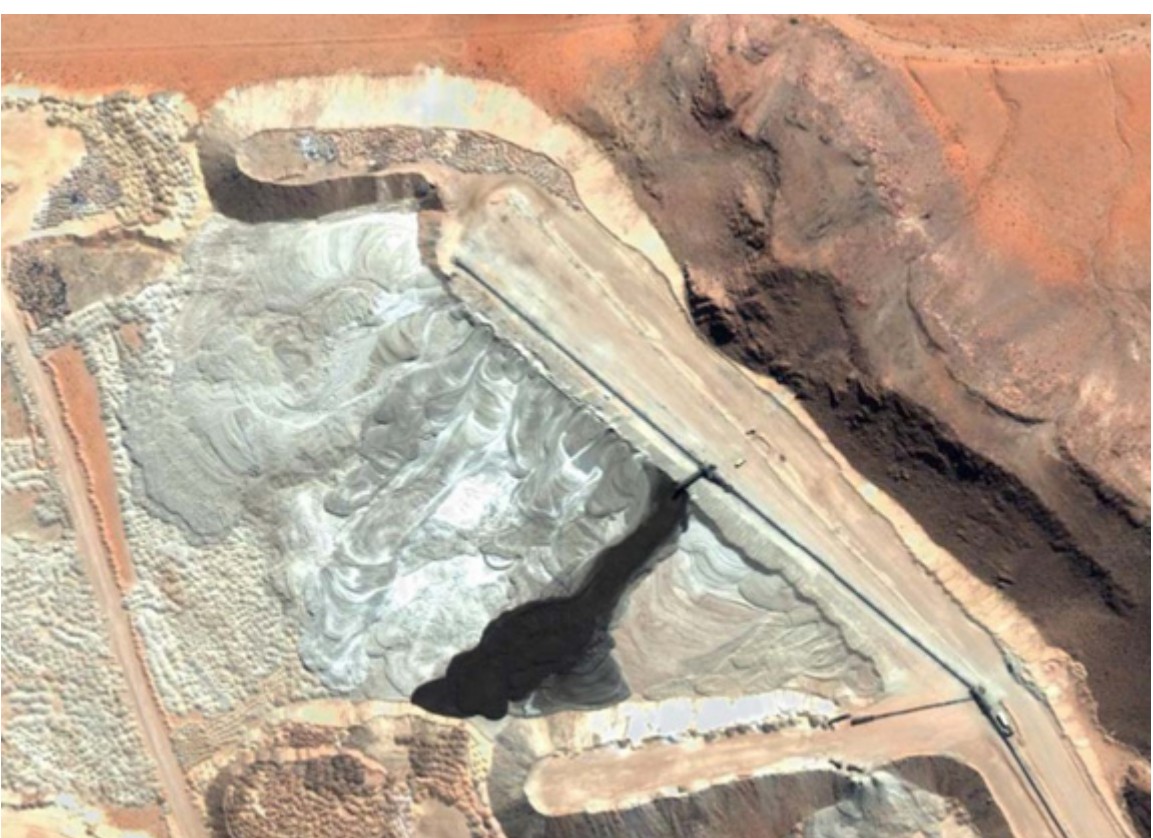

**Figure 50.** Mantos Blancos Aerial View of Disposal of Filtered Coarse Tailings [18,67,68].

### 7.11. Potrerillos Filtered TSF–Valley Topography Configuration–Atacama Desert–Chile

The Potrerillos Smelter is located in the third Region of Atacama, northern Chile at 3000 m above sea level, in the Andes Mountains. Access is through a highway with the city of El Salvador 40 km away. The closest commercial port is Antofagasta, a distant 300 km to the northwest. The city of El Salvador has the El Salvador Bajo commercial airport.

The "Slag Flotation Potrerillos Foundry Teniente Converter" Project includes the elimination of the HLE Slag Cleaning Furnaces, implementing a series of facilities that allow processing of the slag generated in the Teniente CT5 converter, which has an average Cu Law between 8 and 11% (Figure 51) [18,69].

For this, the following processes are implemented:

- Cooling of slag from CT in pots.
- Grinding in SAG mill and Balls.
- Rougher and Scavenger flotation.
- Thickening of concentrate and tailings.
- Filtering of concentrate and tailings
- Transport of filtered tailings to tailings deposit.
- Transportation of concentrate to the concentrate handling area in the smelter.

The final product is copper concentrate, which is recirculated to the Smelter. The daily treatment design capacity of the Flotation Plant is 1700 tonnes of slag, of which the nominal processing capacity corresponds to 1530 tonnes. As a result of the processing, 412 mtpd of copper concentrate and 1288 mtpd of filtered tailings are generated [18,69].

The main target is increasing the water recovery of tailings, and for this, the solid/liquid separation of tailings considers two process unit operations, mainly: (i) thickening (high rate or high-density thickeners), and (ii) filtering with two Ceramic Disc vacuum filters. The tailings are received from the flotation process with solids content by weight (Cw) around 27–30% range, to be thickened to a Cw of approximately 60–65% range, to be fed to

the filters. These units provide a cake-type product typically with a Cw between 85–90% range, with a cake thickness in the order of 10–12 mm, and with moisture content (wet basis) of approximately 10–15% [18,69].

Once the tailings cake is obtained, it needs to be stored in a temporary stockpile to transfer to a filtered tailings conveyance system (Figures 52 and 53). The design of a dry stack TSF needs to consider a filtered tailings conveyance system that needs compatibility with the dry stack construction sequence/plan, using conventional conveyance/haulage and mechanical placement equipment. The main filtered temporary stockpile and transport system used typically is: (i) conveyor belt/chute for temporary filtered tailings stockpile at the discharge of the filter plant, (ii) transfer and loading by a front end loader to haul trucks of filtered tailings, and (iii) haul truck access and operation roads to deposited filtered tailings at TSF [18,69].

The project considers the transport of filtered tailings through 25 tonnes capacity haul trucks (15 m$^3$), which allows for the transportation of approximately 750–2500 m$^3$/day range of generated tailings.

The filtered tailings are commonly deposited in layers of 30 cm thickness, which are exposed to reach their optimum moisture content such as 12%, and are then compacted to reach 95% Proctor Standard (ASTM D698) [19]. The filtered dry stack TSF is a mass composed of several terraces (or benches) and berms. The dry stack TSF is constructed in the upstream direction of a valley topography (Figure 54) [18,69].

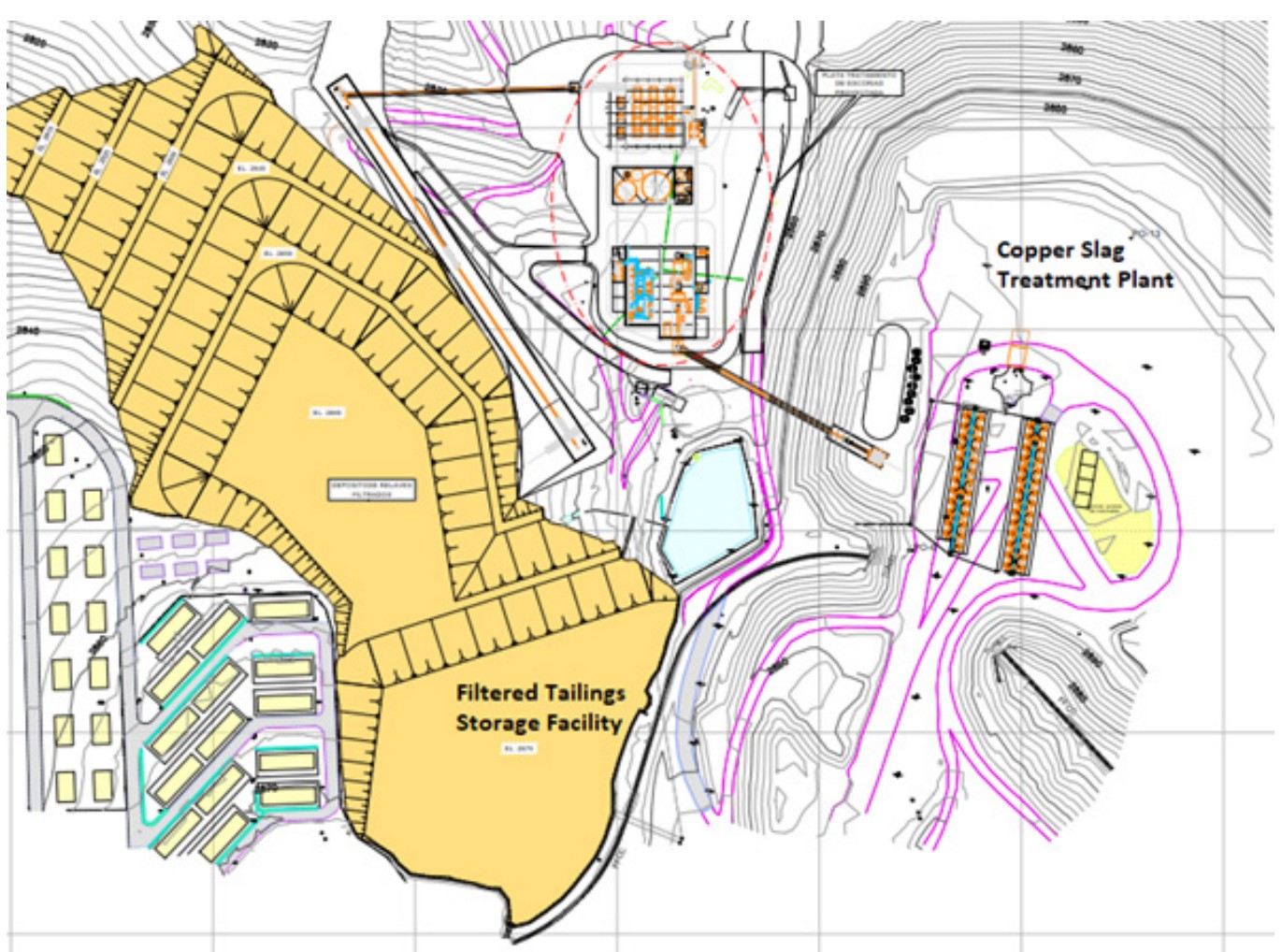

**Figure 51.** Potrerillos Filtered Tailings Storage Facility Layout [18,69].

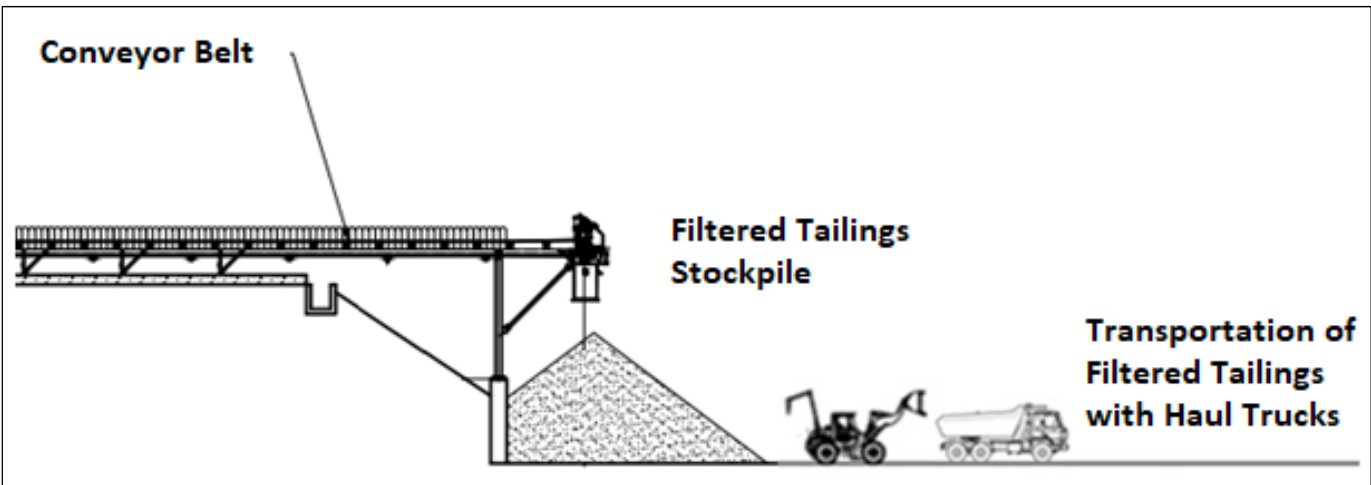

**Figure 52.** Potrerillos Filtered Tailings Storage Stockpile Facility [18,69].

**Figure 53.** Potrerillos Copper Slag Treatment Plant Facility [18,69].

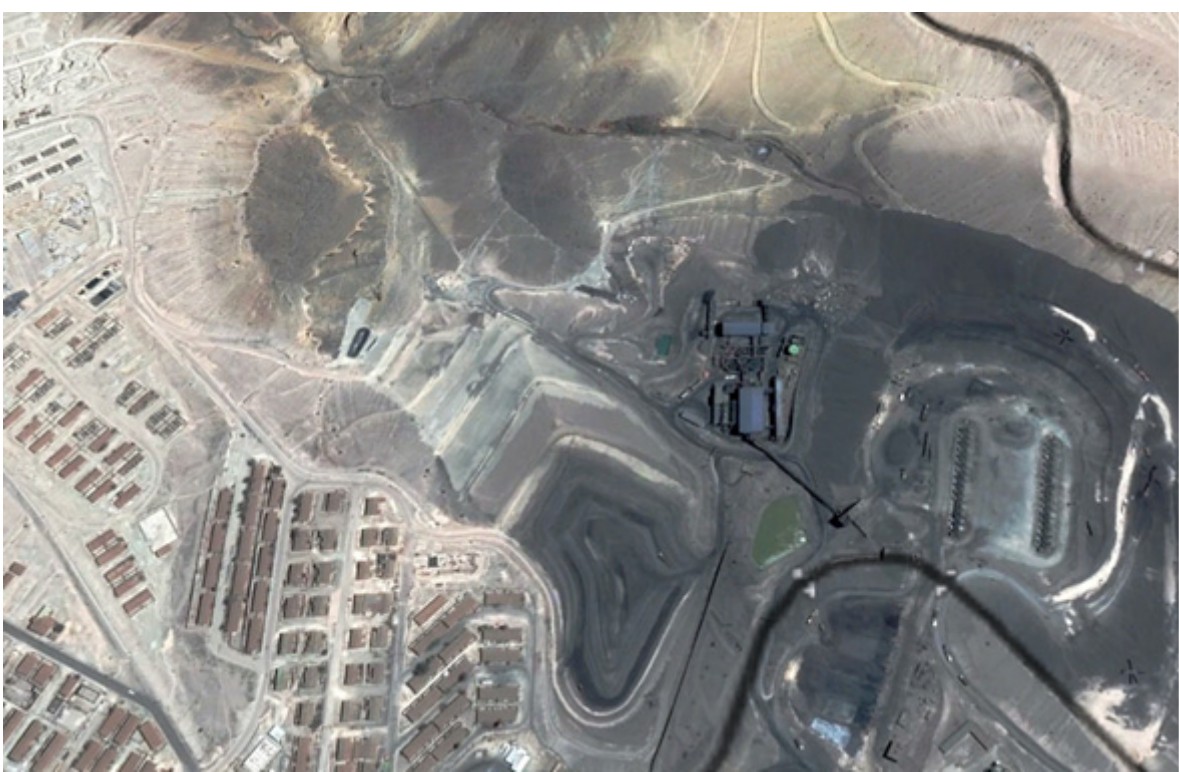

**Figure 54.** Potrerillos Copper Slag Treatment Plant and Filtered Tailings Storage Facility [18,69].

*7.12. Cerro Lindo–Dry Stack of Filtered Tailings–Andean Range–Peru*

The Cerro Lindo Project, property of Compañía Minera Milpo S.A. (Milpo), is located in the province of Chincha, department of Ica, 200 km south of Lima-Peru city. The mine currently processes 7000 mtpd for obtaining polymetallic minerals zinc, lead and copper, using the method of sub-level stoping exploitation. Concentrates produced, in the order of 10% of the processed mineral, are transported by trucks to Lima-Peru city for its commercialization. From the generated tailings, 55% is used for paste filling of the underground mining works, and 45% is deposited on the surface as filtered tailings. The filtered tailings to be deposited in the TSF site called Quebrada Pahuaypite (Figure 55) require compaction in layers of the tailings, due to the elevated site topography and the high seismic activity of the site [20,21,25].

Tailings generated in the process plant are discharged to the surface as filtered tailings and are used as paste fill to be used in underground mining works, 55% of the time tailings are discharged to be used as backfill for underground mining, and 45% of the time they are discharged on the surface [20,21,25].

Tailings are released from the process plant with 30% of solids and are conveyed to a 30-m diameter High Compression Thickener. Underflow tailings from the High Compression Thickener reach 76–78% solids content and are sent to the filter plant. There are two filter plants, one for surface tailings deposition and the other for underground mining. Both of them are located less than 500 m from the above-mentioned facilities [20,21,25].

In the filter plant, tailings are dewatered with belt filters to reach a solids content between 83 and 85% for a proper mix design with cement which varies between 0 and 5% by weight depending on the stope to be filled in the underground mining (elevation 1850 masl). In the filter plant for surface deposition, the filtered tailings have a solids content (Cs) between 87 and 88%, which is equivalent to a moisture content (w%) between 14 and 15%. These tailings at this moisture content can form stacks with slopes of H:V = 1.3:1.0. Tailings are handled with wheel loaders to trucks with a capacity of 25 tonnes.

Trucks discharge tailings into the tailings deposit by sectors to obtain the dewatering and compaction cycles [20,21,25].

Likewise, the topographic conditions and the narrowness of the Quebrada Pahuaypite made it advantageous for the tailings transport to be by means of trucks instead of conveyors. An additional advantage of compacting tailings in layers was that it created a medium of low permeability, diminishing the risks of infiltration of rainwater, and inhibiting the entrance of oxygen, thus minimizing the generation of acid drainage [7].

Tailings are processed in a filtration plant by vacuum belt filters, obtaining a cake with 18% moisture content. The filtered tailings deposit is composed of two desiccation platforms of approximately 2250 and 14,050 m$^2$ each, located at elevations 2028 and 2082 masl, respectively. In these platforms, filtered tailings are discharged, and spread by sectors to obtain desiccation and to allow further compacting [20,21,25].

The tailings deposit has been designed to store 14 Mt of tailings and is built as a large fill which will have an overall slope of H:V = 2.8:1.0 (Figure 55). The tailings storage construction has benches with minimum width of 10 m each and 20 m high (Figure 56). As part of the tailings deposit, a basal drainage system has been provided for seepage collection from the tailings deposit. The installation of inclinometers and underground water monitoring wells is also considered. They are progressively installed in berms that will form the deposit [20,21,25].

To obtain tailings that achieve a minimum initial dry density, that is seismically stable, and not below 2.8 t/m$^3$, the tailings need to be first dewatered until reaching a moisture content in the range of 5–7% and then compacted to 95% standard proctor density. To reach a dry density for compaction, operation cycles from 3–5 days are applied depending on the period of the year. In summer (December to March), the wet season, cycles could last five days. For the rest of the year, cycles could last approximately three days. Operation cycles consist of the following activities: discharge, spreading, drying–compacting (Figure 57).

The layer thickness to form tailings is 30 cm, and after reaching moisture from 5–7%, tailings are compacted through vibrating rollers of 10 tonnes until reaching the required density, which is obtained generally with two cycles of roller passes [20,21,25].

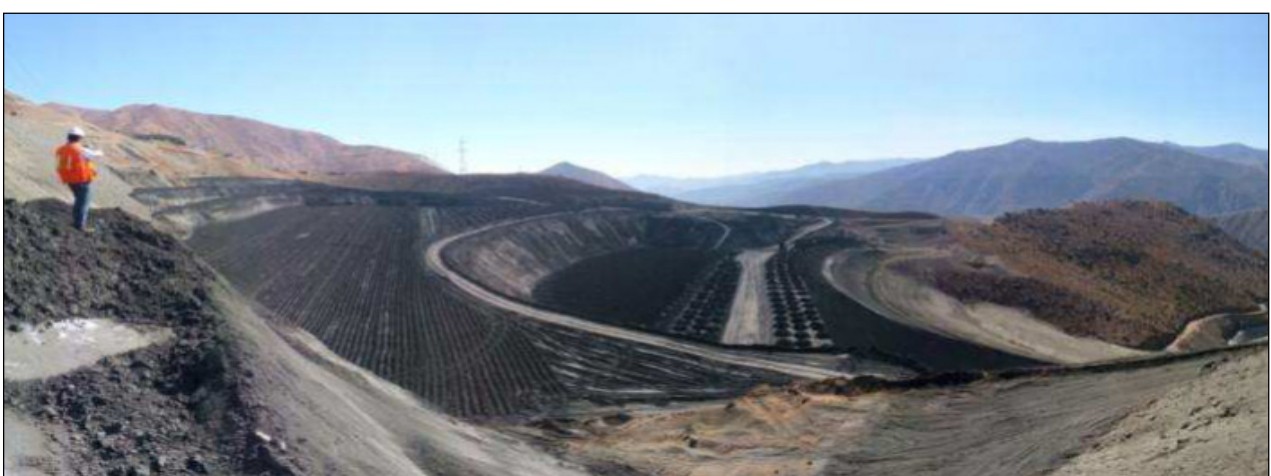

**Figure 55.** Cerro Lindo Filtered TSF Overview [20,21,25].

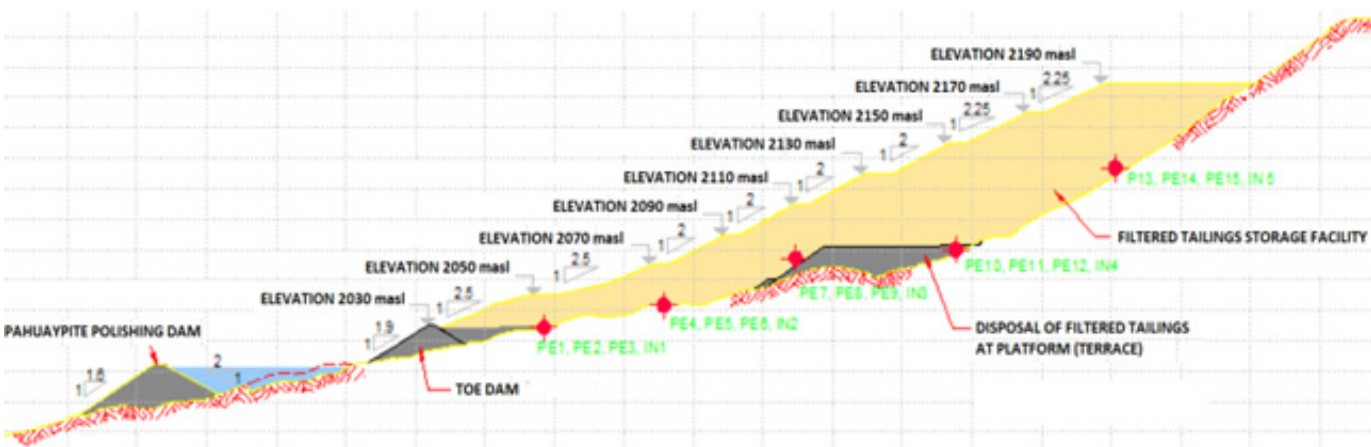

**Figure 56.** Cerro Lindo Filtered TSF Typical Cross Section [20,21,25].

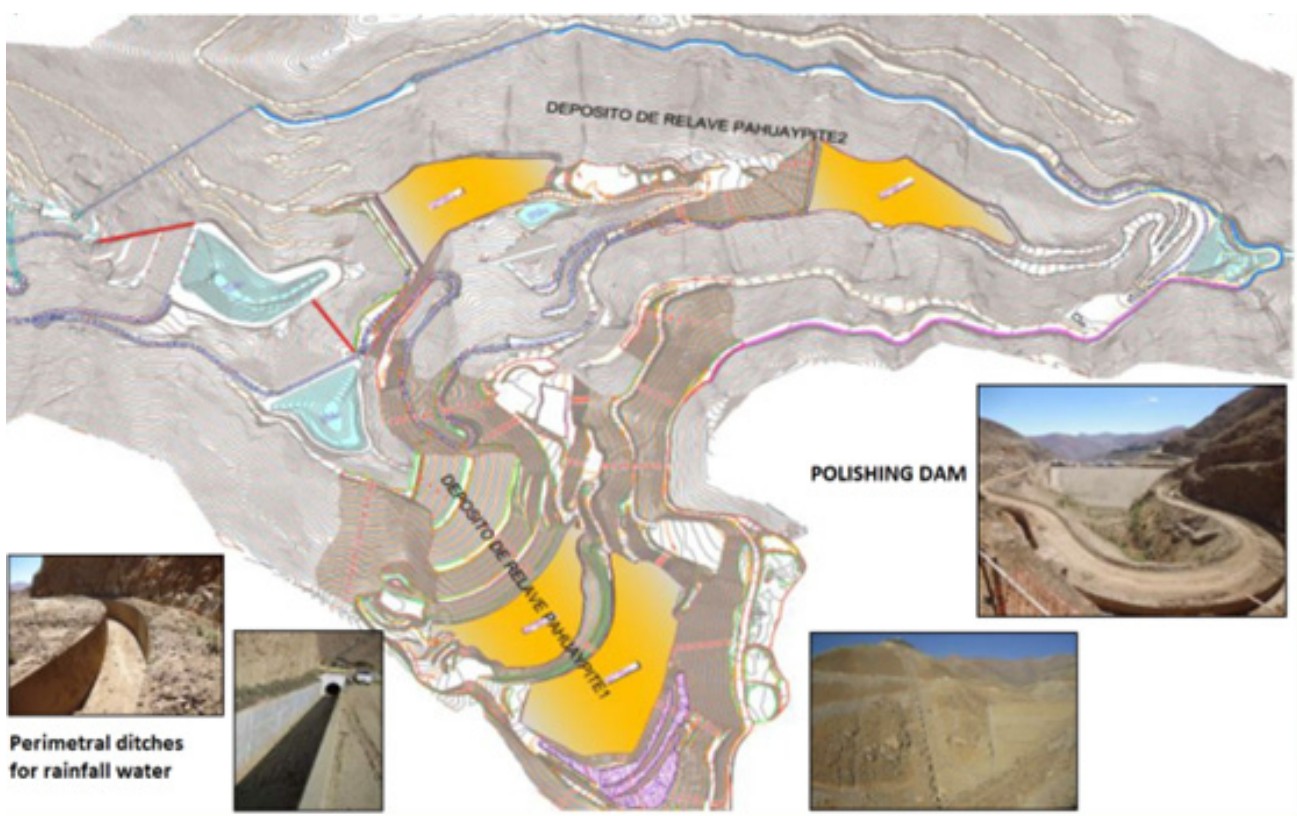

**Figure 57.** Cerro Lindo Filtered TSF Layout—Pahuaypite 1 TSF and Pahuaypite 2 TSF [20,21,25].

*7.13. Poderosa Filtered TSF–Valley Topography Configuration–Andean Range–Peru*

The project area, called Pataz Batolito specifically, the Marañón tailings deposits are located on the right bank of the Marañón river valley, in the district and province of Pataz, department of La Libertad, at an average altitude of 1180 to 1280 masl in the Unit Production Marañón, which in turn is part of the Compañía Minera Poderosa.

At the Marañón metallurgical plant, gold is recovered through the cyanide leaching or direct cyanidation process. The total treatment capacity of the plant is 700 tonnes per day. Once the gold dissolves, it is precipitated through the Merrill Crowe process and melted down to obtain the gold bars that are traded. The recoveries of cyanide solution obtained in the plant are above 92%, the residues are stored in filtered tailings deposits, where all the effluents that can be generated are controlled to avoid contamination [36,37].

The tailings filtration plant has a treatment capacity of 700 mtpd. Thickener No. 4 discharges tailings in a slurry with a density of 1650 tons/m³, passing through an automatic sampler. The slurry is driven by gravity to the agitator, from which it is pumped to two press-type filters. There, the tailings, with a humidity of less than 15%, are deposited in a temporary storage pile by means of a conveyor belt. Finally, it is transported with dump trucks to the Asnapampa tailings deposit for its compaction and final disposal (Figures 58 and 59). The recovered filtered liquid is deposited in the recovery tank, then with the activation of a pump, the solution returns to Thickener No. 3 and the solution is recirculated to the metallurgicalplant [36,37].

With the change of disposal to filtered tailings with 13% moisture equivalent to the optimum moisture content (±2% tolerance), physical stability has been improved for static and pseudo static conditions, as well as ensuring hydrological and hydrogeochemical stability and compliance with these requirements [36,37].

The filtered tailings are deposited in terraces that are adapted to the topography of the place. There are banks 10 m high and berms 2 m wide, with local slopes of H:V = 1.5:1.0 (Figures 60 and 61).

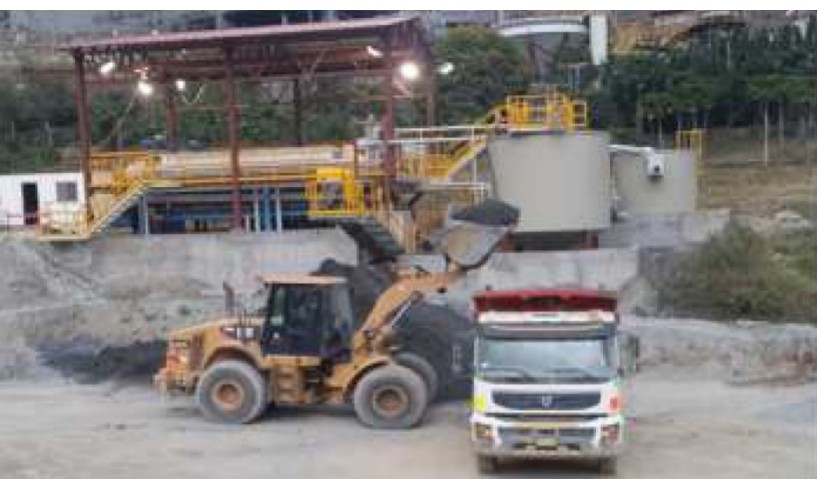

**Figure 58.** Poderosa Filtered Tailings Plant [36,37].

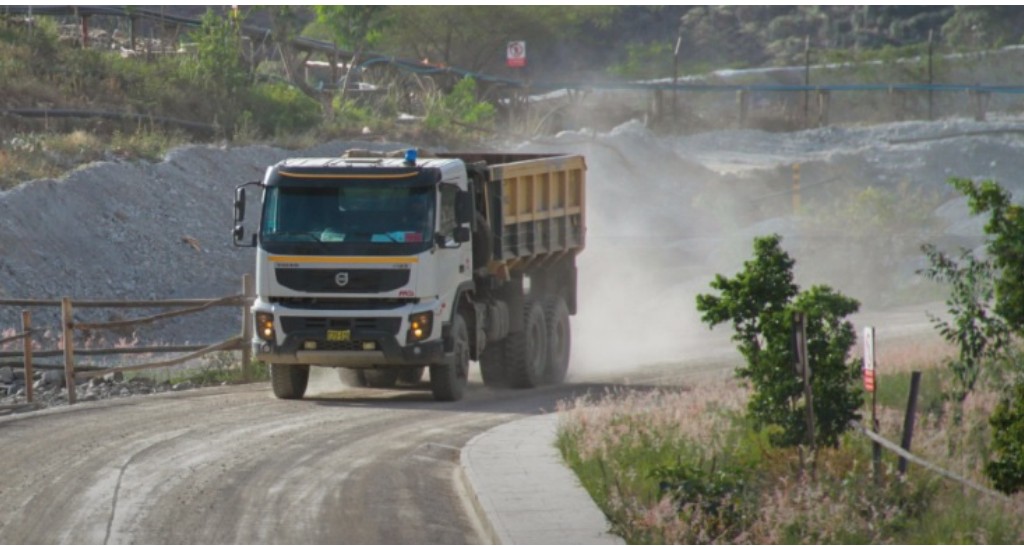

**Figure 59.** Poderosa Haul Trucks with Filtered Tailings [36,37].

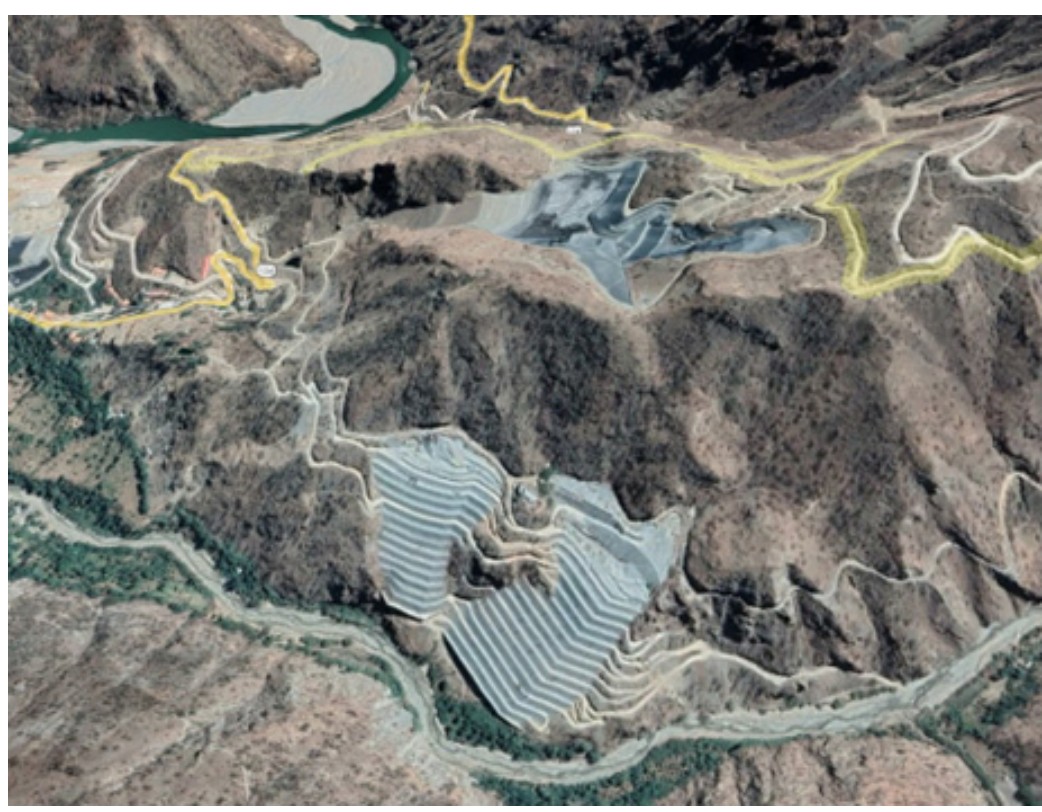

**Figure 60.** Poderosa Filtered Tailings Storage Facility Layout [36,37].

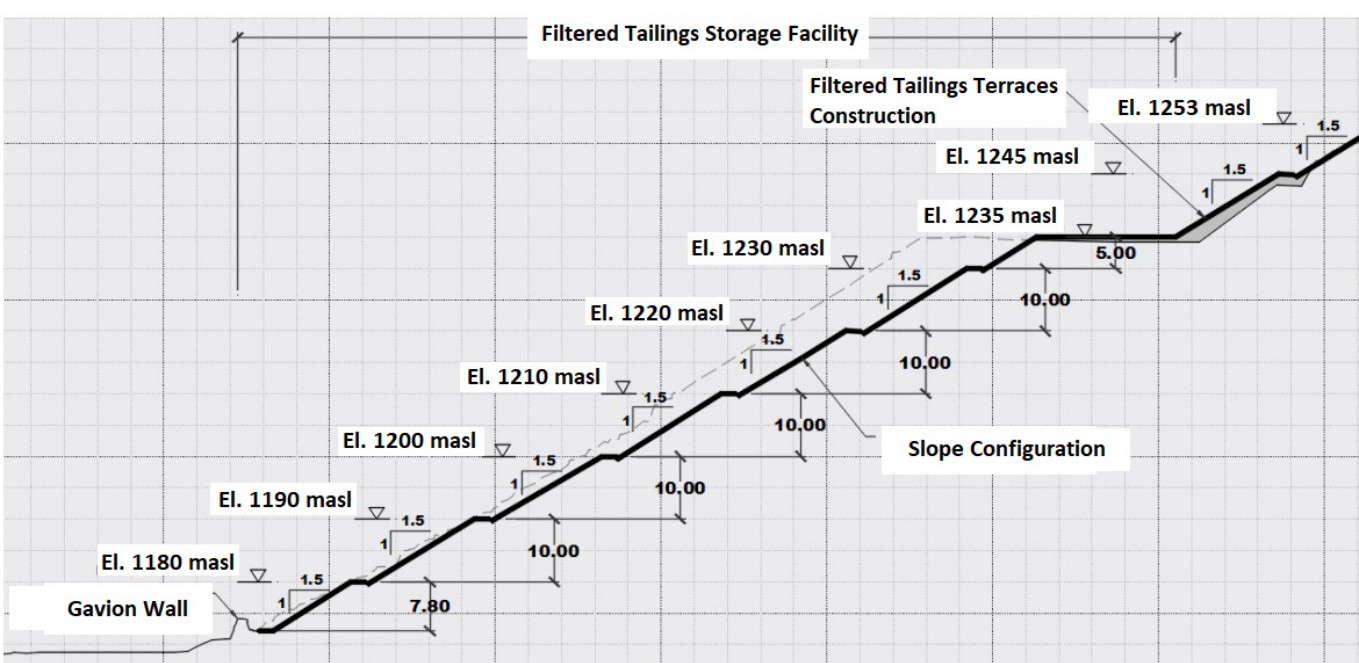

**Figure 61.** Poderosa Typical Cross Section Filtered Tailings Storage Facility [36,37].

### 7.14. Curaubamba Filtered TSF–Valley Topography Configuration–Andean Range–Peru

Consorcio Minero Horizonte (CMH) exploits a gold deposit in the district of Parcoy, province of Pataz, department of La Libertad. As a consequence of this exploitation, mining residues such as tailings and waste are produced, which have to be deposited in suitable

places, and it is in this sense that CMH has decided to expand the storage capacity of the Curaubamba tailings and waste deposit [38,39].

The project area is located in the Curaubamba basin, on the left bank of the Parcoy River; adjacent to CMH's Chilcapampa tailings deposit.

The new tailings filtering plant considers the processing of tailings from the Parcoy plant from its "overflow" process with a projected capacity of 2000 mtpd, receiving tailings with a 30% solid content by weight, until it is thickened and filtered, obtaining a "cake" with approximately 85% solids content by weight and a humidity of 15%. The filtration equipment corresponds to press-type filters [38,39].

The project for the regrowth of the tailings deposit consists of the use of the Curaubamba creek, for the storage of the tailings and waste generated by the mining activity in the plant. The project will have two deposits, one for mine waste material and another for filtered tailings (Figures 62 and 63) [38,39].

The storage capacity of the regrowth of the tailings and waste deposit has been estimated at 22 years, to take advantage of the maximum capacity of the Curaubamba creek [38,39].

The design of the regrowth of the tailings and waste deposits will consist of raising the existing deposits. In other words, in the case of the filtered tailings deposit, it is founded on the final elevation of the existing deposit at the 2605 masl level and grows by means of a 30-m-high compacted embankment until it reaches the 2635 masl level. In the case of the waste rock material deposit, this regrowth is carried out by progressively raising the eight existing banks of waste rock, by a maximum height of 30 m for each bank (Figures 64–66). The following characteristics are present in the filtered tailings deposit:

- Material of the embankment: filtered tailings.
- Maximum elevation: 2635 m above sea level.
- Starting level: 2605 m above sea level.
- Number of TSF: 01.
- TSF height: 30.0 m.
- Deposit slope: H:V = 2.5:1.0.
- Storage Volume: 1.15 M m$^3$.
- Operation time: 22.0 years.

For the transport of tailings, trucks with a capacity of 15 m$^3$ are used, which will move from the processing plant to the tailings deposit, approximately 200 m east of the processing plant, with a frequency of 40 trips/day in the highest processing stage of the plant [38,39].

The layer thickness to form tailings is 30 cm, and after reaching a moisture content from 10–12%, tailings are compacted through vibrating rollers of 10 t until reaching the required density, which is obtained generally with two cycles of roller passes [38,39].

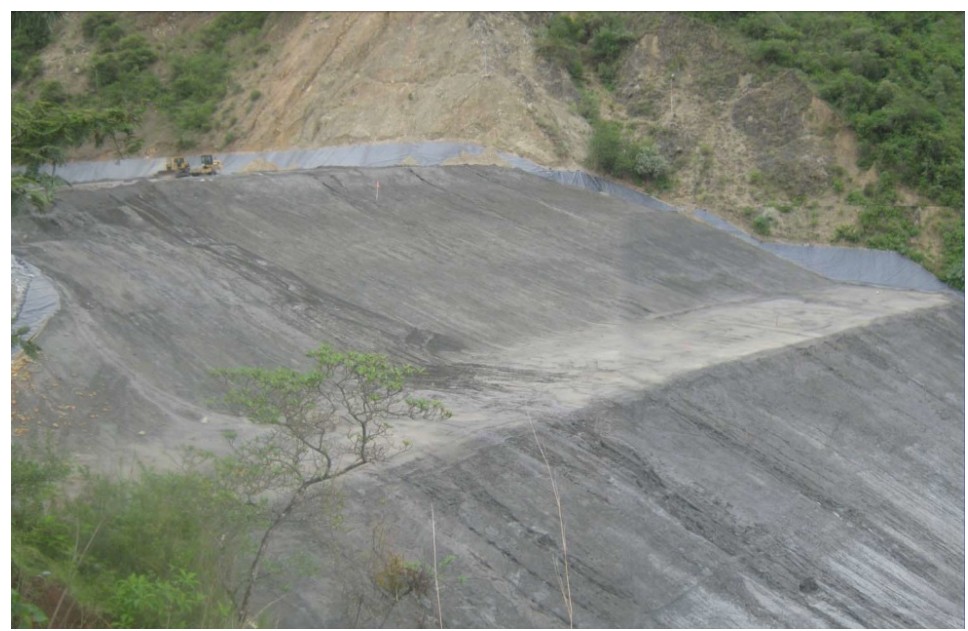

**Figure 62.** Curaubamba Filtered TSF–Terraces View [38,39].

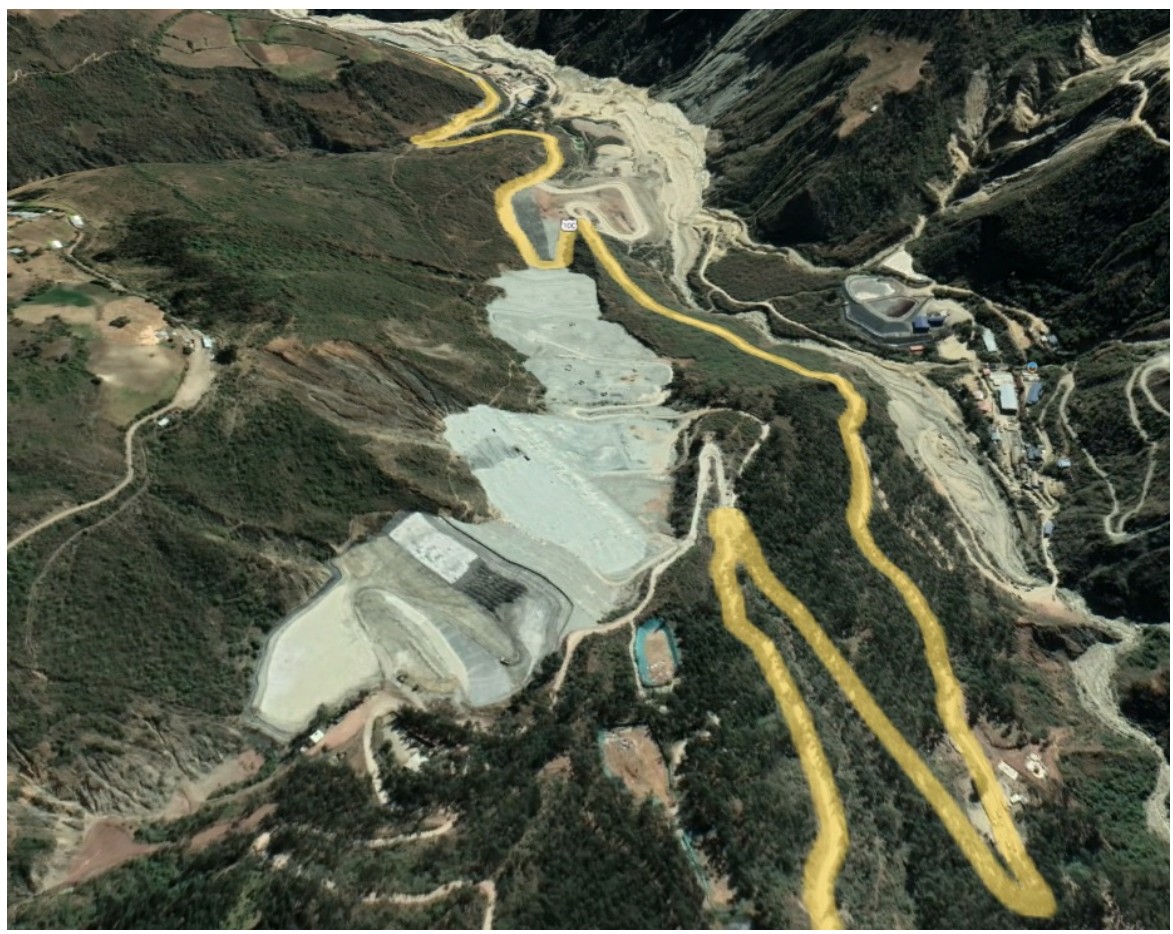

**Figure 63.** Curaubamba Filtered TSF—Layout [38,39].

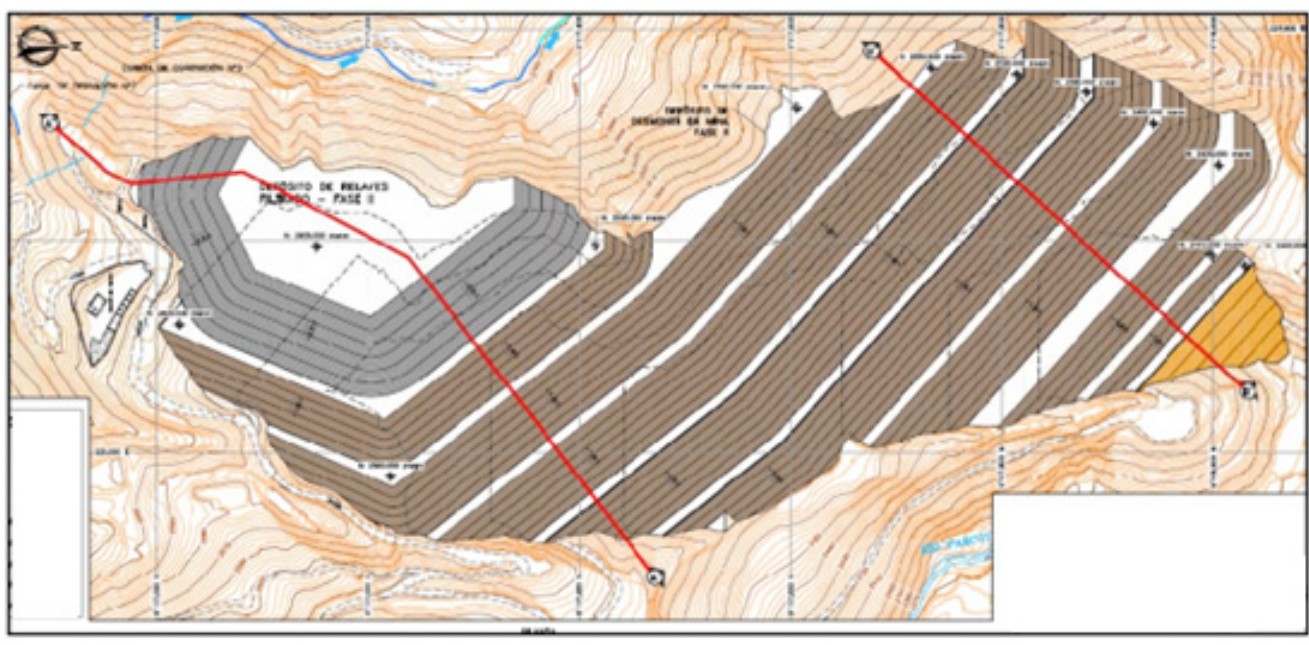

**Figure 64.** Curaubamba Filtered TSF—Terraces Overall View [38,39].

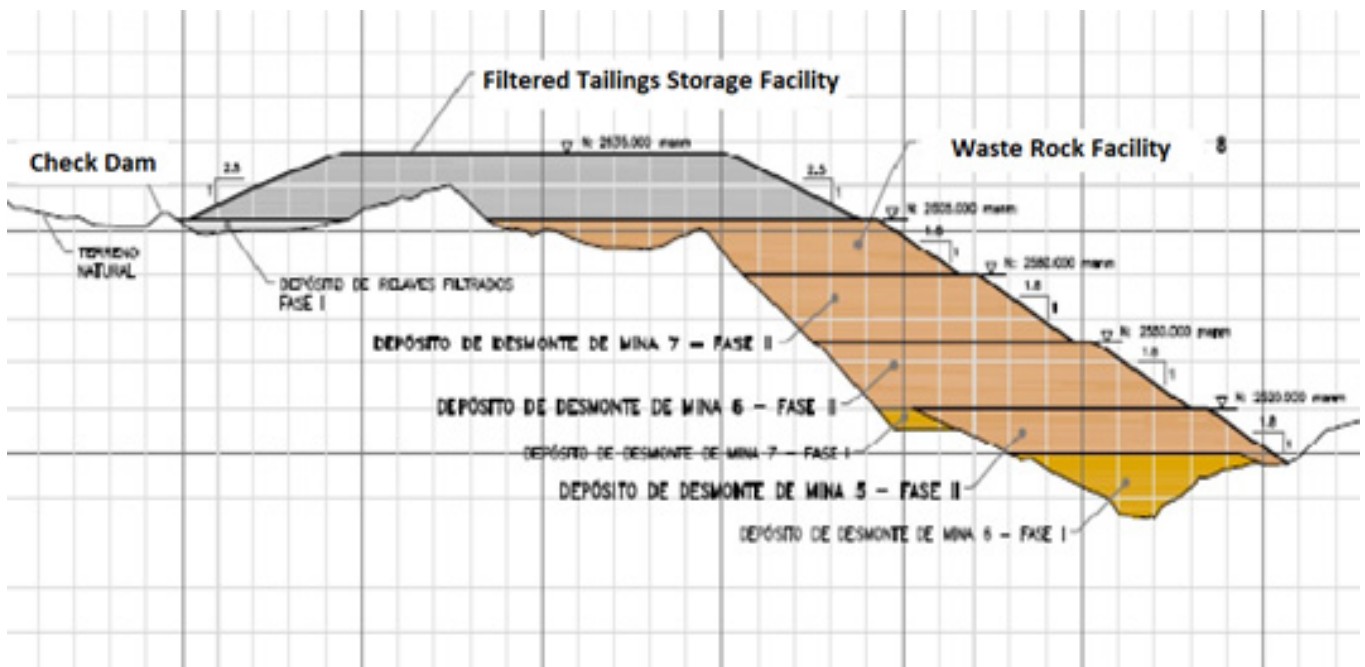

**Figure 65.** Curaubamba Filtered TSF–Typical Cross Section [38,39].

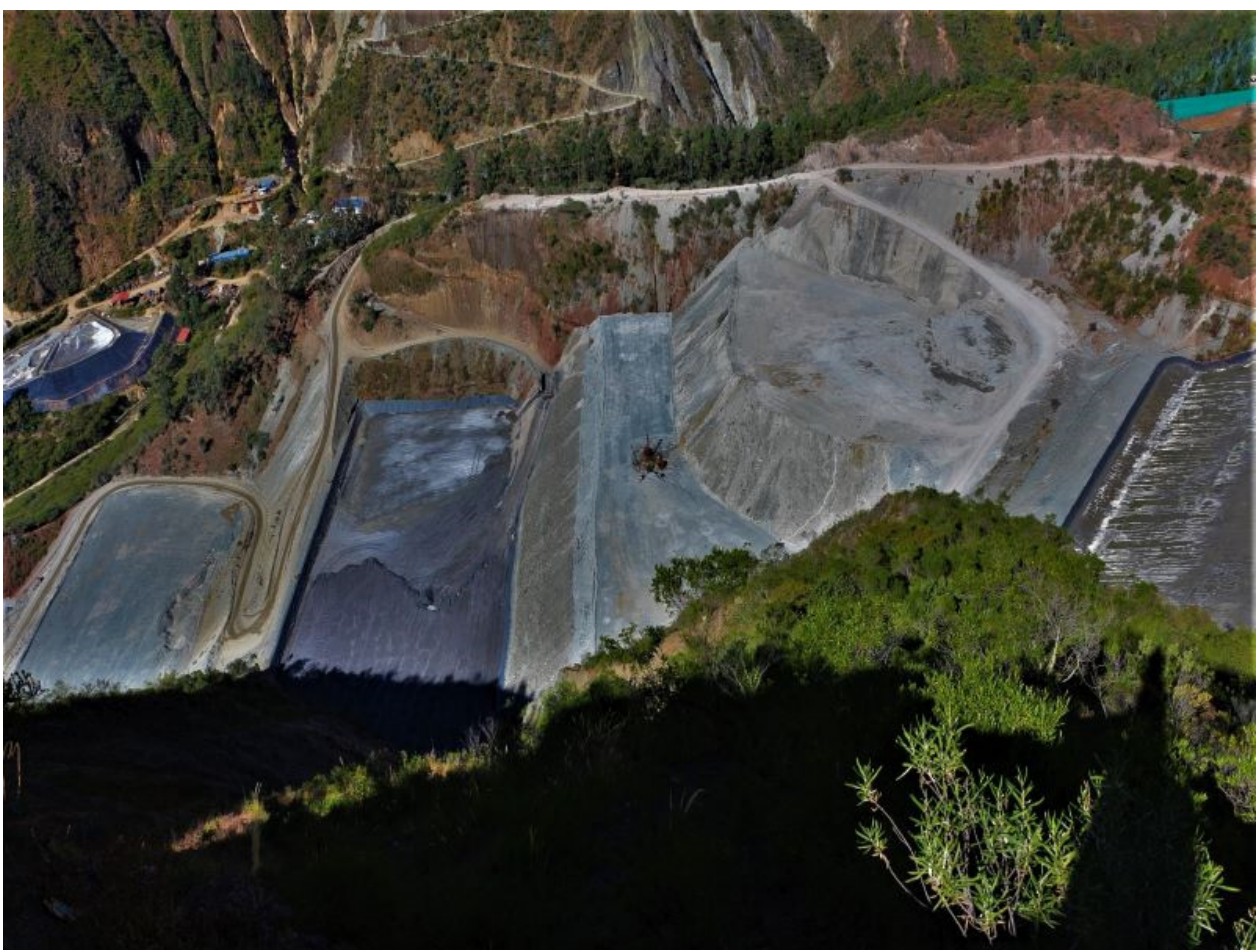

**Figure 66.** Curaubamba Filtered TSF–Benches and Terraces [38,39].

*7.15. Catalina Huanca Filtered TSF–Valley Topography Configuration–Andean Range–Peru*

Catalina Huanca Sociedad Minera S.A.C (CHSM) is a mining company that is dedicated to the exploitation and processing of polymetallic minerals; where minerals such as zinc, lead, and silver are extracted through underground mining and transferred to the plant in 3-axle and 4-axle dump trucks; being processed with a capacity of 2000 mtpd. There is an area for the tailings deposit and auxiliary facilities in the Ramahuayco project area, located in the district of Canaria, province of Víctor Fajardo, department of Ayacucho at an altitude of 3500 m above sea level [40,41].

The extracted ore is processed by conventional flotation at the San Jerónimo Beneficiation Plant, which has a capacity of authorized treatment of 2000 mtpd, in which concentrates are obtained that are transported by highway via Nazca to the port of Callao. The Catalina Huanca Mining Unit currently has 5407 hectares of mining concessions.

The Ramahuayco tailings deposit project is located to the west from the mine, in the upper sector of the Sacllani ravine, 150 m upstream from the mine offices, with respect to the concentrator plant, which is 12 km away (Figure 67) [40,41].

The project is for ore processing with a current nominal production capacity of 2000 mtpd, an average concentrate rate of 15%, and a filtered tailings production of 1850 mtpd. The ore processing is carried out using the conventional flotation system. The generated tailings pass to a thickener and is then transported through pipes to the filtering plant [40,41].

The filtering plant is conformed by a press filter that receives the tailings from the concentrator plant with 65% solids content, where the tailings are transformed into pulp to

filtered tailings, with 88% solids content (Figure 68). After that, the tailings are transported by trucks to the filtered tailings deposit area [40,41].

For stability reasons, filtered tailings require compaction. In this case, in situ tailings dewatering is required. Tailings production, rainfall-evaporation of the site, drying time and areas, and operational flexibility can limit the application of filtered tailings, and the key parameter is the water content in the deposited tailings (Figure 69) [40,41].

Located in the Sacllani basin, the filtered tailings are transported by trucks, from the filtering plant to the tailings deposit facilities, where they are unloaded, spread, dried, and compacted to 95% of the Standard Proctor [19], for which the maximum humidity of tailings compaction must be less than 12% (Figure 70) [40,41].

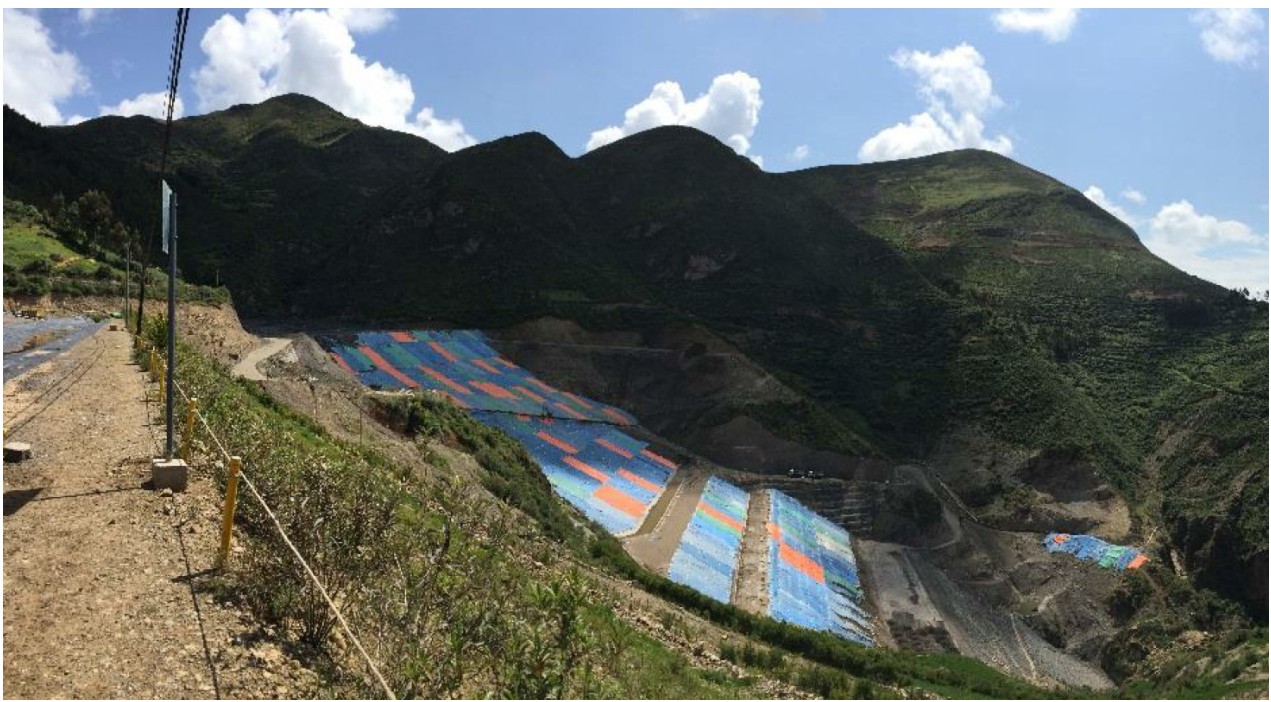

**Figure 67.** Ramahuayco Filtered TSF—Terraces Overview [40,41].

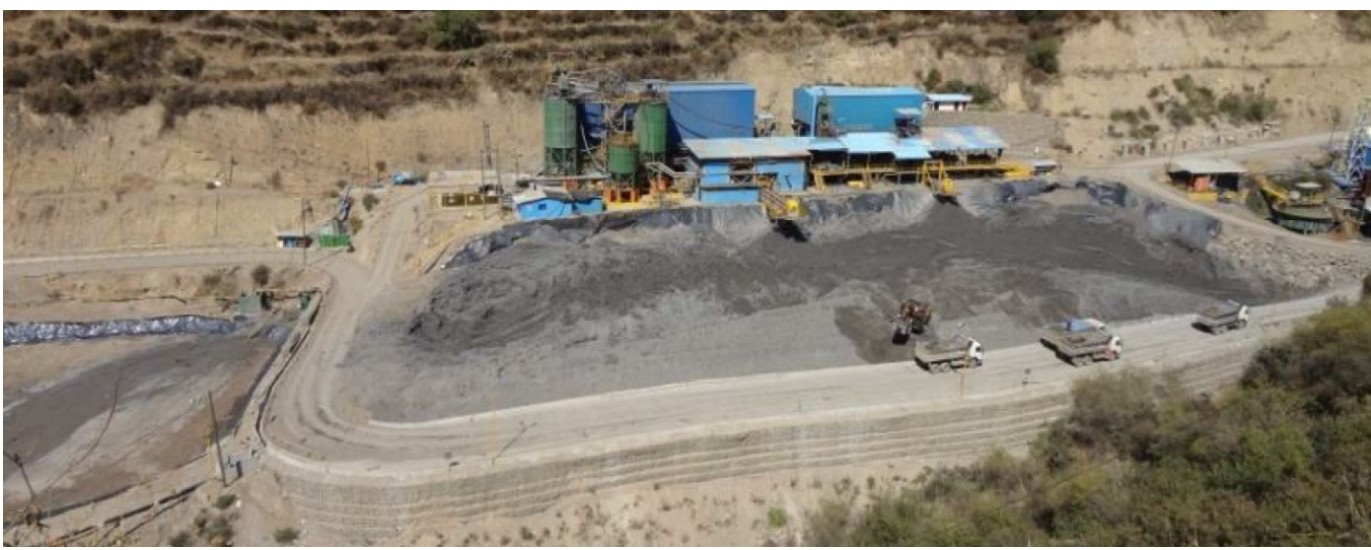

**Figure 68.** Ramahuayco Filtering Plant [40,41].

The tailings deposit, in its first stage, is made up of the main dam and a structure of boulders arranged at the base. The entire structure is supported on limestone rock, the project plans the construction of a dam up to a level of 3584 m above sea level, the formation of a base embankment with borrowed material, and the placement of a foot structure made up of a dam of plain concrete [40,41].

Tailings drying can be done in two ways; the first with a bulldozer which removes the tailings and leaves them in a furrowed form in a certain area; the second is by means of excavators, this is, using its spoons to remove the tailings, passes the shovel, and vents it in such a way that the tailings lose the initial humidity that it has of 17 dropping to 12% to be ready to be mixed with the borrow material [40,41].

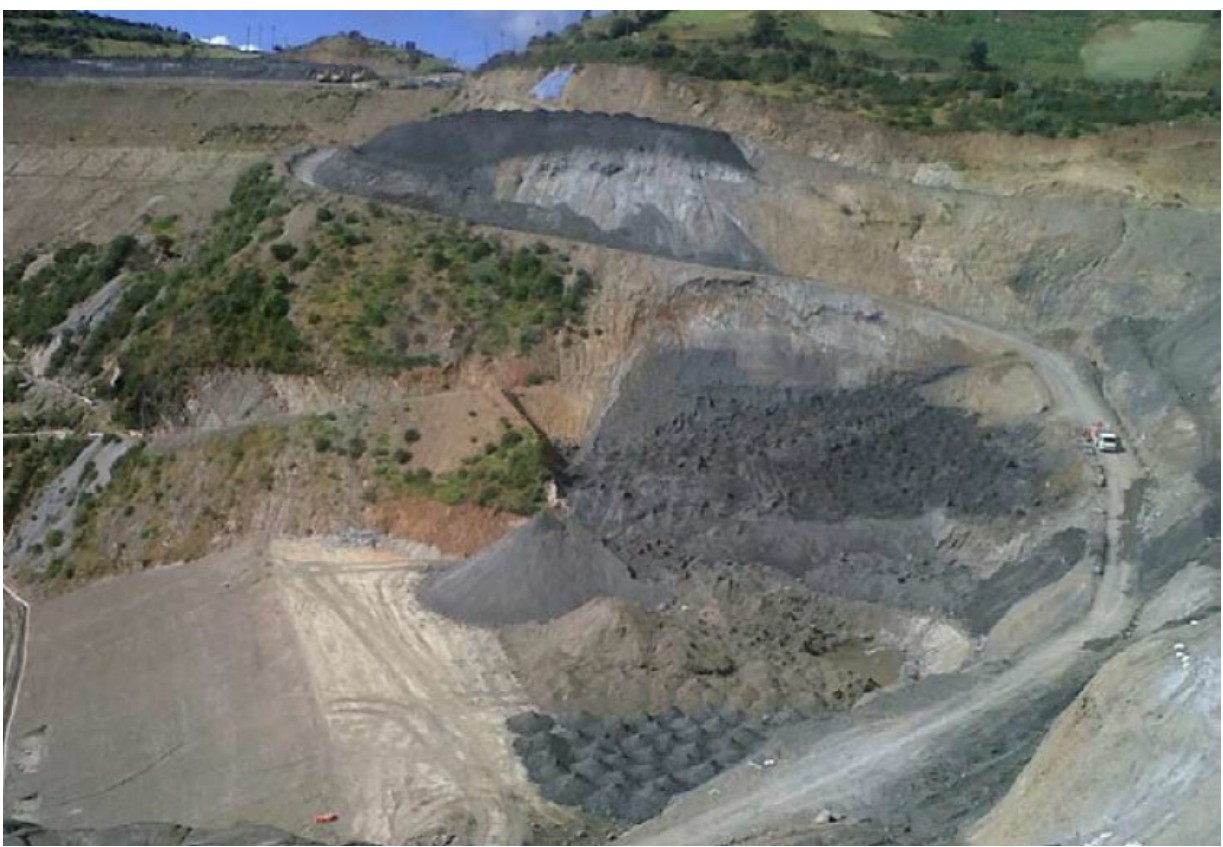

**Figure 69.** Ramahuayco TSF Overview [40,41].

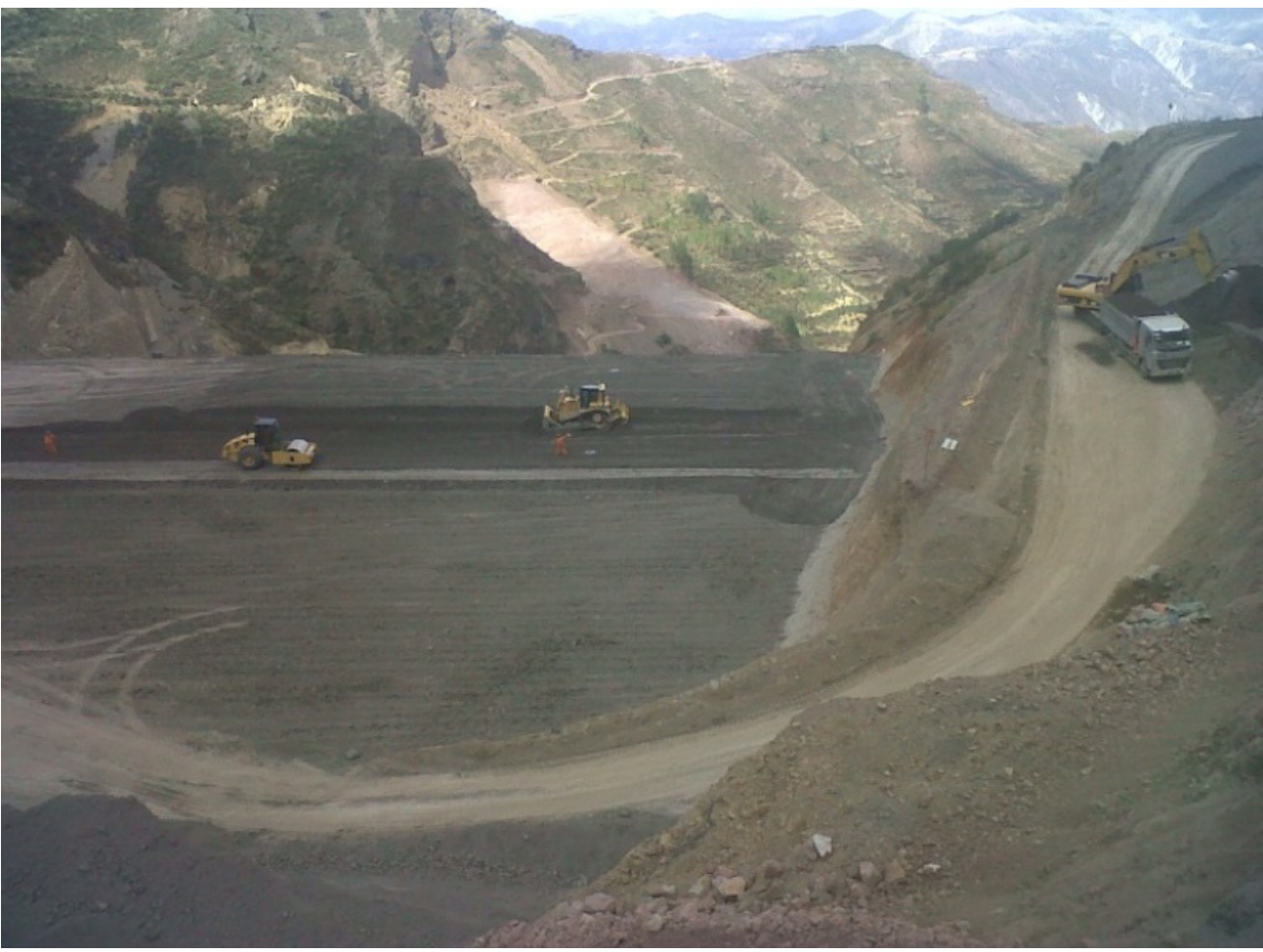

**Figure 70.** Ramahuayco TSF Civil Works [40,41].

*7.16. Tambomayo Filtered TSF–Valley Topography Configuration–Andean Range–Peru*

The Tambomayo project is politically located in the annex of Punachica and Tocallo, district of Tapay, province of Caylloma, in the department of Arequipa. It is located 54 km southwest of the Caylloma district and 100 km east of the Paula mine, at an altitude between 4700 and 4900 m above sea level [42,43,70].

Tambomayo is an epithermal-mesothermal polymetallic deposit in veins of gold and silver with base metals in quartz gangue, emplaced in Tertiary andesitic volcanics. Gold is found in its native state and electrum, while silver is found in sulfides and sulfosalts accompanied by galena and sphalerite. There are two main vein systems: Mirtha, NW–SE oriented, and Paola, N–NE oriented. There are also nearby exploration areas [42,43,70].

The Tambomayo metallurgical process consists of primary crushing, fine grinding, gravimetry, and cyanidation in tanks to obtain a solution rich in gold and silver to be processed at the Merrill Crowe plant (zinc precipitation). The precipitate is dried and melted obtaining doré bars (gold and silver). The cyanidation tailings enter the flotation process, from which lead-silver and zinc-silver concentrates are obtained (Figure 71) [42,43,70].

The average production of tailings to be thickened, filtered, and deposited is 1500 mtpd from the discharge of the tailings thickener from the process plant.

The tailings thickener feeds the filtering plant that considers two press filters (it comes from the cyanide destruction system and goes to the tailings distributor). The filtering plant considers the cloth washing systems, the membrane compression system and the transport by trucks of the filtered tailings to the tailings deposit (includes the truck tire washing system) at the exit of the filtering plant (Figure 72) [42,43,70].

The Filtered Tailings Deposit 1, through a drainage system, collects and directs the waters that are generated through the drying of the tailings that were previously filtered through the filter presses. The percentage of moisture that remains in the remainder of this operation varies between 8 to 14% humidity, being a percentage of filtered water that accounts for between 80 to 140 liters of solution for each metric tonne, and if this is the case, this amount of water will filter through the drainage system and will be stored in this pool for storage and subsequent treatment (Figure 73) [42,43,70].

The filtered tailings are transported by trucks to the tailings deposit area which is lined with geosynthetic geomembrane materials throughout its base. Once the filtered tailings are placed in the deposit, they are spread in 30 cm layers and subsequently compacted with a 10-tonne drum-type vibratory roller (Figure 74) [42,43,70].

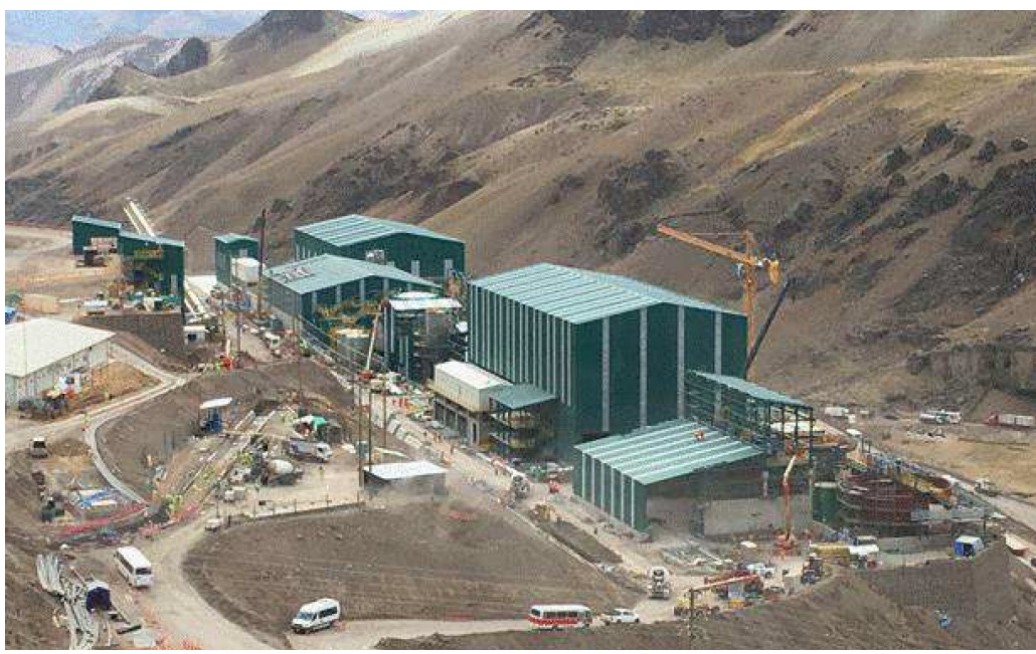

**Figure 71.** Tambomayo Concentrator Plant [42,43].

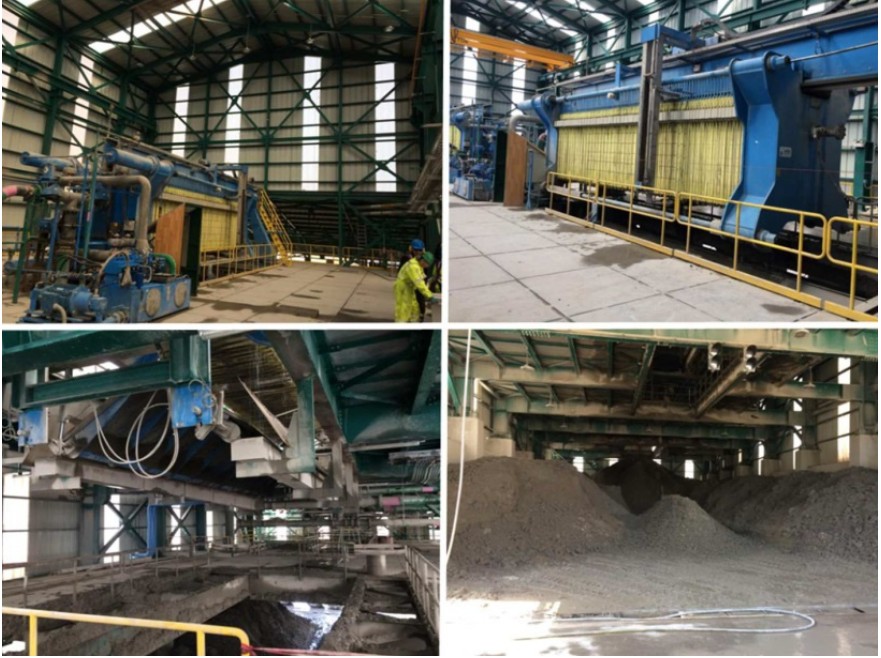

**Figure 72.** Tambomayo Tailings Filtering Plant [42,43].

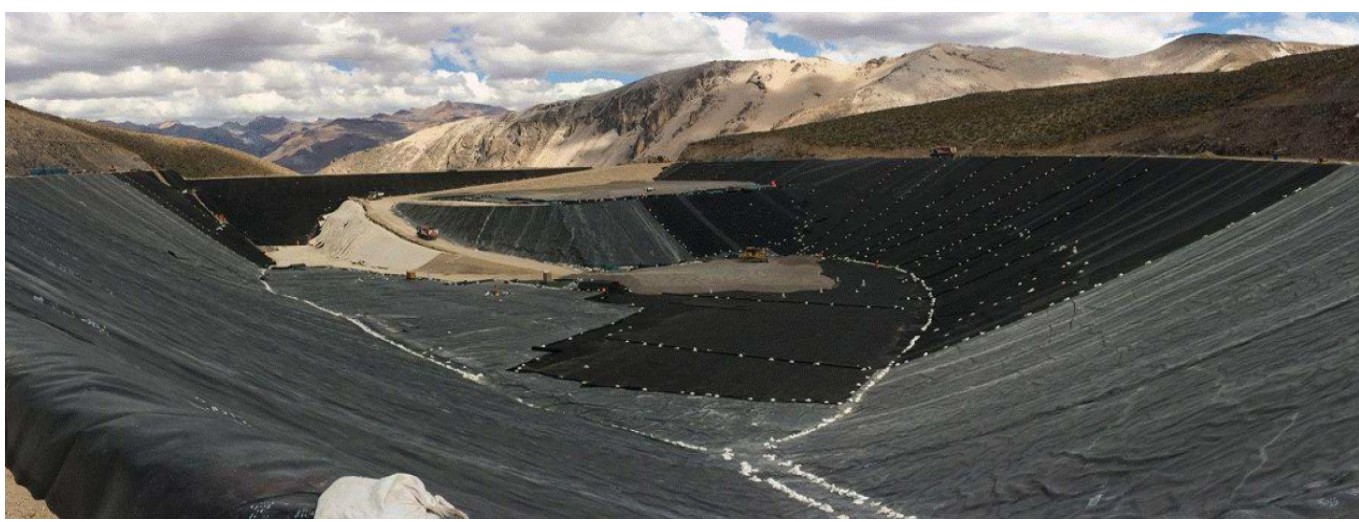

**Figure 73.** Tambomayo Filtered TSF [42,43,71,72].

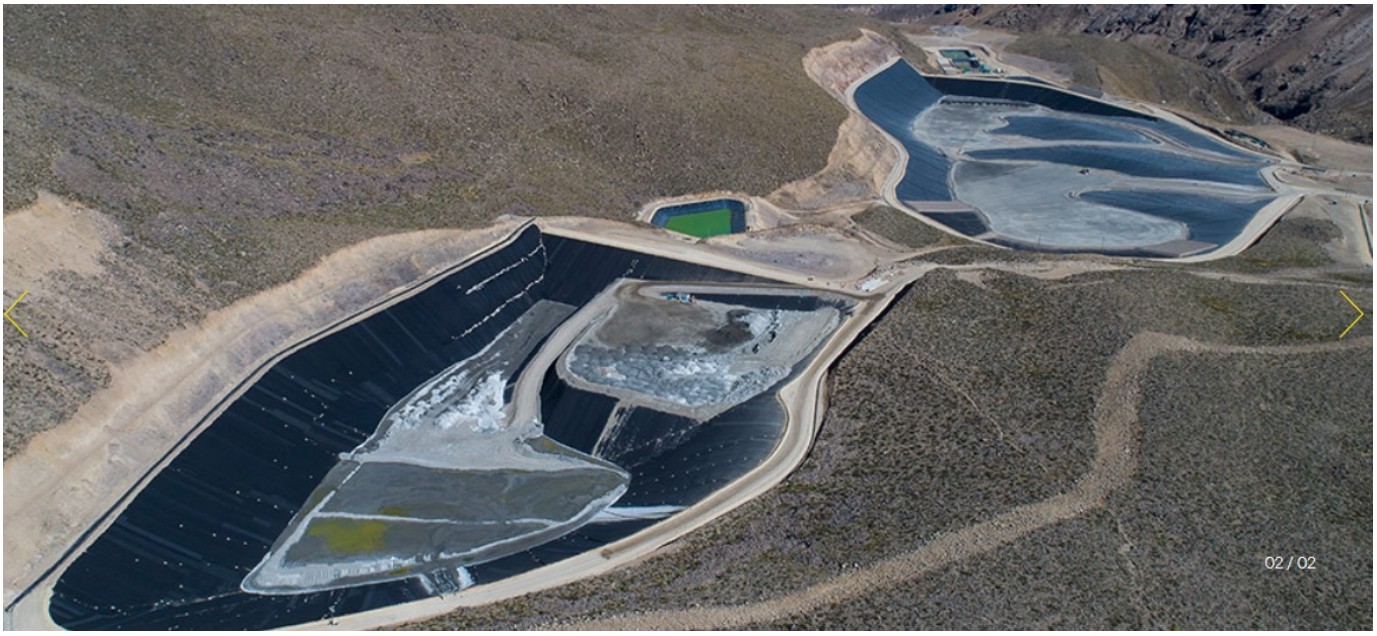

**Figure 74.** Tambomayo Filtered TSF Normal Operation and Contingency Operation Overview [42,43,71,72].

*7.17. Chungar Filtered TSF–Valley Topography Configuration–Andean Range–Peru*

The Chungar Mine is located in the Department of Pasco, Province of Cerro de Pasco, and District of Huayllay, Peru. It is a polymetallic deposit that contains copper, silver, zinc, and lead, an underground working mine, and whose workings are at the 4100 masl.

The process begins with the pumping of the flotation tailings towards the hydraulic fill hydrocyclone battery, which is carried out with a pumping line of 02 pumps in series with 150 HP of power and another line is on standby. The hydrocyclone battery is composed of 05 hydrocyclones, the underflow of the hydrocyclones is stored in 02 silos of 220 and 240 m$^3$ capacity, to later be sent to an underground mine to be used as hydraulic fill. The hydrocyclone overflow is diverted to transfer tank "A" where it is mixed with water from the underground mine. The overflow mixture plus underground mine water (slurry) is sent to the DCT Deep Cone thickener 17 m in diameter × 21 m in height through 01 pumping line (12″ diameter HDPE pipe) composed of 03 pumps in a series of 200 HP power each [44].

The thickened fine tailings or slimes are transported by pipeline to a tailings filtering plant composed of three press-type filters. In this facility, the filtered tailings produced at

4200 mtpd will reach a moisture content of the order of 16% (Figure 75). The tailings are temporarily stockpiled in the filter plant area. The loading of the tailings is by means of a front loader, and for transport, trucks of 15 m$^3$ and 30 tonnes of load capacity are used [44].

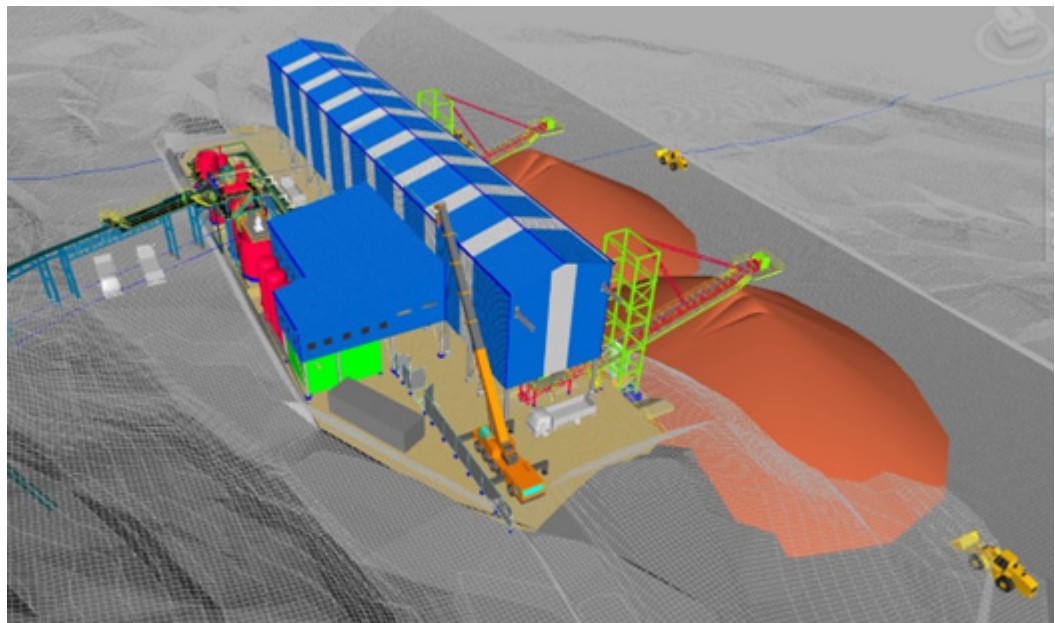

**Figure 75.** Chungar Tailings Filtering Plant.

The tailings deposited with a natural humidity of 16% are spread and will go through a minimum desiccation cycle until reaching a humidity content equal to or better than 14%. The tailings are mixed with mine waste in terraces, then the mixed material is compacted to a minimum density of 95% of the Standard Proctor [19] and humidity of 12%. The surface of the final platform and the sidewalks of the tailings deposit are built with a 1% slope towards both abutments of the deposit, so that rainwater on the platform can run off to an area where it can be quickly evacuated from the reservoir tailings deposit area (Figures 76 and 77) [44].

Filtered tailings disposal requires a suitable climate in which to operate. In wet climates like Chungar Mine, care must be exercised to ensure that the "dry stack" does not resaturate upon exposure to the elements (atmospheric precipitation). It is common practice to design dry stack facilities to contain filtered tailings that are less dense or have a high moisture content in a separate, non-structural area (Figure 76) [44].

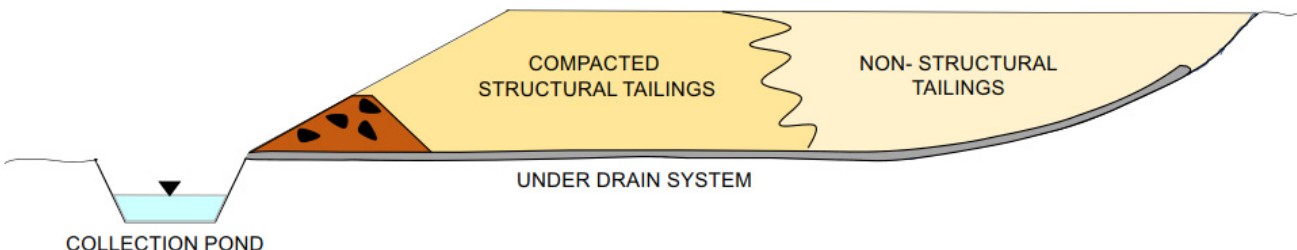

**Figure 76.** Chungar Filtered TSF Disposal Strategy.

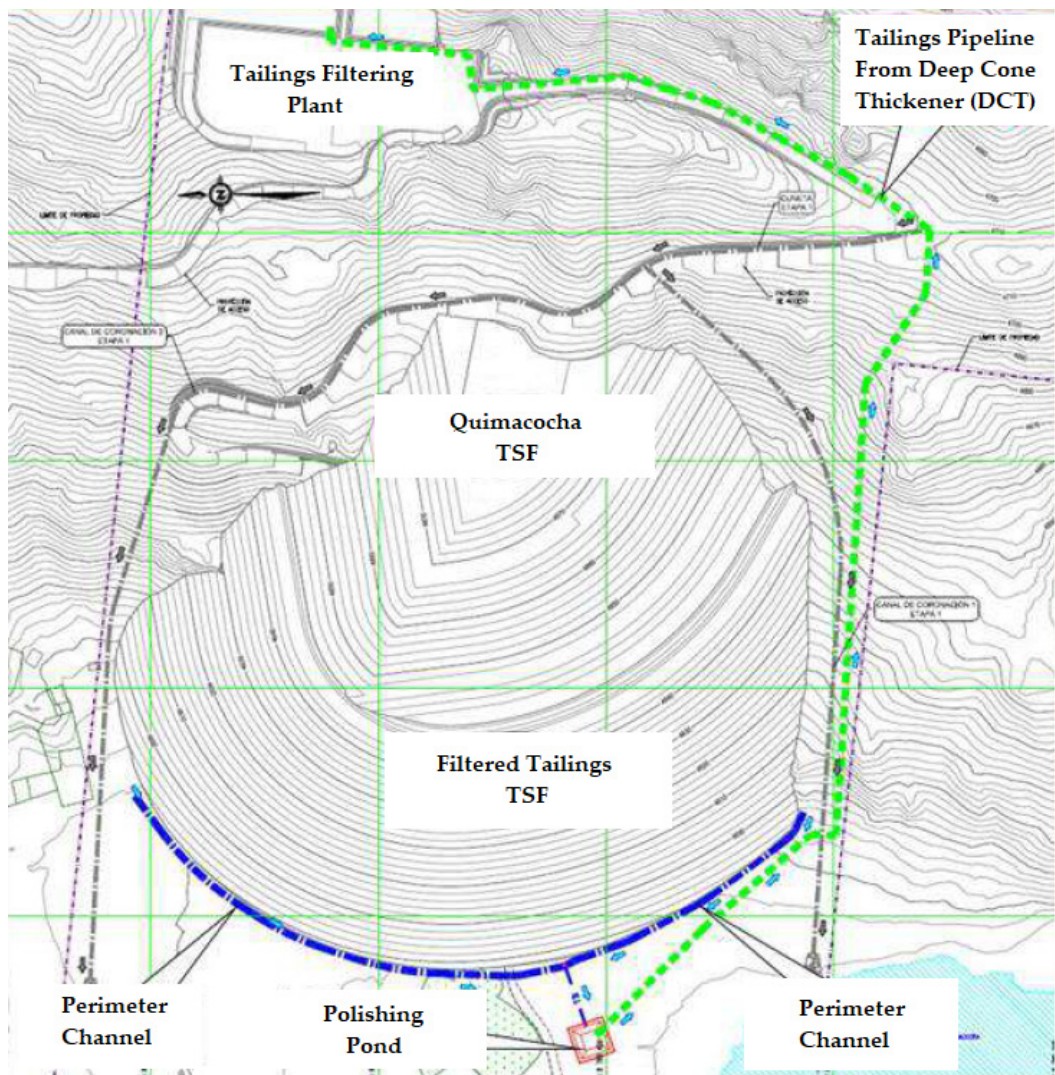

**Figure 77.** Chungar Filtered TSF Layout View.

Filtered tailings is a technology that is environmentally friendly, since it generates negligible seepages, decreasing the risk of transportation of contaminants significantly [12,14,16]. However, a relevant feature to consider is to avoid dust or fine particulate entrainments carried by local winds throughout the life of the TSF. Dust emission controls will play an important role as good design and proper implementation will provide the primary control mechanism for dust in accordance with regulatory air quality requirements [73]. Some dust control alternatives to implement are: soil cover (borrow material), topsoil/revegetation cover, phytostabilization, binder material, or chemical agglomeration [11].

*7.18. Geochemical Stability*

The lack of a tailings supernatant pond, very low seepage from the unsaturated tailings, and high degree of structural stability, allow dry stacks TSFs to develop progressive reclamation in many instances, a closure cover material is provided to manage runoff erosion, and create an appropriate ground surface for project reclamation with a proper construction machinery and trucks trafficability [3,22].

Thus, a cover with granular materials and geosynthetic or topsoil is projected to cover the terraces of filtered tailings typically–this being done during the regular TSF buildup operations, constituting a progressive closure activity. This kind of project typically considers the placement of 0.5 m layers of a granular soil cover, both on the slopes as well as on the berms and terraces of the filtered tailings dry stack deposit. To promote the

geochemical stability of the filtered TSF it is important to provide a cover, avoiding the acid rock drainage (ARD) of the tailings due to exposure to oxygen and water [14].

## 8. Environmental Management, Geochemical Stability and Seepage Monitoring System

### 8.1. Seepage Monitoring System

The TSF closure costs can be considerable; particularly if the tailings contain potentially acid-generating behavior, degrading the strength properties of cycloned tailings sand dams and borrow dams, or if extensive reshaping of the structure is required to create a landform that will need to be stable in the long term. Some governments are addressing the issue by requiring mining companies to establish some form of bonding arrangement to guarantee the safe and successful decommissioning of the facilities at no cost to the community, which is a closure plan which includes the direct, indirect, and post-closure operating, maintenance and surveillance costs for the TSFs. Filtered tailings with a high compaction degree can be placed in a relatively dense state, allowing an aggressive use of the TSF site on any type of topography, and reducing the TSF footprint [74].

The following Table 4 presents a closure cost estimate comparison between conventional TSF and filtered TSF.

**Table 4.** Conventional/Filtered TSF Closure Cost Estimate–Comparative Analysis [73,74].

| TSF Characteristics(M: Millions) | Flat Topography Configuration | | Valley Topography Configuration | |
|---|---|---|---|---|
| | **Conventional TSF** | **Filtered TSF** | **Conventional TSF** | **Filtered TSF** |
| Throughput/Mine lifetime (mtpd/years) | 50,000/20 | 50,000/20 | 10,000/20 | 10,000/20 |
| Tailings Dry Density (t/m$^3$) | 1.3 | 2.0 | 1.3 | 2.0 |
| TSF Capacity/Footprint (Mt/Mm$^3$/Km$^2$) | 347/267/8 | 347/173/4 | 208/160/2.0 | 208/104/1.5 |
| Number of TSF Sites (Impoundment/Dry Stack) | 2/0 | 0/1 | 1/0 | 0/1 |
| Max. TSF Height (Dam or Dry Stack) (m) | 35 | 65 | 150 | 120 |
| Total Closure Cost (MUS$) | 122 | 55 | 47 | 20 |

The above table shows that the estimated closure cost of conventional TSFs is double compared with filtered TSFs. This has a clear impact on aspects of financial guarantee for decommissioning activities and sustainable mining development.

Filtered TSF offers progressive reclamation and mine closure activities during the TSF operation, controlling dust with cover material placement, TSF side slope reclamation, and revegetation (if required) such as the use of the Phytostabilization technique. The combination of these technologies decreases TSF seepage to very low levels, almost avoiding tailings leachate, and reducing the risk of geochemical contamination. Under dry climate conditions, no supernatant TSF ponds need to be monitored during operation, and no freeboard requirement of tailings dams, reducing water losses significantly and eliminating an overtopping event risk, respectively [1–3].

A filtered tailings TSF solution is very competitive today compared to conventional technologies because the cost of closure and post-closure are lower, the environmental and construction permits are easier to obtain, and the authorities and community are more prone to accept it [11].

## 9. New Trends

Filter suppliers are working to improve equipment reliability and capacity. Nowadays some equipment reaches an 8000 mtpd production rate, significant advancements have been achieved, and more filter plants with high production rates will be operating in the next years around the globe. For tailings production over 50,000 mtpd, it is possible to

combine cycloning–high thickening–filtered plant with cyclones, thickeners, and filters, classifying the total tailings to filter the underflow coarse tailings and thicken the overflow fine tailings, providing flexibility and versatility against ore characteristic variations, such as the successful case of Mantos Blancos project [68].

An alternative process to obtain filtered tailings can be used, consisting of the recovery of the coarse fraction of tailings (cycloned tailings sand) through two cycl
oning stages followed by a drainage stage in dewatering vibratory screens to reduce tailings moisture and turn it into a paste to be easily transported to the adjoining dumping facility (Figure 78) [67].

New trends include the development of new agglomerating agents for filtered tailings. Regarding developments with conveyor systems, a new generation of "pipe" or "tubular" conveyor belts is a viable alternative to use with filtered tailings, allowing a confined transport that does not allow liberate dust emissions in very windy areas [11].

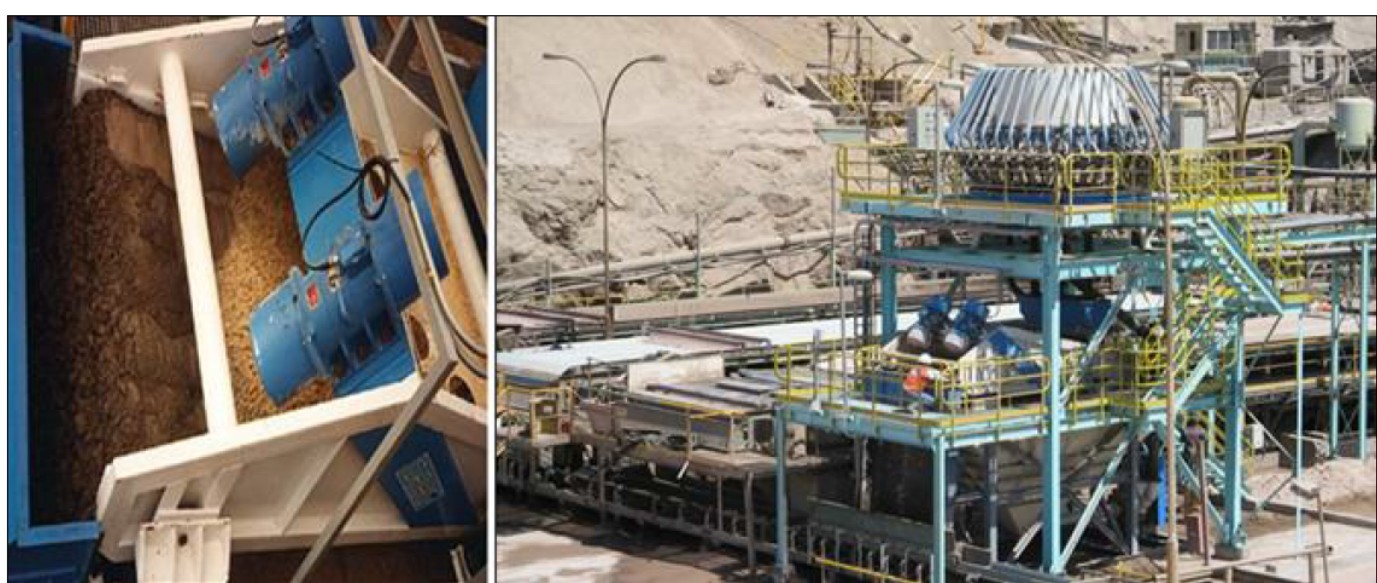

**Figure 78.** Mantos Blancos Cycloning Plant and Vibratory Screens for Tailings [68].

## 10. Discussions—Strengths and Limitations of Filtered Tailings from Practical Experiences Presented

The Table 5 presents the strengths and limitations of filtered tailings technology based on the practical experience recorded in Chile and Peru.

**Table 5.** Strengths and Limitations considering Practical Experience of Filtered Tailings in Chile and Peru.

| Strengths | Limitations |
|---|---|
| May permit storage in areas with steep topography, space restrictions, or other circumstances where conventional tailings facility is impractical. | May not be applicable to all tailings types in all situations (tailings with high clay-sized particle content are more challenging to filter to required moisture contents). |
| Highest water recovery during processing is advantageous in regions where water is scarce. | Filters or filter plants require more operational attention and are subject to system "upsets" due to ore variability, gradation, or operator attention. |
| More flexibility for placement strategy and final configuration, which can allow for progressive reclamation. | Additional storage ponds are required in the case of system upset where filtering must stop. |
| More flexibility for placement strategy and final configuration, which can allow for progressive reclamation. | Trafficability of filtered tailings surfaces can be a challenge depending on tailings moisture contents from the filter plant and climate conditions. |
| Most amenable to dry closure and landform development. | More challenging placement in wet and cold climates. Covered sheds are typically required for periods of wet weather when placement must stop. |
| Failure of physical stability, if it occurs, would likely be local slumping and consequences would be restricted to the local area. | |
| Approval and good reputation by the communities neighboring the mining projects. | |

## 11. Conclusions

The filtered dry stacked tailings cases presented in this paper show that there is important potential to achieve a sustainable tailings management solution in desert areas such as the north of Chile, and the south of Peru, among others. The important benefits offered by this technology are: (i) less total cost of a project over the mine lifetime (construction + operation + closure liability) mainly reducing the makeup water supply costs, (ii) failure risks reduction in seismic zones improving TSF physical stability, (iii) maximum water recovery, and (iv) an environmentally friendly solution allowing for smaller TSF footprints, a progressive reclamation, and effective dust control, allowing EIAs approvals.

These benefits are a good reason to promote new or existing large mining operations to shift from conventional slurry tailings disposal facilities to alternative solutions with highly dewatered tailings disposal facilities, which are day by day more accepted by the environmental authority and communities.

The application of this technology is an excellent alternative providing attributes to streamline and permitting instruments (EIAs and Closure Plans, and besides this, tailings management technology will have lower closure costs).

These are major challenges for engineers, scientists, vendors, and mining operators. To date, we have not solved all our difficulties and there are many interesting solutions waiting to be found.

**Author Contributions:** Conceptualization, C.C.V.: and G.P.C.; formal analysis, C.C.V.; investigation, C.C.V.; resources, G.P.C.; writing—original draft preparation, C.C.V.; writing—review and editing, C.C.V.; visualization, C.C.V.; supervision, G.P.C. All authors have read and agreed to the published version of the manuscript.

**Funding:** This research is funded by the Research Department of Catholic University of Temuco, Chile.

**Data Availability Statement:** The data presented in this study are available on request from the corresponding author.

**Conflicts of Interest:** The authors declare no conflict of interest.

## Abbreviations

| | |
|---|---|
| TSF | Tailings Storage Facility |
| ASTM | American Society for Testing and Materials |
| MSC | Mobile Stacker Conveyor |
| Cw | Slurry tailings solids content by weight |
| mtpd | Metric tonnes per day |
| QA/QC | QA (Quality Assurance) and QC (Quality Control) |
| CCD | Counter Current Decantation |
| TTD | Thickened Tailings Disposal |
| masl | Meters above sea level |
| PLC | Programmable Logic Controller |
| PSD | Particle Size Distribution |
| OMS | Operating, Maintenance, and Surveillance |

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
