# Peer review of "Practical Experience of Filtered Tailings Technology in Chile and Peru: An Environmentally Friendly Solution"

_minerals, doi:10.3390/min12070889_

Round 1

Reviewer 1 Report

A very detailed overview with diagrams of the application of dry filtrtion techniques is presented.

No comments

Reviewer 2 Report

This is certainly a good review of filtered tailings procedures and installations. This leads one to look with great interest at the references. Some of the references, especially the environmental impact statements, are not dated in the reference list, so one must go to the source to see if these are recent. In the text, the installation descriptions are often in future tense ("will be") as it appears much of the information comes from these environmental impact statements. However, it seems that the facilities have been built by looking at the photos. This is confusing. For examples, see page 29 beginning with paragraph 3, page 33, page 39 (For this, the following processes WILL BE implemented:), etc.

Is it possible to have a section discussing any issues that these facilities have incurred or any successes? This would certainly add to the review.

It should be noted that section 2.2 should be Filter Presses, not Vacuum Filters. Also, in this section, it indicates that filter presses are not sensitive to high percentages of <74 micrometer clay minerals. This is not correct. They are sensitive to these fine clays. Fine clays are issues for all dewatering technologies.

Page 43, paragraph 4 appears to be missing some text at the beginning.

Some of the figures, most notably Figure 5, are not of sufficient quality.

Reviewer 3 Report

This is a large and informative review of mining waste disposal sites. At the same time, the large number of references to the works of other authors, and the almost complete absence of the author's own research gives the impression that this corresponds more to a well-illustrated section of a training course, or a review section of a thesis, which should be followed by the author's own research and developments.

There is practically no paragraph, figure, or table that does not contain references to the work of other authors.

Some key sections of the manuscript are completely abridged versions of other already published works. At the same time, it is not possible to check a significant part of the cited references, as they are not available in the public domain. Therefore, I believe that this manuscript does not fit the format of this academic journal and cannot be published.

Round 2

Reviewer 3 Report

This review was very interesting for me personally and will undoubtedly be very useful to a scientific audience, especially considering the direction of the special issue of the journal. I missed the authors' own research, but as a review article demonstrating the results of waste disposal in this region of the world, it is certainly interesting.
